# From word models to executable models of signaling networks using automated assembly

Benjamin M Gyori[1,†] iD, John A Bachman[1,†] iD, Kartik Subramanian[1], Jeremy L Muhlich[1], Lucian Galescu[2] & Peter K Sorger[1,*] iD

## Abstract

Word models (natural language descriptions of molecular mechanisms) are a common currency in spoken and written communication in biomedicine but are of limited use in predicting the behavior of complex biological networks. We present an approach to building computational models directly from natural language using automated assembly. Molecular mechanisms described in simple English are read by natural language processing algorithms, converted into an intermediate representation, and assembled into executable or network models. We have implemented this approach in the Integrated Network and Dynamical Reasoning Assembler (INDRA), which draws on existing natural language processing systems as well as pathway information in Pathway Commons and other online resources. We demonstrate the use of INDRA and natural language to model three biological processes of increasing scope: (i) p53 dynamics in response to DNA damage, (ii) adaptive drug resistance in BRAF-V600E-mutant melanomas, and (iii) the RAS signaling pathway. The use of natural language makes the task of developing a model more efficient and it increases model transparency, thereby promoting collaboration with the broader biology community.

**Keywords** computational modeling; natural language processing; signaling pathways

**Subject Categories** Computational Biology; Methods & Resources; Signal Transduction

**Mol Syst Biol. (2017) 13: 954**

See the Glossary for abbreviations used in this article.

## Introduction

Biophysics and biochemistry are the foundations of quantitative reasoning about biological mechanisms (Gunawardena, 2014a). Historically, systems of biochemical mechanisms were described in reaction diagrams (familiar graphs involving forward and reverse arrows) and analyzed algebraically. As such systems became more complex and grew to include large networks in mammalian cells, word models (natural language descriptions) became the dominant way of describing biochemical processes; word models are frequently illustrated using pictograms and informal schematics. However, formal approaches are generally required to understand dynamics, multi-component switches, bistability, etc. Dynamical models and systems theory have proven extremely effective in elucidating mechanisms of all-or-none response to apoptosis-inducing ligands (Rehm *et al*, 2002; Albeck *et al*, 2008), sequential execution of cell cycle phases (Chen *et al*, 2004), the interplay between stochastic and deterministic reactions in the control of cell fate following DNA damage (Purvis *et al*, 2012), drug sensitivity and disease progression (Lindner *et al*, 2013; Fey *et al*, 2015), bacterial cell physiology (Karr *et al*, 2012), the responses of ERK kinase (Chen *et al*, 2009) and the NF-κB transcription factor (Hoffmann *et al*, 2002) to environmental stimuli, and similar biological processes. The challenge arises in linking a rich ecology of word models to computational representations of these models that can be simulated and analyzed. The technical environments used to create and explore dynamical models remain unfamiliar to many biologists, and a substantial gap persists between the bulk of the literature and formal systems biology models.

A variety of methods have been developed to make mechanistic modeling more powerful and efficient. These include fully integrated software environments (Loew & Schaff, 2001; Hoops *et al*, 2006), graphical formalisms (Kolpakov *et al*, 2006; Le Novère *et al*, 2009), tabular formats (Tiger *et al*, 2012), high-level modular and rule-based languages (Danos *et al*, 2009; Mallavarapu *et al*, 2009; Smith *et al*, 2009), translation systems for generating Systems Biology Markup Language (SBML) models from pathway information (Ruebenacker *et al*, 2009; Büchel *et al*, 2013), and specialized programming environments such as PySB (Lopez *et al*, 2013). In addition, the BioModels database has provided a means to retrieve and reuse existing models (Juty *et al*, 2015). Such tools have increased transparency and reusability but not sufficiently to bridge the gap between verbal descriptions and computational models.

1  Laboratory of Systems Pharmacology, Harvard Medical School, Boston, MA, USA
2  Institute for Human and Machine Cognition, Pensacola, FL, USA
   *Corresponding author. Tel: +1 617 432 6901/6902; E-mail: peter_sorger@hms.harvard.edu
   †These authors contributed equally to this work

## Glossary

**Application programming interface (API)**
a standardized interface by which one software system can use services provided by other software, often remotely; in the current context, INDRA accesses NLP systems and pathway databases via APIs. INDRA exposes an API that other software can build upon. API is used here interchangeably with *Interface* (e.g., INDRA's TRIPS *Interface*).

**Assembler**
a module in INDRA that constructs a model, network, or other output from INDRA *Statements*.

**Executable model**
a computational model that can be simulated to reproduce the observable dynamical behavior of a system; often, but not always, a system of linked differential equations.

**Extraction knowledge base (EKB)**
a collection of events and terms relevant to molecular biology that is the result of natural language processing with TRIPS (Box 1).

**Grounding**
a sub-task of NLP related to NER which assigns unique identifiers to named entities in text by linking them to ontologies and databases; in the current context, this involves creating links to databases such as UniProt, HGNC, GO, or ChEBI.

**Knowledge representation**
a formalism that allows aggregation of information, potentially from multiple sources, in a standardized computable format; in the current context, INDRA *Statements* serve as a common knowledge representation for mechanistic information.

**Logical form (LF)**
a graph representing the meaning of a sentence; an intermediate output of natural language processing in the TRIPS system (Box 1).

**Model assembly**
the process of automatically generating a model in a given computational formalism from an intermediate knowledge representation; in our context from INDRA *Statements*.

**Molecular mechanism**
used in this paper to refer to processes involved in changing the state of a molecular entity or in describing its interaction with another molecular entity as represented by a set of linked biochemical reactions. Descriptions of mechanisms are common in the biomedical literature and key assertions are captured in databases in formats such as BioPAX. The information we extract from such descriptions are interchangeably referred to as *mechanistic information*, *mechanistic assertions*, *mechanistic facts*, and *mechanistic findings*.

**Named entity recognition (NER)**
a sub-task of NLP concerned with the recognition of special words in a text that are not part of the general language; in the current context, NER is used to identify proteins, metabolites, drugs, and other terms (which are generally referred to as *named entities*).

**Natural language (NL)**
language that humans commonly use to communicate in speech and writing; in the current context, restricted to the English language.

**Natural language processing (NLP)**
the algorithmic process by which a computer interprets natural language text.

**Policies**
user-defined settings that affect the automated assembly process.

**Processor**
a module in INDRA that constructs INDRA *Statements* from a specific input format.

**Template extraction**
the process by which INDRA *Processors* extract INDRA *Statements* from various input formats.

To date, most attempts to make modeling more accessible have focused on graphical interfaces in which users draw reaction diagrams that are then used to generate equations. This approach is attractive in principle, since informal diagrams are a mainstay of most scientific presentations, and schematic diagrams are essential in engineering, but it has proven difficult in practice to accommodate the simultaneous demands of accurately rendering individual biochemical reactions while also depicting large numbers of interacting components. It is particularly difficult to create graphical interfaces that model the combinatorially complex reactions encountered in animal cell signaling (Stefan *et al*, 2014).

In this paper, we explore the idea that natural language can serve as a direct input for dynamical modeling. Natural language has many benefits as a means of expressing mechanistic information: In addition to being familiar, it can concisely capture experimental findings about mechanisms that are ambiguous and incomplete. Extensive work has been performed on the use of software to convert text into computable representations of natural language, and such natural language processing (NLP) tools are used extensively to mine the scientific literature (Krallinger *et al*, 2012; Fluck & Hofmann-Apitius, 2014). To our knowledge however, natural language has not been widely used as a direct input for mechanistic modeling of biological or chemical processes. A handful of studies have explored the use of formal languages resembling natural language for model creation (Kahramanoğullari *et al*, 2009; Wasik *et al*, 2013) but these systems focus on capturing low-level reaction mechanisms and require that descriptions conform to a precisely defined syntax.

Three technical challenges must be overcome to convert natural language into executable models. The first is reading text with a machine in a manner that reliably identifies mechanistic assertions in the face of variation in how they are expressed. The second is designing an intermediate knowledge representation that captures often-ambiguous and incomplete mechanisms without adding unsubstantiated assumptions (thereby implementing the rule: "don't know, don't write"). This intermediate representation must be compatible with existing machine-readable sources of network information such as pathway databases. The third challenge is translating mechanistic assertions from the intermediate representation into executable models involving different mathematical formalisms and levels of detail; this involves supplying necessary assumptions left out of the original text.

The method and software tool described in this paper, the Integrated Network and Dynamical Reasoning Assembler (INDRA), addresses these challenges and makes it possible to construct different types of executable models directly from natural language and fragmentary information in pathway databases. In contrast to previous approaches to incorporating natural language in models, INDRA can accommodate flexibility in style and syntax through the use of NLP algorithms that normalize variability in expression into logical forms that effectively represent the underlying meaning (Box 1). Mechanisms extracted from natural language and other sources are converted into *Statements* (the INDRA intermediate representation) and then translated into one of several types of models depending on the specific use case. We describe this process

in some detail because it relates directly to how we understand and communicate biological mechanisms in papers and conversations. The essential challenge is converting the informality and ambiguity of language, which is frequently a benefit in the face of incomplete information, into a precise set of statements (or equations) needed for an executable mathematical model.

As a test case, we show that INDRA can be used to automatically construct a model of p53 dynamics in response to DNA damage from a few simple English statements; we show that the qualitative behavior of the INDRA model matches that of an existing mathematical model constructed by hand. In a second, more challenging test, we show that an ensemble of models of the MAP kinase pathway in cancer cells can be built using literature-derived text describing the interaction between BRAF[V600E] and drugs used to treat melanoma. Finally, we use natural language and INDRA to assemble a large-scale model of the RAS pathway as defined by a community of RAS biology experts; we show how this model can be updated using sentences gathered from the RAS community.

## Results

### INDRA decouples the curation of mechanistic knowledge from model implementation

A core concept in INDRA is that the identification, extraction, and regularization of mechanistic information (curation) is a distinct process from model assembly and implementation. Mechanistic models demand a concrete set of assumptions (about catalytic mechanisms, stoichiometry, rate constants, etc.) that are rarely expressed in a single paper or molecular interaction entry stored in a database. Models must therefore combine relatively general assertions about mechanisms extracted from available knowledge sources (e.g., that enzyme E "activates" substrate S) with information or assumptions about molecular details (e.g., that the enzyme acts on the substrate S in a three-step ATP-dependent mechanism involving an activating site on the substrate) derived from general knowledge about biochemistry and biophysics. Precisely how such details are constructed depends on the requirements of the mathematical formalism, the specific biological use case, and the nature of the hypothesis being tested. A similar concept was recently introduced for rule-based modeling in Basso-Blandin *et al* (2016) and in the context of graphical model diagrams in O'Hara *et al* (2016). In both works, the authors make a distinction between the curation and representation of mechanistic knowledge and its executable implementation.

Text-to-model conversion in INDRA involves three coupled steps. First, text is processed into a machine-interpretable form and the identities of proteins, genes, and other biological entities are *grounded* in reference databases. Second, the information is mapped onto an intermediate knowledge representation (INDRA *Statements*) designed to correspond in both specificity and ambiguity to descriptions of biochemistry as found in text (e.g., "*MEK1 phosphorylates ERK2*"). Third, the translation of this intermediate representation into concrete reaction patterns and then into executable forms such as networks of ordinary differential equations (ODEs) is performed in an *assembly* step. In this process, *Statements* capture mechanistic information available from the

knowledge source without additions or assumptions, deferring interpretations of specific reaction chemistry that are often unresolved by the knowledge source but must be made concrete to assemble a model.

### Information flow from natural language input to a model

The three steps in text-to-model conversion are implemented in a three-layer software architecture. An input layer comprising *Interface* and *Processor* modules (Fig 1A, block 1) is responsible for communicating with language processing systems (e.g., the TRIPS NLP system, see Box 1) and pathway databases (e.g., the Pathway Commons database) to acquire information about mechanisms. An intermediate layer contains the library of *Statement* templates (Fig 1A, block 2), and an output layer contains *Assembler* modules that translate *Statements* into formats such as networks of differential equations or protein–protein interaction graphs (Fig 1A, block 3). INDRA is written in Python and available under the open-source BSD license. Source code and documentation are available at http://indra.bio; documentation is also included in the Appendix.

As an example of text being converted into an executable model, consider the sentence "*MEK1 phosphorylates ERK2 at threonine 185 and tyrosine 187*". Figure 1B shows eight lines of Python code implementing this example; the numbers alongside each code block correspond to the three layers of the INDRA architecture in Fig 1A and implement the flow of information between the user, INDRA, and external tools shown in Fig 1C. The user first enters the sentence to be processed and calls the *process_text* command in the INDRA TRIPS *Interface.* This function sends a request to the web service exposed by the TRIPS NLP system (Allen *et al*, 2015; Fig 1B and C, block 1). INDRA can also call on the REACH NLP system, which has complementary capabilities (Valenzuela-Escarcega *et al*, 2015), but in this paper we focus exclusively on TRIPS. TRIPS parses the text into its *logical form* (Box 1, Appendix Fig S1A) and then extracts mechanisms relevant to molecular biology into an extraction knowledge base (EKB; Box 1, Appendix Fig S1B). Included in this process are entity recognition and grounding whereby MEK1 is recognized as a synonym of the HGNC gene name MAP2K1 and grounded to UniProt Q02750, and Erk2 is grounded to MAPK1 and UniProt P28482. These terms are explained in Box 1, in Appendix Section 2.1, and in Allen *et al* (2015). The TRIPS *Processor* in INDRA extracts *Statements* directly from the EKB output returned by TRIPS (Fig 1B and C, block 2). The translation of *Statements* into concrete models is performed by an INDRA *Assembler*. In this example, a PySB *Assembler* was used to build a rule-based model in PySB (Lopez *et al*, 2013) and generate an SBML-compatible reaction network (Fig 1B and C, block 3). Because the Phosphorylation *Statements* in this example are compatible with multiple concrete reaction patterns, the user specifies a *policy* for assembly: Here, we used the "two-step" policy, which implements phosphorylation with reversible enzyme–substrate binding (polices are described below). The resulting reaction network was instantiated as a set of ODEs and simulated using default parameter values to produce the temporal dynamics of all three phosphorylated forms of ERK2 (labeled MAPK1; Fig 1C, bottom right). The same rule-based model can also be analyzed stochastically using network-free simulators (Danos *et al*, 2007b; Sneddon *et al*, 2011).

**Box 1:    Natural language processing using TRIPS**

To convert text into computable representations that capture syntax and semantics, INDRA uses external NLP software systems exposed as web services. This paper focuses on DRUM (Deep Reader for Understanding Mechanisms; http://trips.ihmc.us/parser/cgi/drum), which is a version of the general-purpose TRIPS NLP system customized for extracting biological mechanisms from natural language text. TRIPS has been developed over a period of decades and used for natural language communication between humans and machines in medical advice systems, robotics, mission planning, etc. (see, for example, Ferguson & Allen, 1998; Chambers et al, 2005; Allen et al, 2006).

The first step in processing natural language with TRIPS is a "shallow" or syntactic analysis of text to identify grammatical relationships among words in a sentence, recognize named entities such as proteins, amino acids, small molecules, cell lines, etc., and link these entities to appropriate database identifiers (the process of *grounding*). TRIPS uses this information to perform a "deep" semantic analysis and try to determine the meaning of a sentence in terms of its logical structure. This process draws on a general-purpose semantic lexicon and ontology that defines a range of word senses and semantic relations among words. The output of this process is represented as a logical form (LF) graph (Manshadi et al, 2008). The LF graph represents the sense of each word (e.g., "protein") and captures the semantic roles of relevant arguments (e.g., "affected") for each predicate (e.g., "activation"). The LF also represents tense, modality, and aspect information—information that is crucial for determining whether a statement expresses a stated fact, a conjecture, or a possibility.

The LF graph is then transformed into an extraction knowledge base (EKB) containing extractions relevant for the domain, in this case molecular biology. LF graphs compactly represent and normalize much of the variation and complexity in sentence structure; EKBs can therefore be extracted from the LF using a relatively small set of rules. The EKB is an XML file containing entries for *terms* (e.g., proteins, drugs), *events* (e.g., activation, modification) involving those terms, and higher-level *causal relations* between the events. The EKB also contains additional information such as the text from which a given term or event was constructed.

A more thorough technical description of TRIPS/DRUM is given in Appendix Section 2.1 and in Allen et al (2015); a broader overview of NLP systems can be found in Allen (2003).

## INDRA Statements represent mechanisms from multiple sources

Integrated Network and Dynamical Reasoning Assembler *Statements* serve as the bridge between knowledge sources and assembled models, and we therefore describe them in detail. *Statements* are implemented as a class hierarchy that groups related mechanisms; a Unified Modeling Language (UML) diagram of existing *Statement* classes is shown in Appendix Fig S2. Each INDRA *Statement* describes a mechanism involving one or more molecular entities, along with information specific to the mechanism and any supporting evidence drawn from knowledge sources. For example, the phosphorylation *Statement* shown schematically in Fig 2A contains references to enzyme and substrate *Agents* (which in this case refers to MAP2K1 and MAPK1, respectively), the phosphorylated residue and position on the substrate, and one or more *Evidence* objects with supporting information. An *Agent* is an INDRA object that captures the features of the molecular state necessary for a participant to take part in a molecular process (Fig 2B). This includes necessary post-translational modifications, bound cofactors, mutations, cellular location, and state of activity (Fig 2B and Appendix Fig S4). Agents also include annotations that *ground* molecular entities to unique

identifiers in one or more databases or ontologies (e.g., HGNC, UniProt, ChEBI; Fig 2B). *Evidence* objects contain references to supporting text, citations, and relevant experimental context (Fig 2C).

An important feature of both *Statements* and *Agents* is that they need not be fully specified. If there is no information in the source pertaining to a specific detail in a *Statement* or *Agent*, then the corresponding entry is left blank; this is an example of the "don't know, don't write" principle. INDRA and the rule-based models it generates are designed to handle information that is incomplete in this way. For example, the Phosphorylation *Statement* shown in Fig 2A indicates that the phosphorylation of substrate MAPK1 *can occur* when the enzyme MAP2K1 is phosphorylated at serine residues S218 and S222, but other aspects of the state of MAP2K1 are left unspecified (e.g., whether MAP2K1 is phosphorylated at S298, or bound to a scaffold protein such as KSR). *Statements* capture the ambiguity inherent in the vast majority of statements about biological processes, thereby permitting multiple interpretations: For example, phosphorylation of MAP2K1 at S218 and S222 could be necessary and sufficient for activity against MAPK1, necessary but not sufficient, sufficient but not necessary, or neither sufficient nor necessary, depending on other molecular context outside the scope of the *Statement*. The ability of *Statements* to capture knowledge from input sources while making as few additional assumptions as possible is an essential feature of the text-to-model conversion process. It also conforms closely to the way individual experiments are described and interpreted since single experiments investigate only a subset of the facts pertaining to a biochemical mechanism and its implementation in a model. The ambiguity in *Statements* is resolved during the *assembly* step by explicitly declaring assumptions and generating a fully defined executable model.

Users can inspect INDRA *Statements* in several complementary ways: (i) by inspecting *Statements* as Python objects, (ii) by rendering *Statements* visually as graphs (Appendix Fig S3A), and (iii) by serializing *Statements* into a platform-independent JSON exchange format (Appendix Fig S3B). The semantics of INDRA *Statements* as well as the semantics describing the role that *Agents* play in each INDRA *Statement* are grounded in the Systems Biology Ontology (SBO; Courtot et al, 2011) facilitating integration and reuse in other applications. These capabilities are demonstrated in Appendix Notebook 1.

## Normalized extraction of findings from diverse inputs using mechanistic templates

The principal technical challenge in extracting mechanisms from input sources is identifying and normalizing information contained in disparate formats (e.g., BEL, BioPAX, TRIPS EKB) into a common form that INDRA can use. INDRA queries input formats for patterns corresponding to existing *Statement* types (templates), matching individual pieces of information from the source format to fields in the *Statement* template. This procedure is implemented for each type of input, making it possible to extract knowledge in a consistent form. Template matching does not guarantee that every mechanism found in a source can be captured by INDRA, but it does ensure that when a mechanism is recognized, the information is captured in a normalized way that enables downstream model assembly. The process is therefore configured for high precision at the cost of lower recall.

INDRA implements template-matching extraction for each input format using a set of *Processor* modules. In the case of natural

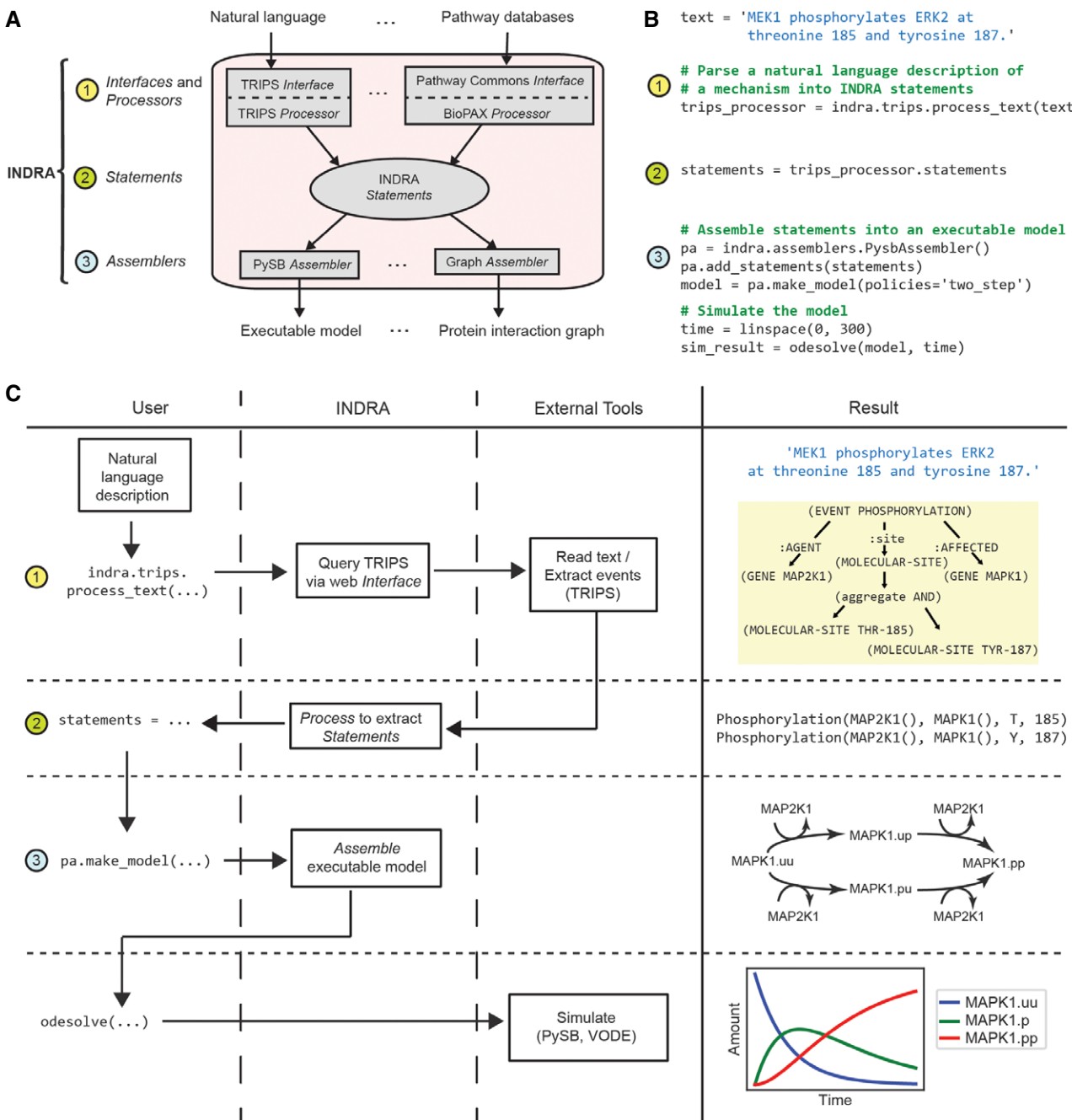

**Figure 1. Building a model from natural language with INDRA.**

A   The architecture of INDRA consists of three layers of modules (1–3). In layer (1), interfaces collect mechanisms from natural language processing systems (e.g., TRIPS *Interface*) and pathway databases (e.g., Pathway Commons *Interface*) and Processors (e.g., TRIPS *Processor*, BioPAX *Processor*) extract INDRA *Statements* from their outputs. Statements, the internal representation in INDRA, constitute layer (2). In layer (3), INDRA *Statements* are assembled into various model formats by *Assembler* modules (e.g., PySB *Assembler*, Graph *Assembler*).

B   A Python script is used to assemble and simulate a model from the *text* "MEK1 phosphorylates ERK2 at threonine 185 and tyrosine 187". The *process_text* method of INDRA's TRIPS *Processor* is called to send the text to the TRIPS NLP system (1) and then process the output of TRIPS to construct INDRA *Statements* (2). Then, a PySB *Assembler* is constructed, the *Statements* are added to it, and an executable model is assembled using the PySB *Assembler*'s *make_model* method with a "two-step" policy (3). Finally, the model is simulated for 300 s using PySB's *odesolve* function.

C   User input, INDRA modules, and external tools form a sequence of events to turn a natural language sentence into a model and simulation. The natural language description from the user is passed to INDRA's TRIPS *Interface*, which sends the text to TRIPS (1). The TRIPS system processes the text and creates an Extraction Knowledge Base graph (Results column; yellow box). INDRA receives the results from TRIPS and constructs two INDRA *Statements* from it, one for each phosphorylation event (Results column), which are returned to the user (2). The user then instantiates a PySB *Assembler* and instructs it to assemble an executable model (3) from the given INDRA *Statements* (a schematic biochemical reaction network shown in Results column). Finally, the user calls an ODE solver via PySB's *odesolve* function to simulate the model for 300 s (simulation output shown in Results column).

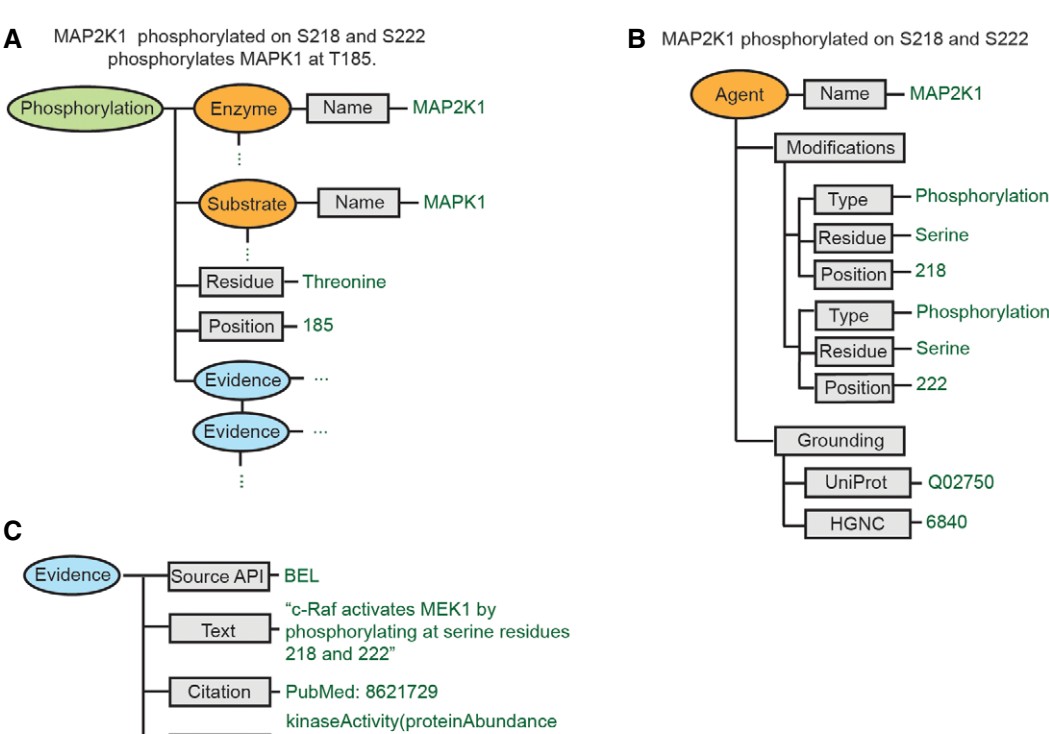

**Figure 2.   INDRA Statements represent molecular agents and biochemical mechanisms.**

A   The mechanism "MAP2K1 that is phosphorylated at S218 and S222 phosphorylates MAPK1 on T185" is represented in INDRA as a Phosphorylation *Statement* with an enzyme *Agent* (MAP2K1), a substrate *Agent* (MAPK1), a residue (threonine), and a position (185) argument. The state of the MAP2K1 *Agent* is expanded in panel (B). A *Statement* can have one or more *Evidences* associated with it, with an example expanded in panel (C).

B   The *Agent* representing "MAP2K1 that is phosphorylated at S218 and S222" has two modification conditions: serine phosphorylation at 218 and serine phosphorylation at 222. The grounding to the UniProt and HGNC databases associated with the *Agent* is also shown.

C   An *Evidence* object is shown which is associated with an INDRA *Statement* obtained from the BEL Large Corpus (see Box 2) as the source. The *Evidence* object represents the evidence text for the entry ("c-Raf activates MEK1 by phosphorylating at serine residues 218 and 222"), the citation associated with the entry (PubMed identifier 8621729), the original BEL statement (shown under Source ID) and any annotations that are available, including the organism (in this example, 9606, which is the identifier for *Homo sapiens*). In some cases, epistemic information is known about the *Statement*, such as whether it is an assertion or a hypothesis, and the *Evidence* object has a corresponding field to carry this information.

language, the EKB (see Glossary and Box 1) output from TRIPS serves as an input for the TRIPS *Processor* in INDRA. For a statement such as "*MAP2K1 that is phosphorylated on S218 and S222 phosphorylates MAPK1 at T185*", the EKB extraction graph (Fig 3, top left) has a central node (red text) corresponding to a *phosphorylation* event that applies to three *terms*: MAP2K1 as the agent for this event, MAPK1 as the entity affected by this event, and "*threonine-185*" playing the specific role of being the site where the event occurs (green text depicts the grounding in UniProt and HGNC identifiers). A second *phosphorylation* event (yellow box) involving S218/S222 of MAP2K1 is recognized by TRIPS as a nested property of MAP2K1 phosphorylation. It is a precondition for the primary *phosphorylation* event on *MAPK1*.

Integrated Network and Dynamical Reasoning Assembler establishes that this extraction graph corresponds to an INDRA Phosphorylation *Statement* and then exploits the fact that the template for such a *Statement* has entries for an enzyme, a substrate, a residue,

and a position (Fig 2A). The AGENT in the TRIPS EKB is identified as the enzyme which itself has a modification (phosphorylation) at specified positions (S218 and S222). The AFFECTED portion of the TRIPS EKB is identified as the substrate MAPK1. The extracted INDRA *Statement* collects this information along with target residue ("threonine") and position ("185") on the substrate. The end result is a biochemically plausible depiction of a specific type of reaction from a short fragment of text.

Extraction of a Phosphorylation *Statement* from databases using BioPAX or BEL follows the same general procedure. The INDRA BioPAX *Processor* uses graph patterns to search for reactions in which a substrate on the right-hand side gains a phosphorylation modification relative to the left-hand side (Fig 3, center left). The *Processor* identifies this as a phosphorylation reaction and constructs a Phosphorylation *Statement* for each such reaction that it finds.

In the case of BEL, statements consisting of subject–predicate–object expressions describe the relationships between molecular

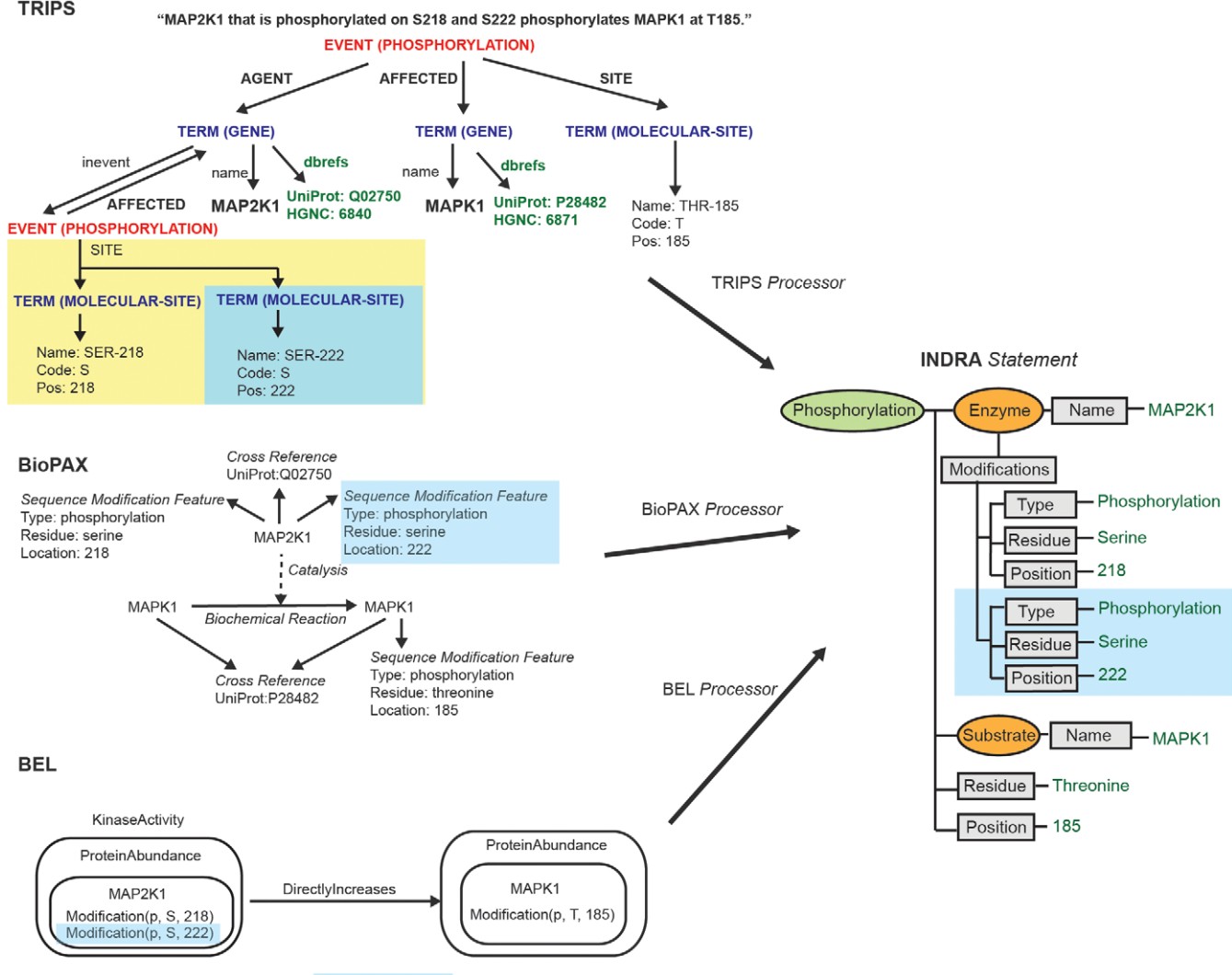

**Figure 3.   INDRA Statements constructed from TRIPS NLP extractions, BioPAX, and BEL.**

An identical INDRA *Statement* is constructed from three knowledge sources. A corresponding fragment of each source format (representing the phosphorylated state of MAP2K1 on S222) is highlighted in blue. Top left: A TRIPS EKB (see Box 1) graph is shown for the sentence "MAP2K1 that is phosphorylated on S218 and S222 phosphorylates MAPK1 at T185". The main phosphorylation *event* has *agent*, *affected*, and *site* arguments, each of them referring to a *term*. The *agent term* resolves to a *gene* with name MAP2K1 and database references to UniProt and HGNC. The MAP2K1 *term* also refers to an additional *event* in which it is *affected* (yellow background). This additional event represents the phosphorylated state at two molecular sites: serine 218 and serine 222. The *affected* term associated with the main phosphorylation event is MAPK1 with its associated UniProt and HGNC references. Finally, the site argument of the main event is a *molecular site* resolving to threonine 185. Middle left: A BioPAX *Biochemical Reaction* is shown with unmodified MAPK1 on the left-hand side and MAPK1 with a *Sequence Modification Feature* of phosphorylation at threonine 185 on the right-hand side. Both the left- and the right-hand sides use the same *Cross Reference* to a UniProt identifier. A *Catalysis* is associated with the *Biochemical Reaction* with MAP2K1 as the controller. MAP2K1 has two *Sequence Modification Features*: phosphorylation at serines 218 and 222. MAP2K1 also refers to a UniProt identifier via a *Cross Reference*. Two alternative visual representations of the same BioPAX *Reaction* are given in Appendix Fig S5. Bottom left: A graphical representation of a BEL statement is shown in which the *subject* is the *Kinase Activity* of the *Protein Abundance* of the modified MAP2K1 (with phosphorylations at serines 218 and 222). The *object* of the statement is the *Protein Abundance* of modified MAPK1 (phosphorylation at threonine 185) with the predicate being *Directly Increases*. Below the graphical representation, the statement is also given in *BEL script* format. Right: All example mechanisms from the three knowledge sources are constructed as the same INDRA *Phosphorylation Statement* with MAP2K1 as the enzyme (subject to modification conditions) and MAPK1 and the substrate. The *Evidence* associated with the INDRA *Statement* (not shown) constructed would be different for each knowledge source.

entities or biological processes (Box 2). INDRA's *BEL Processor* queries a BEL corpus (formatted as an RDF graph) for expressions consistent with INDRA *Statement* templates. For example, Phosphorylation *Statements* are extracted by searching for expressions in which the subject represents the kinase activity of a protein that

*directly increases* an object representing a modified protein (Fig 3, bottom left); *directly increases* is a predicate used when molecular entities interact physically. Triples that fit this pattern are extracted into an INDRA Phosphorylation *Statement* with the subject as the enzyme and the object as the substrate.

---

**Box 2: BioPAX and BEL**

BioPAX is a widely used format for describing biological interactions that facilitates exchange and integration of pathway information from multiple sources (Demir *et al*, 2010). BioPAX is the core exchange format underlying the Pathway Commons database, which aggregates information from over 20 existing sources including Reactome, NCI-PID, KEGG, PhosphoSitePlus, BioGRID, and Panther (Cerami *et al*, 2011). Pathway Commons provides a web service with an interface for submitting queries about pathways and recovering the result as a BioPAX graph; a query could involve finding all proteins and interactions in the neighborhood of a specified protein or finding all paths between two sets of proteins.

BioPAX employs a Web Ontology Language (OWL) knowledge representation centered around biochemical processes and reactants and is applicable to metabolic, signaling, and gene regulatory pathways. The representation of reactions in BioPAX is flexible: An arbitrary set of complexes and standalone molecules on the left-hand side of a reaction can produce complexes and molecules on the right-hand side subject to one or more catalytic controllers.

The Biology Expression Language (BEL) facilitates the curation of knowledge from the literature in a machine-readable form. While BioPAX is designed to capture direct, molecular interactions, BEL can express indirect effects and higher-level cellular- or organism-level processes; for example, BEL can represent results such as *the abundance of BAD protein increases apoptosis*. Each BEL Statement records a scientific finding, such as the effect of a drug or other perturbation on an experimental measurement, along with contextual annotations such as organism, disease, tissue, and cell type. BEL Statements are structured as subject, predicate, object (RDF) triples: The subject and object are BEL Terms identifying molecular entities or biological processes, and the predicate is a relationship such as *increases* or *decreases*. BEL has been used to create both public and private knowledge bases for machine reasoning; the BEL Large Corpus (see www.openbel.org) is currently the largest openly accessible BEL knowledge base and consists of about 80,000 statements curated from over 16,000 publications.

---

## Assembly of alternative executable models from mechanistic findings

The role of INDRA *Assemblers* is to generate models from a set of *Statements*. This step is governed not only by the relevant biology, but also by the requirements of the target formalism (e.g., ODE systems, rule-based models, or graphs) and decisions about model complexity (e.g., the number of variables, parameters, or agents). INDRA has multiple *Assemblers* for different model formats; here, we focus on the PySB *Assembler*, which creates rule-based models that can either be simulated stochastically or as networks of differential equations (Danos *et al*, 2007a; Faeder *et al*, 2009). Models assembled by INDRA's PySB *Assembler* can be exported into many widely used modeling formalisms such as SBML, MATLAB, BNGL, and Kappa using existing PySB functions (Lopez *et al*, 2013).

Assembling an INDRA Phosphorylation *Statement* into executable form requires a concrete interpretation of information that is almost always unspecified or ambiguous in the source text or database object. We illustrate this process using four alternative ways to describe the phosphorylation of MAPK1 by MAP2K1 (Fig 4). As a first step, the assembly of this *Statement* requires a concrete interpretation of a partially specified state of the enzyme agent: MAP2K1 sites S218 and S222 are specified as being phosphorylated but no

information is available about other sites or binding partners. In assembling rules, the PySB *Assembler* omits any unspecified context, exploiting the "don't care, don't write" convention (Box 3) so that the states of unspecified sites are treated as being irrelevant for rule activity. The default interpretation is therefore that phosphorylation of MAP2K1 at S218 and S222 is *sufficient* for kinase activity; whether or not it is also *necessary* is determined by other rules involving MAP2K1 that may be in the model.

The second step in the assembly of a Phosphorylation *Statement* is generating a concrete set of biochemical reactions that constitute an executable model. The challenge here is that the concept of protein "phosphorylation" can be realized in a model in multiple different ways. For example, a "one-step" policy converts an INDRA Phosphorylation *Statement* into a single bimolecular reaction in which a product (a phospho-protein) is produced in a single irreversible reaction without explicit consideration of enzyme–substrate complex formation. One-step reactions can be modeled using a variety of rate laws depending on modeling assumptions, including a pseudo-first-order rate law (Fig 4, "one-step policy, pseudo-first-order" comprising one reaction rule and one free parameter) in which the rate of the reaction is proportional to the product of the enzyme and substrate concentrations. Such a representation is not biophysically realistic, since it does not reproduce behaviors such as enzyme saturation, but it has the advantage of requiring only one free parameter. Alternatively, a one-step reaction can be modeled with a Michaelis–Menten rate law (Fig 4, "one-step policy, Michaelis-Menten") which generates one reaction rule and two free parameters; this policy makes a quasi-steady-state assumption about the enzyme–substrate complex (Chen *et al*, 2010). One-step mechanisms are convenient for modeling coarse-grained dynamics and causal flows in complex signaling networks (Salazar & Höfer, 2006). A "two-step policy" is more realistic and creates two rules: one for reversible enzyme–substrate binding and one for product release (Fig 4, "two-step policy"; two reaction rules and three free parameters). This is the most common interpretation of a phosphorylation reaction in existing dynamical models and correctly captures enzyme saturation, substrate depletion, and other important mass-action effects. However, the two-step policy does not explicitly consider ATP as a substrate, and cannot model the action of ATP-competitive kinase inhibitors at the enzyme active site. The "ATP-dependent" policy corrects for this and explicitly models the binding of ATP and substrate as separate reaction steps (Fig 4, "ATP-dependent policy") generating three reaction rules and five free parameters. Other mechanistic interpretations of "phosphorylation" are also possible: for example, two-step or ATP-dependent policies in which the product inhibits the enzyme by staying bound (or rebinding) after the phospho-transfer reaction (Gunawardena, 2014b). Such rebinding can have a substantial impact on kinase activity.

It might appear at first glance that the most biophysically realistic policy is preferable in all cases. However, a fundamental tradeoff always exists between model complexity and faithfulness to underlying detail: As the biochemical representation becomes more detailed, the number of free parameters and intermediate species increases, reducing the identifiability of the model (Raue *et al*, 2009). Given such a tradeoff, the benefit of having multiple *assembly policies* becomes clear: Alternative models can automatically be constructed from a single high-level biochemical assertion

 

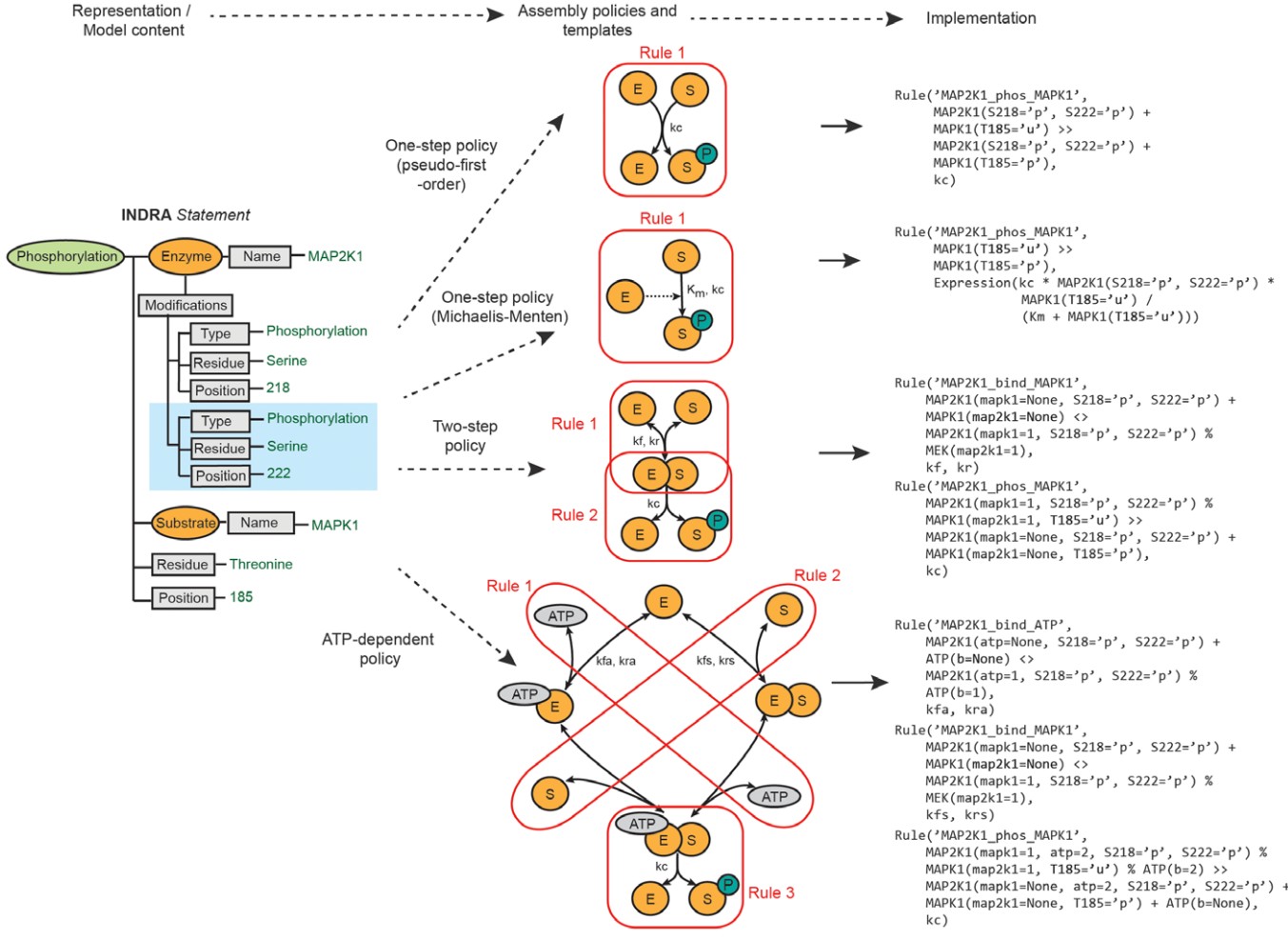

**Figure 4.  INDRA Statements are assembled into biochemical rules via assembly policies.**

The flow from representation and model content to implementation is governed by assembly policies and biochemical rule templates (top). A Phosphorylation INDRA *Statement* with enzyme (MAP2K1) and substrate (MAPK1) can be assembled using several policies including one-step policy with pseudo-first-order rate law (center, top), one-step policy with Michaelis–Menten rate law (center, second from top), two-step policy (center, second from bottom), and ATP-dependent policy (center, bottom). Each policy corresponds to a template for a generic enzyme (E) and a substrate (S). The one-step policies assume that the enzyme catalyzes the phosphorylation of the substrate in a single step such that the transient enzyme–substrate complex is not modeled. This is represented as a single rule irrespective of the associated rate laws (Rule 1; red boxes and PySB rules). The two-step policy assumes the reversible formation of an enzyme–substrate complex and an irreversible catalysis and product release step corresponding to two overlapping rules (Rules 1–2; red boxes). The ATP-dependent policy assumes a template in which the enzyme has to bind both the substrate and ATP but can bind them in an arbitrary order. This corresponds to two rules: one for ATP binding and one for substrate binding. A third rule describes the release of the phosphorylated substrate from the enzyme–substrate complex (Rules 1–3; red boxes).

depending on their suitability for a particular modeling task. The transparency and repeatability of model generation using assembly policies is especially important for larger networks in which hundreds or thousands of distinct species are subject to adjustment as the biophysical interpretation changes. Assembly policies can be applied globally to the model or to specific *Statement* types (e.g., a one-step policy for IncreaseAmount *Statements* vs. a two-step policy for Phosphorylation *Statements*). In the current implementation of INDRA, policies cannot be applied to individual *Statements*; this extension is feasible but would require that the user maintains consistency among *Statements* involving the same reactants.

To enable simulation of reaction networks as ODEs in the absence of data on specific rate parameters, INDRA uses a set of biophysically plausible default parameters; for example, association rates are diffusion limited ($10^6$ M$^{-1}$ s$^{-1}$), off-rates default to $10^{-1}$ s$^{-1}$ (yielding a default $K_D$ of 100 nM) and catalytic rates default to 100 s$^{-1}$. These parameter values can be adjusted manually or obtained by parameter estimation. An extensive literature and wide range of tools exist for parameter estimation using experimental data, and they are directly applicable to models assembled by INDRA (Mendes & Kell, 1998; Moles *et al*, 2003; Eydgahi *et al*, 2013; Thomas *et al*, 2015). For simplicity, we do not discuss this important topic further and rely below either on INDRA default parameters or on manually adjusted parameters (as listed in the Appendix) to facilitate dynamical simulations.

## Box 3:   Rule-based modeling and PySB

Accurate simulation of biochemical systems requires that every species be explicitly tracked through time. The combinatorial nature of protein complex assembly, post-translational modification, and related processes causes the number of possible molecular states in many signaling networks to explode and exceed the capacity for efficient simulation (Stefan *et al*, 2014). For example, full enumeration of complexes involved in EGF signaling would require more than $10^{19}$ molecular species differing in their states of oligomerization, phosphorylation, and adapter protein binding (Feret *et al*, 2009). Rule-based modeling (RBM) languages such as Kappa and BioNetGen (BNGL) address this challenge by allowing interactions among macromolecules to be defined using "rules" specifying the local context required for a molecular event to occur (Danos *et al*, 2007a; Faeder *et al*, 2009). The molecular features that do not affect the event are omitted from the rule, a convention known as "don't care, don't write". Specifying molecular interactions as rules has two chief benefits: (i) It makes the representation of a model much more compact and transparent than a set of equations and (ii) it enables the simulation of very complex systems using network-free methods (Danos *et al*, 2007b). RBMs can also be translated into conventional modeling formalisms such as networks of ODEs.

Executable model assembly in INDRA is built on PySB, a software system that embeds a rule-based modeling language within Python, thereby enabling the use of macros and modules to concisely express recurring patterns such as catalysis, complex assembly, sub-pathways (Lopez *et al*, 2013). Rule-based modeling languages are well suited to building executable models from high-level information sources such as natural language because assertions about mechanisms typically specify little molecular context. INDRA converts such assertions into one or more model rules using *policies* that control the level of detail.

## Modeling alternative dynamical patterns of p53 activation

As an initial test of using INDRA to convert a word model and accompanying schematic into an executable model, we turned to a widely cited review in *Cell* that describes the canonical reaction patterns controlling the responsiveness of mammalian signal transduction systems to stimulus (Purvis & Lahav, 2013). Figure 5 of Purvis and Lahav (2013) depicts the dynamics of p53 response to single-stranded and double-stranded DNA breaks (SSBs and DSBs). Using a schematic illustration, Purvis and Lahav explain that pulsatile p53 dynamics arises in response to DSBs but sustained dynamics are induced by SSBs. The difference is attributed to negative feedback from the Wip1 phosphatase to the DNA damage-sensing kinase ATM, but not to the related kinase ATR. We wrote a set of simple declarative phrases (Fig 5B and C) corresponding to edges in the schematic diagram (Fig 5A) that represent activating or inhibitory interactions between Mdm2 (an E3 ubiquitin-protein ligase), p53, Wip1, and ATM (or ATR; yellow numbers in Fig 5A–C). We then used INDRA to read the text (the "word models") and assemble executable models in PySB. These models were instantiated as networks of ODEs and simulated numerically. For each model, we plotted p53 activation over time using standard Python libraries (Oliphant, 2007).

We found that our initial word models (comprising sentences 1–5 in Fig 5B and sentences 1–6 in 5C) failed to reproduce the p53 dynamics expected for SSBs and DSBs: In our INDRA models, SSBs induced steady, low-level activation of p53 and DSBs failed to induce oscillation (Appendix Fig S6). One feature not explicitly included in the Purvis and Lahav diagrams and hence missing from

our initial word models is negative regulation of Mdm2 and Wip1. Visual representations of signaling pathways frequently omit such inhibitory mechanisms despite their impact on dynamics (Heinrich *et al*, 2002) (Purvis and Lahav were aware of these inhibitory reactions since they are found in ODE-based models of p53 dynamics from the same research group (Batchelor *et al*, 2011); because the diagram's purpose was to illustrate the specific role of negative feedback, these reactions were likely omitted for clarity). The mechanisms that inactivate Mdm2 involve binding by the catalytic inhibitor p14ARF (Agrawal *et al*, 2006) and those for Wip1 involve HIPK2-mediated phosphorylation and subsequent ubiquitin-dependent degradation (Choi *et al*, 2013) (depicted by dotted arrows and pink numbers in Fig 5A). We added these reactions to the model as simple natural language phrases (denoted by pink numbers in Fig 5B and C).

When the updated word models were assembled using INDRA and simulated as ODEs, p53 exhibited sustained activation in response to SSBs but did not oscillate in response to DSBs (Appendix Fig S6). We then realized that the DSB response model lacked a fundamental property of an oscillatory system, namely a time delay (Novák & Tyson, 2008). This delay had previously been modeled by Lahav and colleagues (Batchelor *et al*, 2011) by using delay differential equations but time delays can also be generated by positive feedback (Novák & Tyson, 2008). Both ATM and ATR are known to undergo activating auto-phosphorylation (Bakkenist & Kastan, 2003; Liu *et al*, 2011). We therefore added phrases describing auto-activation of ATM or ATR to the word models (denoted by dotted arrow and green numbers in Fig 5A, corresponding to green numbers in 5B and C). When assembled by INDRA, the extended word models successfully generated p53 oscillation in response to DSBs (Fig 5C). The presence of oscillations was robust to changes in kinetic parameters and initial conditions (Appendix Table S3 and Appendix Fig S6). Moreover, in the expanded model ATR-dependent p53 activation by SSBs still resulted in sustained p53 activation (Fig 5B, Appendix Table S2 and Appendix Fig S6). The key point in this exercise is that features essential for the operation of a dynamical system (e.g., degradation and auto-activation) were omitted from an informal diagram focusing on feedback for reasons of brevity and clarity, but this had the unintended consequence of decoupling the text from the pathway schematic and the schematic from the dynamics being described. By converting word models directly into executable computational models, we ensure that verbal descriptions and dynamical simulations are congruent.

The p53 model offers an opportunity to test how robust INDRA (and the TRIPS reading system) are to changes in the way input text is phrased. When we tested eight alternatives for the phrase "*Wip1 inactivates ATM*" ranging from "*Wip1 has been shown to deactivate ATM*" to "*ATM is inactivated by Wip1*" (Fig 5D, right, green sidebar) and found that all eight generated the same INDRA *Statement* and thus the same model as the original sentence. However, NLP is sensitive to spelling errors such as "*deaactivates*" [*sic*] and to grammatical errors such as "*Wip1 inactivate ATM*". In addition, some valid linguistic variants are not recognized, representing a limitation of extraction into INDRA *Statements* (Fig 5D, right, red sidebar). We also tested whether differences in the way biological entities are named affects recognition and grounding; we found that *Wip1, WIP-1, WIP1, PPM1D,* and *Protein phosphatase 1D* as well as *ATM, Atm,*

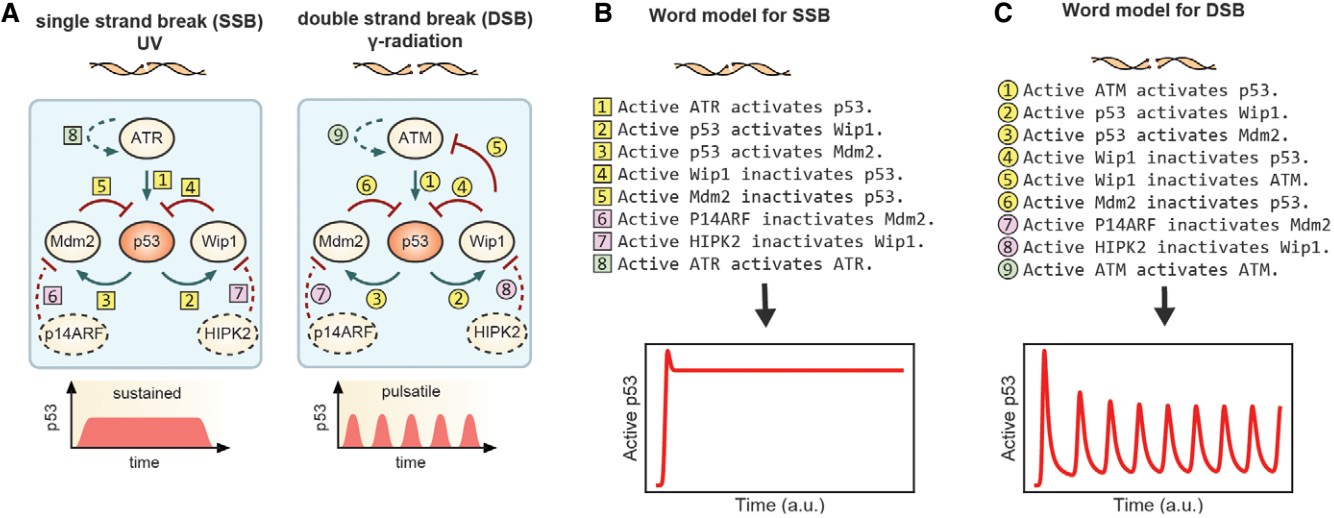

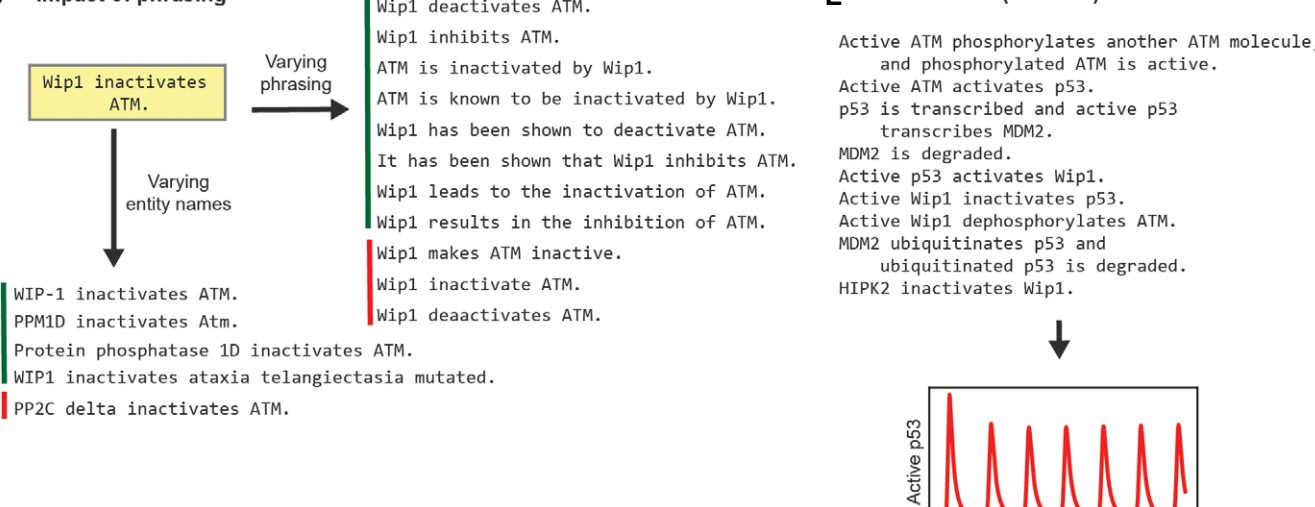

**Figure 5.  Modeling patterns of p53 activation dynamics from natural language.**

A   Patterns of p53 activation dynamics upon double-strand break DNA damage (left) and single-strand break DNA damage (right), adapted from Purvis and Lahav (2013). Edges with yellow numbers correspond to the original diagram in Purvis and Lahav (2013); pink and green numbers correspond to mechanisms added subsequently, as described in the text.

B   Natural language descriptions of the mechanisms involved in single-strand break DNA damage (SSB) response corresponding to the diagram on the left-hand side of (A) and dynamical simulation of p53 activity from the corresponding INDRA-assembled model (below).

C   Natural language descriptions of the mechanisms involved in double-strand break DNA damage (DSB) response corresponding to the diagram on the right-hand side of (A) and dynamical simulation of p53 activity from the corresponding INDRA-assembled model (below).

D   For the base sentence "Wip1 inactivates ATM", variants in the names of entities are shown below with four examples that produce the intended result (green sidebar) and one example that does not (red sidebar). To the right, eleven linguistic variants of the sentence are shown with eight producing the intended result (green sidebar) and three that do not, including one with a grammatical error and one with a spelling error (red sidebar).

E   The POMI1.0 model is a mechanistically more detailed variant of the double-strand break response model (which is shown in the right-hand side diagram of A, with its natural language description shown in B). The model assembled with INDRA produces oscillations in p53 activity over time when simulated (bottom).

and *ataxia telangiectasia mutated* all worked as expected (Fig 5D, bottom, green). However, the recognition of protein and gene names in text is challenging; for instance, "PP2C delta" was not recognized as a synonym for Wip1 (Fig 5D, bottom, red), though the more common variant "PP2Cδ" is.

We then used INDRA to assemble a more detailed and mechanistically realistic model of p53 activation following DSBs (Fig 5E; POMI1.0). While the model in Fig 5C contained only generic activating and inhibitory reactions, the goal of POMI1.0 was to test INDRA concepts such as phosphorylation, transcription, ubiquitination, and

In normal cells, signal transduction via MAPK is initiated when an extracellular growth factor such as EGF induces dimerization of receptor tyrosine kinases (the EGFR RTK, for example) on the cell surface. Dimerization and subsequent activation of RTKs results in assembly of signaling complexes at the plasma membrane and conversion of RAS-family proteins (HRAS, KRAS, and NRAS) to an active, GTP-bound state. RAS-GTP activates members of the RAF family of serine/threonine kinases (ARAF, BRAF, and RAF1), which serve as the first tier in a three-tier MAP kinase signaling cascade: RAF proteins phosphorylate MAP2K/MEK family proteins, which in turn phosphorylate the MAPK/ERK family proteins that control transcription factor activity, cell motility, and other aspects of cell physiology. MAPK signaling is subject to regulation by feedback mechanisms that include inhibitory phosphorylation of EGFR and SOS by ERK, inhibition of the GRB2-mediated scaffold by the SPRY family of proteins, and inhibition of ERK by DUSP proteins (Lito *et al*, 2012).

MAPK/ERK signaling is a key regulator of cell proliferation and is mutated in a variety of human cancers (Dhillon *et al*, 2007), with dramatic effects on cellular homeostasis. Overall, ~20% of all cancers carry driver mutations in one of the genes that encode MAPK pathway proteins (Stephen *et al*, 2014) and in the case of melanoma, 50% of cancers carry activating point mutations in BRAF (most commonly BRAF V600E). ATP-competitive inhibitors such as vemurafenib provide significant clinical benefit in treating BRAF-mutant melanoma. However, remission of disease is transient, as tumors and tumor-derived cell lines develop resistance to vemurafenib over time (Lito *et al*, 2012). Recent studies have identified feedback regulation, bypass mechanisms, and other context-dependent factors responsible for restoring ERK signaling to pre-treatment levels (Lito *et al*, 2012, 2013; Shi *et al*, 2012). For example, in the BRAF-V600E cell line A375, vemurafenib has been shown to suppress EGF-induced ERK phosphorylation completely upon treatment (Lito *et al*, 2013) but ERK phosphorylation levels rebound within 48 h, with a concurrent increase in the level of RAS-GTP, the active form of RAS (Lito *et al*, 2012). It is the biology of this adaptation that we aim to capture in an INDRA model.

underlying mechanism using natural language, we observed that including the mechanism "*Active ATM phosphorylates another ATM molecule*" was essential for oscillation; the phrase "*ATM phosphorylates itself*" generated a valid set of reactions but did not create oscillations for any of the parameter values we sampled. The difference is that "*Active ATM phosphorylates another ATM molecule*" corresponds to a trans-phosphorylation reaction (other phrasings also work, such as "*Active ATM trans-phosphorylates itself*")—one molecule of ATM phosphorylates another molecule of ATM. In contrast, "*ATM phosphorylates itself*" implies modification in *cis*, which is incapable of generating oscillations in the p53 network. ATM trans-phosphorylation represents a form of positive feedback since the flux through the phosphorylation reaction increases with the concentration of the reaction product, namely phosphorylated ATM. As described in detail by Novák and Tyson, positive feedback in such reaction mechanisms can create the "dynamical hysteresis" necessary for a time delay (Novák & Tyson, 2008). It is well known that ATM and ATR auto-phosphorylations occur in *trans* (Bakkenist & Kastan, 2003; Liu *et al*, 2011), validating this aspect of the model. This result highlights a danger in the use of word models alone: Differences in mechanism that profoundly impact network dynamics can be obscured by ambiguous and imprecise natural language. Such ambiguities are propagated by INDRA and can be identified by the user at multiple (intermediate) stages of the extraction and assembly process (see Appendix iPython Notebook 1). The phrase "*Active ATM phosphorylates another ATM molecule*" is not particularly elegant English, but it is unambiguous; understanding that "*ATM phosphorylates itself*" is insufficient for p53 oscillation highlights the essential difference between *trans* and *cis* phosphorylation.

The foregoing analysis of the Lahav and Purvis review illustrates several beneficial features of direct text-to-model conversion: (i) the possibility of identifying subtle gaps and deficiencies in word models with the potential to profoundly affect network dynamics and function; (ii) the ability to maintain precise congruence between verbal, pictorial, and computational representations of a network; and (iii) a reminder to include neglected negative regulatory mechanisms when explaining network dynamics. We propose that future figures of this type include accompanying declarative text (precisely stated word models) on the basis of which graphs and dynamical models can be created. We have found that it is remarkably informative to experiment with language and then render it in

degradation. We also used modifiers to describe the molecular state required for a protein to participate in a particular reaction (e.g., "*ubiquitinated p53 is degraded*"). The set of ten phrases shown in Fig 5E were assembled into 11 rules, 12 ODEs, and 18 parameters (Appendix Table S4). When we simulated the resulting ODE model, we observed the expected oscillation in p53 activity (Fig 5E and Appendix Fig S6). By adding and removing different aspects of the

**Figure 6.    INDRA-built models of vemurafenib resistance in response to growth factor signals.**

A    Simplified schematic representation of the observed ERK phosphorylation phenomena in BRAF-V600E mutants that are hypothesized to be the basis of adaptive resistance. In untreated BRAF-V600E cells (left), mutant BRAF is constitutively active independently of RAS and leads to higher ERK phosphorylation levels (thick green edge) and stronger negative feedback to SOS (thick red edge). Upon vemurafenib treatment, in the short term (center), ERK phosphorylation is decreased due to BRAF V600E inhibition (thin green edge). Over time, resistance develops (right); the ERK-SOS feedback loop becomes weaker (thin red edge) and increased RAS activity induces RAF dimerization, leading to a rebound in ERK phosphorylation (thick green edge).

B    MEMI1.0 is described in 14 sentences which are assembled into 28 PySB rules and 99 ordinary differential equations. Simulation of phosphorylated ERK (blue) and active RAS (green) is shown relative to their respective values at time 0, when vemurafenib is added. The model simulation shows that upon vemurafenib addition, the amount of phosphorylated ERK is quickly reduced and stays at a low level, while the amount of active RAS is unchanged.

C    In MEMI1.1, by extending three existing sentences (4, 5, 14) and adding two new ones (15, 16) (changes shown in orange), the ERK-SOS negative feedback is modeled and assembled into 34 rules and 275 ODEs. The model simulation (right) reproduces RAS reactivation (green) upon vemurafenib treatment; however, the experimentally observed rise in ERK phosphorylation (blue) is not reproduced.

D    MEMI1.2 extends MEMI1.1 by adding a sentence (17) and replacing an existing sentence with two new sentences (8A and 8B) (changes shown in green). INDRA produces a model consisting of 37 rules and 353 ODEs. Model simulations are able to reproduce the expected rise in RAS activation (green) and the increased phosphorylation of ERK (blue).

**A**

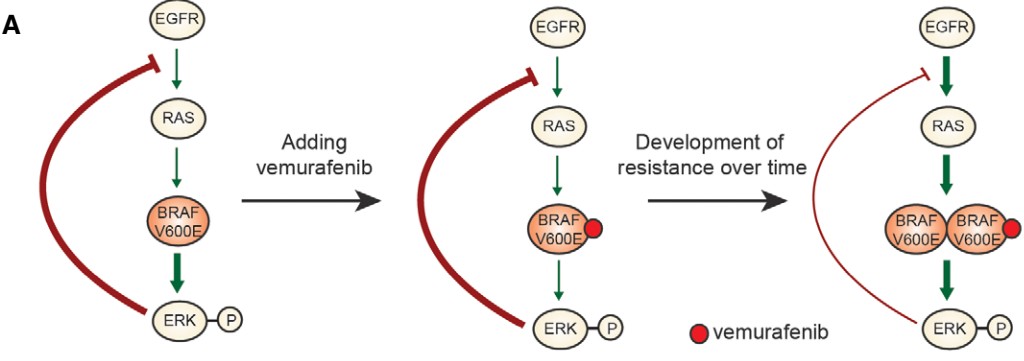

**B**    **MEMI 1.0**

1. The growth factor ligand EGF binds EGFR.
2. The EGFR-EGF complex binds another EGFR-EGF complex.
3. The EGFR-EGFR complex binds GRB2.
4. EGFR-bound GRB2 binds SOS.
5. GRB2-bound SOS binds RAS that is not bound to BRAF V600E.
6. SOS-bound RAS binds GTP.
7. GTP-bound RAS that is not bound to SOS binds BRAF V600E.
8. Vemurafenib binds BRAF V600E.
9. BRAF V600E that is not bound to vemurafenib phosphorylates MEK.
10. PP2A-alpha dephosphorylates MEK that is not bound to ERK.
11. Phosphorylated MEK is activated.
12. Active MEK that is not bound to PP2A-alpha phosphorylates ERK.
13. Phosphorylated ERK is activated.
14. DUSP6 dephosphorylates ERK.

28 rules
41 parameters
99 ODEs

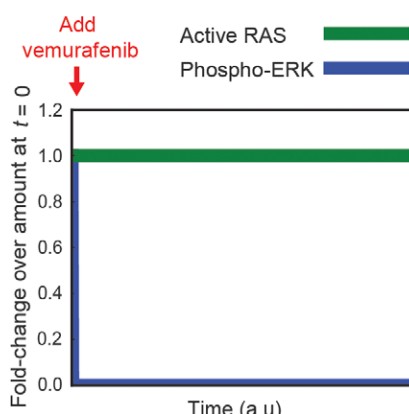

**C**    **MEMI 1.1 = MEMI 1.0 + ERK-SOS feedback**

4. EGFR-bound GRB2 binds SOS that is not phosphorylated on a serine.
5. GRB2-bound SOS that is not phosphorylated on serine binds RAS that is not bound to BRAF V600E.
14. DUSP6 dephosphorylates ERK that is not bound to SOS.
15. SOS not bound to RAS is phosphorylated on Serine by active ERK not bound to DUSP6.
16. A phosphatase dephosphorylates SOS on serine.

34 rules
50 parameters
275 ODEs

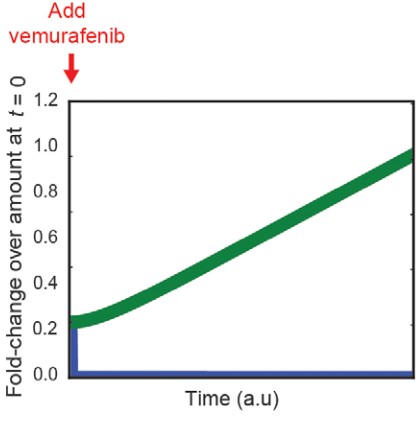

**D**    **MEMI 1.2 = MEMI 1.1 + BRAF dimerization**

17. RAS-bound BRAF V600E binds RAS-bound BRAF V600E.
8A. Vemurafenib binds BRAF V600E that is not bound to BRAF V600E.
8B. Vemurafenib binds BRAF V600E that is bound to BRAF V600E.

37 rules
54 parameters
353 ODEs

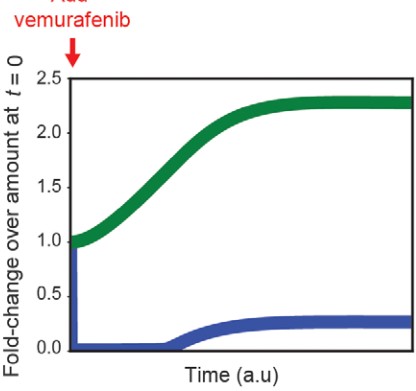

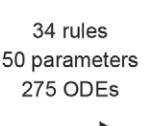

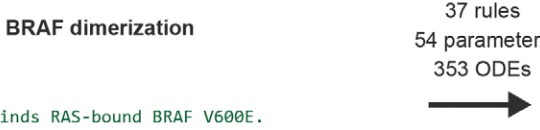

**Figure 6.**

    

computational form: It was this type of experimentation that led us to rediscover for ourselves the importance of negative regulation and nonlinear positive feedback in generating p53 oscillations.

## Modeling resistance to targeted therapy by vemurafenib

The MAPK/ERK signaling pathway is a key regulator of cell proliferation, differentiation, and motility and is frequently dysregulated in human cancer (Box 4). Multiple ATP-competitive and non-competitive (allosteric) inhibitors have been developed targeting kinases in this pathway. The most clinically significant drugs bind RAF and MEK kinases in BRAF-mutant melanomas. For patients whose tumors express an oncogenic BRAF$^{V600E/K}$ mutation, treatment with the BRAF inhibitor vemurafenib (or, in more recent practice, a combination of the BRAF inhibitor dabrafenib and MEK inhibitor trametinib) results in dramatic tumor regression. Unfortunately, this is often followed by drug resistance and disease recurrence 6–18 months later (Larkin *et al*, 2014). The mechanisms of drug resistance are under intensive study and include an adaptive response whereby MAPK signaling is reactivated in tumor cells despite continuous exposure to BRAF inhibitors (Lito *et al*, 2012, 2013; Shi *et al*, 2012). Reactivation of MAPK signaling in drug-treated BRAF$^{V600E/K}$ cells is thought to involve disruption of ERK-mediated negative feedback (Fig 6A). The biochemistry of this process has been investigated in some detail and is subtle. For example, differential affinity of BRAF kinase inhibitors to monomeric and dimeric forms of BRAF is partly responsible for the ERK rebound (Kholodenko, 2015; Yao *et al*, 2015). Many of these processes have not been subjected to detailed kinetic modeling within the scope of the MAPK signaling pathway, and several mechanistically distinct hypotheses have been advanced to describe the same drug adaptation phenomenon. Adaptation to BRAF inhibitors therefore represents a potentially valuable application of dynamical modeling to a rapidly moving field of cancer biology (Kholodenko, 2015).

We sought to use natural language to rapidly create models of MAPK signaling in melanoma cells using mechanisms drawn from the literature, with a particular focus on a series of influential papers from the Rosen laboratory (Joseph *et al*, 2010; Lito *et al*, 2012; Poulikakos *et al*, 2010; Yao *et al*, 2015). We also sought to establish whether different biochemical hypotheses could be easily tested by modifying models at the level of natural language.

The baseline MAPK model (Melanoma ERK Model in INDRA; MEMI1.0) consists of 14 sentences describing canonical reactions involved in ERK activation by growth factors (Fig 6B, MEMI1.0) and corresponds in scope to previously described models of MAPK signaling (Birtwistle *et al*, 2007; Stites *et al*, 2007). In the baseline model, BRAF$^{V600E}$ constitutively phosphorylates MEK as long as it is not bound to vemurafenib (sentence 9: "*BRAF V600E that is not bound to Vemurafenib phosphorylates MEK*"). A two-step policy involving reversible substrate binding was used to assemble all phosphorylation and dephosphorylation reactions. For simplicity, we did not specify residue numbers or capture multi-site phosphorylation, instead modeling each step in the MAPK cascades as a single, activating phosphorylation event. With these assumptions, 14 sentences were processed by TRIPS to yield 14 INDRA Statements that were assembled into 28 PySB rules and 99 differential equations; the network of coupled ODEs was then simulated. 65 of the 99 species in the model involve complexes assembling around

EGFR, which are generated by the biochemical reactions described in the sentences that constitute the word model.

A key property of vemurafenib-treated BRAF$^{V600E}$ cells as described by Lito *et al* is that the drug initially reduces pERK below its steady-state level but pERK then rebounds despite the continued presence of vemurafenib. Levels of RAS-GTP (the active form of RAS) also increase during the rebound phase (Lito *et al*, 2012). In MEMI1.0, addition of EGF causes activation of RAS and phosphorylation of ERK at steady state. Addition of vemurafenib rapidly reduces pERK levels (Fig 6B) but extended simulations under a range of EGF and vemurafenib concentrations show that the amount of active RAS depends only on the amount of EGF and is insensitive to the amount of vemurafenib; moreover, no rebound in pERK is observed in the presence of vemurafenib (Fig 6B and Appendix Fig S7A). Thus, MEMI1.0 fails to capture drug adaptation.

In a series of siRNA-mediated knockdown experiments, Lito *et al* showed that pERK rebound involves an ERK-mediated negative feedback on one or more upstream pathway regulators such as Sprouty proteins (SPRY), SOS, or EGFR. To identify a specific mechanism that might be involved, we used the BioPAX and BEL search capabilities built into INDRA. We queried Pathway Commons (Cerami *et al*, 2011) for BioPAX reaction paths leading from ERK (MAPK1 or MAPK3) to SOS (SOS1 or SOS2) and obtained multiple INDRA *Statements* for a MAPK1 phosphorylation reaction that had one or more residues on SOS1 as a substrate (including SOS1 sites S1132, S1167, S1178, S1193, and S1197). However, Pathway Commons did not provide any information on the effects of these phosphorylation events on SOS activity. To search for this, we used INDRA's BEL *Interface* to query the BEL Large Corpus (Catlett *et al*, 2013; Box 2) for all curated mechanisms directly involving SOS1 and SOS2. We found evidence that ERK phosphorylates SOS and that ERK inactivates SOS (Corbalan-Garcia *et al*, 1996). We did not find a precise statement in either database stating that phosphorylation of SOS inactivates it, but the publication referred to in the BEL Large Corpus as evidence of this interaction (Corbalan-Garcia *et al*, 1996) describes a mechanism whereby SOS phosphorylation interferes with its association with the upstream adaptor protein GRB2. To include the inhibitory phosphorylation of SOS by ERK, we therefore modified three sentences (Fig 6C, Model 2, Sentences 4, 5, and 14) in Model 1 and added two new sentences (Fig 6C, Model 2, sentences 15 and 16). Thus, although INDRA can assemble *Statements* derived from databases directly into models, in this case human curation (via changes to the natural language text) was required to identify gaps in the mechanisms available from existing sources.

The inclusion of SOS-mediated feedback produced 16 declarative sentences that were translated into a MEMI1.1 model having 34 rules and 275 ODEs. Assembly of MEMI1.1 involved imposing assumptions to limit combinatorial complexity. For instance, in sentence 15 (Fig 6C), we specified that ERK cannot be bound to DUSP6 for ERK to phosphorylate SOS. While it is not known whether or not ERK can bind both DUSP6 and SOS at the same time, allowing for this possibility introduces a "combinatorial explosion" (Faeder *et al*, 2005; Feret *et al*, 2009) in the number of reactions and makes mass-action simulation difficult. It is common to make simplifying assumptions of this type in dynamical models (see, for instance; Chen *et al*, 2009), and an advantage of using natural language is that the assumptions are clearly stated. When MEMI1.1 was simulated, we observed that, given a sufficient level of basal

activity by addition of EGF, addition of vemurafenib resulted in dose-dependent increases in active RAS over pre-treatment levels (Appendix Fig S7B). However, pERK levels remained low, suggesting that negative feedback alone (at least as modeled in MEMI1.1) is insufficient to explain the rebound phenomenon observed by Lito *et al* (Fig 6C, Appendix Fig S7B).

It has been suggested that RAF dimerization plays an important role in cellular responsiveness to RAF inhibitors (Lavoie *et al*, 2013; Yao *et al*, 2015). Both wild-type and BRAF$^{V600E}$ dimers have a lower affinity for vemurafenib as compared to their monomeric forms (Yao *et al*, 2015). Moreover, Lito *et al* observed that the reactivation of ERK following vemurafenib treatment was coincident with formation of RAF dimers, leading to the suggestion that vemurafenib-insensitive dimers in cells play a role in the reactivation of ERK signaling (Kholodenko, 2015). To model this possibility, we created MEMI1.2 in which binding of vemurafenib to monomeric or dimeric BRAF is explicitly specified by separate sentences, allowing the effects of different binding affinities to be explored (Fig 6D). Assembly of this model yielded 353 ODEs, many of which were required to represent the combinatorial complexity of BRAF dimerization and vemurafenib binding (Appendix Fig S8). Simulation showed that RAS activation increases and settles at a higher level following vemurafenib treatment, with the magnitude of the increase dependent on the amount of EGF and the concentration of drug (Fig 6D, Appendix Fig S7C). Following a period of pERK suppression, rebound in pERK levels to ~30% of their maximum is observed (Fig 6D) effectively recapturing the key findings of Lito *et al* subsequent work has shown that resistance to vemurafenib can also involve proteins such as DUSP, SPRY2 (Lito *et al*, 2013), and CRAF (Montagut *et al*, 2008). These mechanisms do not feature in the models described here, but could be included in MEMI by adding a few phrases to the word model.

This example demonstrates that it is possible to use INDRA to model signaling systems of practical interest at a scope and level of detail at which interesting biological hypotheses can be explored and tested. Comparison of models MEMI1.0 to 1.2 suggests that both feedback and BRAF dimerization are necessary for vemurafenib adaption and pERK rebound, in line with experimental evidence. The number of free parameters in these models varies, and we have not performed formal model calibration or verification, so the conclusion that MEM1.2 is superior to 1.0 is not rigorously proven. However, INDRA-assembled rule sets represent a solid starting point for such downstream analysis.

One issue we encountered in assembling these models was controlling complexity arising from the formation of multiple protein complexes from a single set of reactants. This is a known challenge in dynamical modeling of biochemical networks with poorly understood implications for cellular biochemistry (Faeder *et al*, 2005; Harmer *et al*, 2010; Sneddon *et al*, 2011). From the perspective of an INDRA user, this is likely to manifest itself as a property that can be diagnosed at the level of PySB rules or ODE networks, which can be inspected interactively (see Appendix Notebook 2). Unwanted combinatorial complexity can be resolved in two ways: (i) by using natural language to make additional assumptions about molecular context and (ii) by choosing assembly policies minimizing combinatorial complexity by reducing complex formation (e.g., Michaelis–Menten instead of two-step policy). Both strategies are illustrated in Appendix Notebook 2.

**An extensible and executable map of the RAS signaling pathway**

The BRAF pathway described above is part of a larger immediate-early signal transduction network with multiple receptors as inputs and transcription, cell motility, and cell fate determination as outputs. RAS is a central node in this network and is an important oncogenic driver (Stephen *et al*, 2014). The ubiquity of RAS mutations in cancer has led to renewed efforts to target oncogenic RAS and RAS effectors. As a resource for the community of RAS researchers, the NCI RAS Initiative has created a curated pathway diagram that defines the scope of the RAS pathway as commonly understood by a community of experts (Stephen *et al*, 2014). Such pathway diagrams can serve as useful summaries, but unless they are backed by an underlying computable knowledge representation, they are of limited use in quantitative data analysis.

We used INDRA to describe the RAS signaling network and automatically generated a diagram (Fig 7A, right) corresponding to the community-curated Ras Pathway v1.0 diagram (available at http://www.cancer.gov/research/key-initiatives/ras/ras-central/blog/what-do-we-mean-ras-pathway). We described the interactions in natural language (Fig 7A left, full text shown in Appendix Section 2.4) and used TRIPS to convert the description into INDRA *Statements*. A node-edge graph was generated using INDRA's *Graph Assembler* and rendered using Graphviz (Fig 7A, right). Although different stylistically, the pathway map assembled using INDRA matches the original one drawn by hand in the following ways: It (i) includes the same set of proteins, (ii) represents the same set of interactions among these proteins, and (iii) recapitulates the semantics and level of mechanistic detail of the original diagram in that interactions are represented as directed positive and negative edges or undirected edges indicating complex formation. The pathway map is also

**Figure 7.  An INDRA-assembled extensible and executable pathway map of RAS signaling.**

A   Positive and negative activations as well as complex formation between proteins are written in natural language (left) to describe simplified interactions in the RAS pathway (for full text, see Appendix Section 2.5). The INDRA-assembled graph is shown on the right showing activations (black), inhibitions (red), and binding (blue).

B   A correction on the pathway map is made by editing the original text. One sentence is removed (red sentence) and is replaced by another one (green sentence) as a basis for the updated assembly whose relevant parts are shown as a graph below. P90RSK is removed as a substrate of mTORC2 and added as a substrate of MAPK1 and MAPK3 (green highlight).

C   The pathway map is extended with a new branch by adding four additional sentences describing JNK signaling. The newly added pathway (green highlight; gene names appearing as their standard gene symbols, for instance, "HPK1" in the original sentences is represented as the node MAP4K1) provides a parallel path from EGFR to the JUN transcription factor, both of which were included in the original model.

D   Simulation results of Boolean models assembled from natural language under different inhibitor conditions. The "Basic model" contains the links shown in (A); the "Extended model" contains the extensions shown in (C). Each trace represents the activity of JUN in the presence of growth factors averaged over 100 stochastic simulations (see Materials and Methods).

**A**

```
Growth-factor proteins activate EGFR, ERBB2 and FGFR.
...
SOS and RASGRF activate HRAS, NRAS and KRAS.
RASGRP activates HRAS, KRAS and NRAS.
SPRY deactivates HRAS, KRAS and NRAS.
The RASA-ARHGAP35 complex deactivates HRAS, NRAS and KRAS.
...
HRAS, NRAS and KRAS activate ARAF, BRAF and RAF1.
ARAF, BRAF and RAF1 activate MAP2K1 and MAP2K2.
MAP2K1 and MAP2K2 activate MAPK1 and MAPK3.
MAPK1 and MAPK3 activate ETS, JUN and FOS.
KSR binds ARAF, BRAF and RAF1.
KSR binds MAP2K1 and MAP2K2.
KSR binds MAPK1 and MAPK3.
ETS, FOS and JUN activate MDM2, CCND1 and DUSP.
MDM2 deactivates TP53.
CCND1 activates CDK4 and CDK6.
CDK4 and CDK6 deactivate pRB.
DUSP deactivates MAPK1 and MAPK3.
SOS and RASGRF activate RHOA and RHOB.
AKT deactivates TSC1 and TSC2.
TSC1 and TSC2 deactivate RHEB.
RHEB activates mTORC2.
STK11 activates AMPK.
AMPK deactivates mTORC2.
mTORC2 deactivates EIF4EBP1.
mTORC2 activates P90RSK.
TIAM activates RAC and RAC activates PAK.
```

TRIPS NLP
INDRA
Graph Assembler

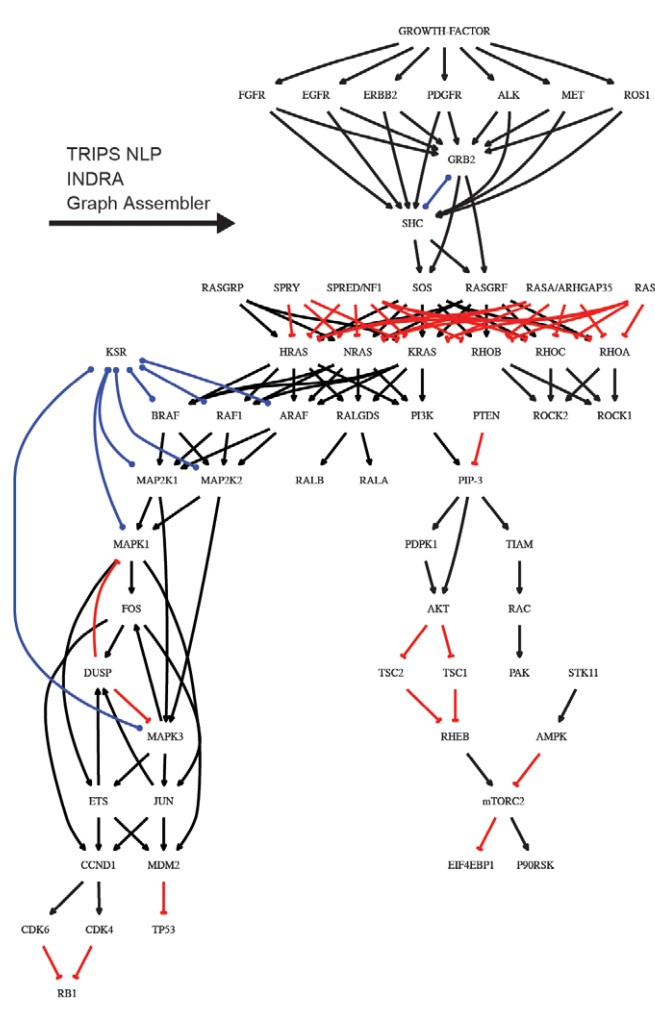

**B**

- mTORC2 activates P90RSK.
+ MAPK1 and MAPK3 activate P90RSK.

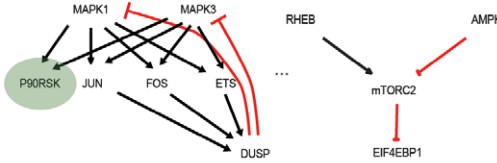

**C**
+ GRB2 activates HPK1, and HPK1 activates MAP3K7.
+ MAP3K7 activates MKK4 and MKK7.
+ MKK4 and MKK7 activate JNK1 and JNK2.
+ JNK1 and JNK2 activate JUN.

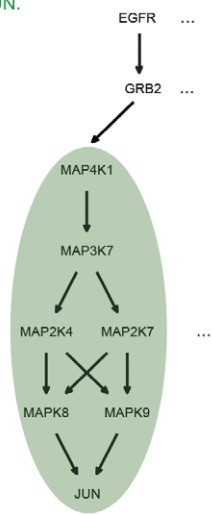

**D**

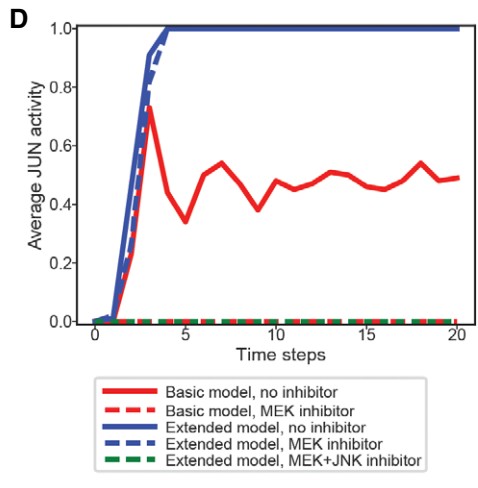

Basic model, no inhibitor
Basic model, MEK inhibitor
Extended model, no inhibitor
Extended model, MEK inhibitor
Extended model, MEK+JNK inhibitor

**Figure 7.**

visually comparable to one drawn by hand, and allows natural language-based editing and extension of the underlying set of mechanisms. For example, following distribution of v1.0 RAS diagram, the RAS Initiative solicited verbal feedback from a large number of RAS biologists both in person and via a discussion forum. Suggestions from the community consisted largely of corrections and pathway extensions. Using INDRA, these revisions of the network can be made directly, simply by editing the natural language source material. For example, one contributor noted that in the published pathway diagram (Fig 7A, right), P90RSK is activated by the mTORC2 complex, whereas in fact it is actually a substrate of MAPK1 and MAPK3 (https://www.cancer.gov/research/key-initiatives/ras/ras-central/blog/2014/what-do-we-mean-ras-pathway#comment-1693526648). To account for this correction, we modified the natural language description by replacing the sentence "*mTORC2 activates P90RSK*" with "*MAPK1 and MAPK3 activate P90RSK*". The pathway map obtained following automated assembly of the revised text correctly reflects the change suggested by the contributor (Fig 7B).

Several readers also suggested expanding the pathway map to include other relevant proteins. Extensions of this type are easy to achieve using natural language: For example, we extended the v1.0 RAS diagram to include *JNK*, a MAP kinase that is activated in many cells by cytokines and stress (Anafi *et al*, 1997; Antonyak *et al*, 1998; Wagner & Nebreda, 2009). This was achieved by adding four sentences (Fig 7C, top), including "*MAP3K7 activates MKK4 and MKK7*" and "*MKK4 and MKK7 activate JNK1 and JNK2*". The subnetwork appended to the diagram is shown in Fig 7C (bottom). Note that we used common names for the JNK pathway kinases in the word model but INDRA canonicalized these to their official gene names (e.g., "HPK1", "MKK4", and "JNK1" were converted to MAP4K1, MAP2K4, and MAPK8, respectively).

The set of mechanisms used to generate the diagrams in Fig 7A–C can also be translated into a qualitative predictive model. We used the Simple Interaction Format (*SIF*) *Assembler* in INDRA to generate a Boolean network corresponding to the natural language pathway description in Fig 7A (see Appendix Section 2.4 for the rules comprising the network). Such a Boolean network can be used to predict the effects of perturbations such as ligand or drug addition. For example, we simulated the effects of adding growth factors and MEK inhibitors on phosphorylated c-Jun. The Boolean network simulation correctly predicted that c-Jun would be phosphorylated in the presence and absence of MEK inhibitor (Fig 7D, blue). We then instantiated the extended network in Fig 7C (which identifies the JNK pathway as a possible contributor to c-Jun phosphorylation). In this case, joint inhibition of JNK and MEK was required to fully inhibit c-Jun phosphorylation (Fig 7D, green). The biology in this example is relatively straightforward but it demonstrates that natural language descriptions of mechanisms, along with automated assembly into executable forms, can be used as an efficient and transparent way of creating extensible knowledge resources for data visualization and analysis.

## Discussion

In this paper, we described a software system, INDRA, for constructing executable models of signal transduction directly from text. The process uses natural language reading software (TRIPS, in this paper) to convert text into a computer-intelligible form, identifies

biochemical mechanisms, and then casts these mechanisms in an intermediate knowledge representation that is decoupled from both input and output formats. The intermediate representation, comprising a library of INDRA *Statements*, is then used to assemble computational models of different types including networks of ODEs, Boolean networks, and interaction graphs according to user-specified policies that determine the level of biophysical detail. We have applied INDRA to three successively more ambitious use cases: (i) translating a diagram and accompanying text describing p53 regulation by DNA damage, (ii) modeling adaptive drug resistance in BRAF$^{V600E}$ melanoma cells exposed to the BRAF inhibitor vemurafenib, and (iii) constructing a large-scale model of RAS-mediated immediate-early signaling based on a crowd-sourced schematic drawing. These examples demonstrate the surprising but encouraging ability of machines to exploit the flexibility and ambiguity of natural language and then add prior knowledge about reaction mechanisms needed to create well-defined executable models.

The p53 POMI model represents a use case corresponding in scope to the mechanistic hypotheses typically presented in the literature in verbal or graphical form. We based POMI on a word model found in a review and found it necessary to add several additional mechanisms to reproduce the described oscillations in p53 (these include negative regulatory reactions and a positive feedback step involving auto-phosphorylation of ATM in *trans*). Editing and updating the model to explore alternative hypotheses was accomplished strictly at the level of the natural language description. This example highlights the potential of natural language, assembled into executable model form, to expose important and frequently overlooked differences between a formal representation of a mechanism (in this case, a network for ODEs) and a diagram that purports to describe it. Direct conversion of text into models via INDRA helps to minimize such mismatches while keeping the description in an accessible and easily editable natural language form. We propose that pathway schematics found in the conclusions of molecular biology papers include a set of declarative statements that match the schematic and any depiction of dynamics arising from simulation. Ensuring congruence among these representations will improve general understanding of cellular biology and make schematics and their underlying assumptions accessible to machines.

The BRAF$^{V600E}$ MEMI model involved a much greater number of molecular species and reactions due to the combinatorics of complex formation among BRAF, vemurafenib, MEK, and RAS. In INDRA, formation of unlikely polymers in the model assembled by INDRA was controlled by providing stricter molecular context on mechanisms in the form of natural language (e.g., "DUSP6 dephosphorylates ERK *that is not bound to SOS*"). While managing combinatorial complexity is a key challenge in building models of signaling, a benefit of using INDRA is that assumptions made regarding combinatorial complexity are made explicit either in the form of the natural language description or the policies chosen for model assembly (e.g., one-step Michaelis–Menten vs. two-step). The broader RAS pathway is the largest network tackled in this paper, but by restricting the mechanisms to positive and negative regulation and binding it remains manageable. Such a model could in principle be solicited directly from the community and we plan to release the INDRA RAS model to the same group of experts that helped Frank McCormick and colleagues build and improve the original RAS schematic (Stephen *et al*, 2014).

## Challenges in generating executable models from text and databases

Automating the construction of detailed biochemical models from text involves overcoming three technical challenges. The first is turning text into a computable form that correctly captures the biochemical events described in a sentence (typically verbs or actions) and the precise biomolecules involved (typically the subjects and objects of a phrase or sentence). This is possible in our system because TRIPS can extract meaning from sentences describing complex causal relationships in the face of variations in syntax (Box 1). TRIPS performs an initial shallow syntactic search to identify and ground named entities (genes, proteins, drugs, etc.) and then uses generic ontologies to perform "deep" semantic analysis, determining the meaning of a sentence in terms of its logical structure.

The second challenge involves extracting and normalizing information about mechanisms contained in NLP output. INDRA extracts mechanistic information from graphs generated by TRIPS by searching for matches to a predefined set of templates corresponding to biochemical processes (e.g., phosphorylation, transcription, binding, activation; Fig 3). These templates regularize the description of biochemistry in text by capturing relevant information in pre-determined fields: For example, a template for phosphorylation is structured to have a protein kinase, a phosphorylated substrate, and a target site. Information extracted by this template-matching procedure is stored in corresponding fields in *Statements,* INDRA's intermediate representation; missing fields are left blank. INDRA *Statements* currently encompass terms and reactions commonly found in signal transduction pathways and gene regulation; however, the system is being extended to include a wider variety of biochemical processes.

The third challenge in text-to-model conversion is assembling an executable model from high-level mechanistic facts acquired from input sources. Knowledge of reaction type and reactant identity is insufficient to construct a biophysical model: Additional information derived from an understanding of classes of biochemical mechanism is almost always required. For example, the conversion of a phosphorylation *Statement* into a reaction network can involve one-step kinetics, reversible two-step kinetics, or two-step kinetics with explicit ATP binding. Conversion of *Statements* into explicit models is controlled by the imposition of *assembly policies* (Fig 4). Greater biophysical realism comes at the cost of increased model complexity and reduced parameter identifiability. Thus, there is no single optimal approach to model instantiation: The level of detail is determined by the purpose of the model and the way it will be formulated mathematically.

Constructing executable models of signaling networks from pathway databases using BioPAX or BEL presents several challenges, despite the fact that this information is structured and often manually curated by experts. BioPAX reactions and BEL statements often lack the uniqueness (i.e., many variants of the same mechanism are curated) and context (i.e., participants in curated mechanisms are missing necessary molecular state) required to build coherent executable models automatically. For instance, Pathway Commons contains a multitude of representations of the reaction whereby MAPK21 phosphorylates and activates MAPK1 (Appendix Fig S10). These reactions differ in their molecular details

including which phosphorylation sites are involved and what the assumptions about the state (activity, modification, bound cofactors) of MAP2K1 are. These reactions cannot simultaneously be included in a single, coherent model as they would result in causal inconsistencies. We therefore require the user to determine which INDRA Statements extracted from a database should be included in a given model. INDRA then subjects this information to an analogous process as text, using templates and assembly policies to control the generation of specific reaction patterns. In the future, manual selection of relevant BioPAX or BEL statements could be replaced by, or supplemented with, automated tools ensuring the selection of coherent subsets of mechanisms to be included in a model.

### Separating model content and implementation

Most approaches to modeling biological networks directly couple the specification of scope and collection of relevant facts to the mathematical implementation. For example, in an ODE-based model, molecular species are directly instantiated as variables and related to each other using one or more differential equations containing terms determined by each mass-action reaction (Fig 8A "ordinary differential equations" and Fig 8B, left). Although conceptually straightforward, the lack of separation between content and implementation [an issue also discussed in Basso-Blandin *et al* (2016)] makes it difficult to update a model with new findings from the literature or new hypotheses, to change the level of biophysical detail, or to switch mathematical formalisms. Programmatic modeling overcomes some of these problems by allowing the construction of models at a higher level of abstraction in which users implement reusable and composable macros and modules (Fig 8A "PySB Macro" and Fig 8B, center; Pedersen & Plotkin, 2008; Mallavarapu *et al*, 2009; Smith *et al*, 2009; Lopez *et al*, 2013). The mathematical equations necessary for simulation are then generated automatically from the abstract representations.

Integrated Network and Dynamical Reasoning Assembler introduces a further level of abstraction whereby a user describes a set of reactions in natural language or searches for related mechanisms in pathway databases and then uses a machine to turn these facts into executable models (Fig 8A "natural language" and Fig 8B, right). In this process, a user has full control over the *content* of the model and the level of detail, as specified by policies, but model *assembly* happens automatically. Such decoupling simplifies the creation of dynamical models from natural language descriptions, enables the creation of closely related models differing in detail or mathematical formalism, and makes sure that verbal and mathematical descriptions of the same process are in correspondence (Fig 8B, right).

The decoupling of biological knowledge from specific applications reflects the way in which biologists gather mechanistic information and apply it to specific research questions. We acquire informal knowledge through years of reading and experience, but this knowledge remains highly flexible; it allows for uncertainty about particular details and can be applied to a diverse set of problems in the laboratory. The ambiguity inherent in verbal descriptions of mechanisms conforms closely to the way in which individual experiments are designed and interpreted: It is

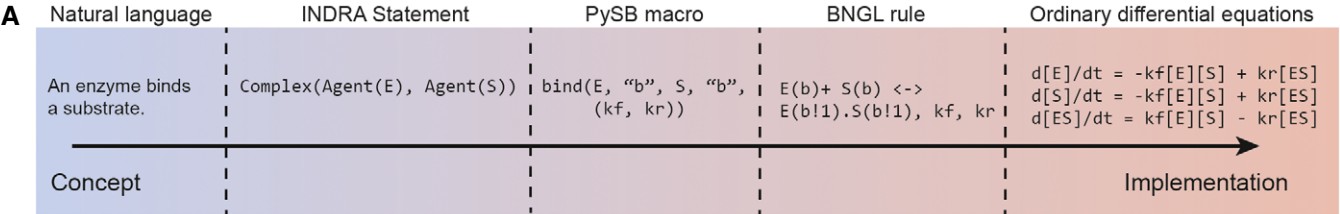

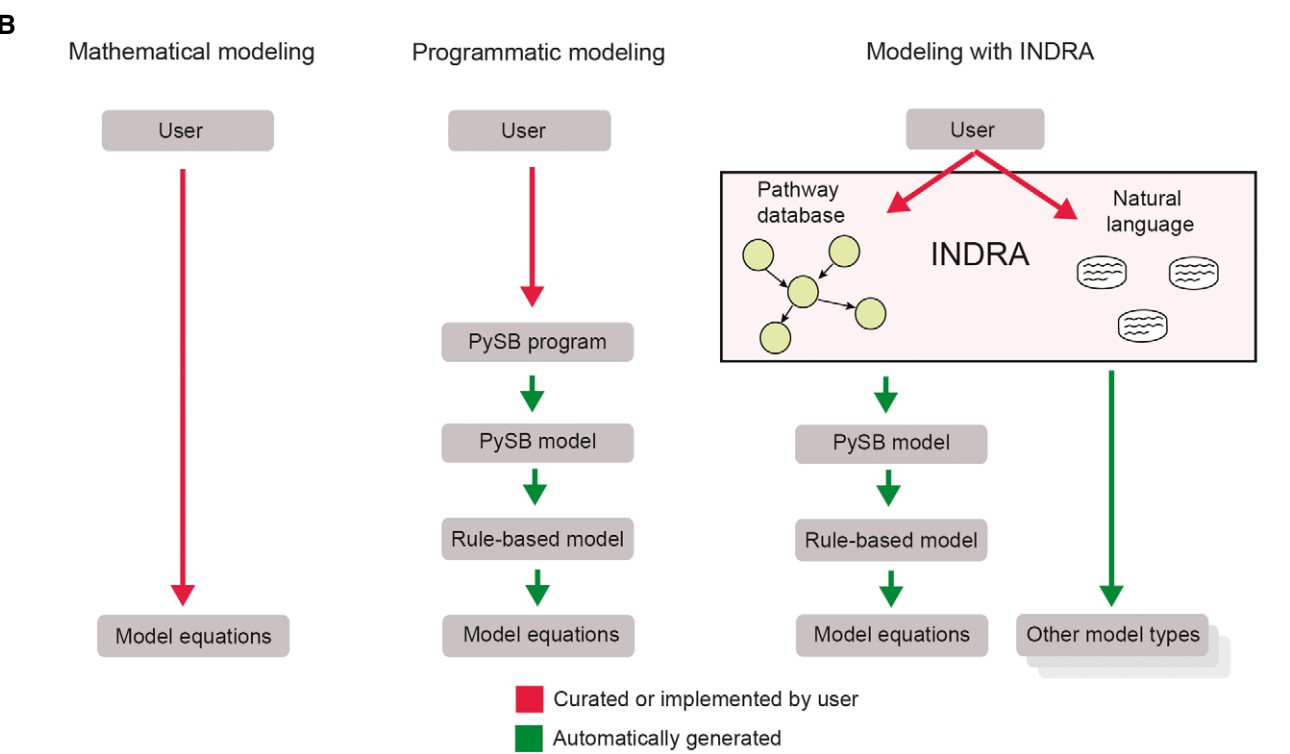

**Figure 8. Approaches to building dynamical models of biochemical mechanisms.**

A  Stages of describing a mechanism from concept to implementation. The mechanism "an enzyme binds a substrate" is shown at different levels of abstraction from mechanistic concept to equation-level implementation. The conceptual description can be expressed in natural language, which can be formalized as an INDRA *Statement* between an enzyme and a substrate *Agent*. The PySB description and a corresponding BioNetGen description (see Box 3) describe a particular implementation of this mechanism in terms of a single rule, which corresponds to a "low-level" instance of three differential equations describing the temporal behavior of the enzyme, substrate, and their complex in time.

B  Comparison of "classical" mathematical modeling (left), programmatic modeling with PySB (center), and modeling with INDRA (right). In each paradigm, red arrows show processes that are done by the user and green arrows show ones that are automatically generated.

extremely rare for one experiment to elucidate the status of all relevant sites of post-translational modification, regulatory subunit binding, or allosteric regulation of an enzyme. Natural language allows biologists to communicate provisional and changing knowledge without prematurely resolving ambiguities or presupposing the biological context or experimental format in which the knowledge might be applied.

**Relationship to previous work**

Several software tools have been developed to partially automate the construction of executable models from bioinformatics databases such as KEGG, Pathway Commons (Ruebenacker *et al*, 2009; Büchel *et al*, 2013; Wrzodek *et al*, 2013; Turei *et al*, 2016).

Automating model translation in this way increases throughput and maintains links between model assumptions and curated findings in databases, eliminating the need for labor-intensive annotations of hand-built models (Le Novère *et al*, 2005). Such approaches have been particularly successful in the field of metabolism in which knowledge about enzyme–substrate reactions is well curated and closely corresponds in level of detail to what is required for mechanistic modeling (Büchel *et al*, 2013). In signal transduction, curation is less complete, the number of molecular states and interactions is far higher, and networks vary dramatically from one cell type to the next. This complexity has been addressed for the most part by using strictly qualitative formalisms that describe positive and negative influences between nodes (Büchel *et al*, 2013; Turei *et al*, 2016). In contrast, INDRA uses an intermediate representation that

encompasses both mechanistic processes (e.g., phosphorylation) and empirical causal influences (e.g., activation and inhibition). The model assembly procedure makes use of mechanistic information where available, but can incorporate qualitative influence relationships when mechanisms are not known. In its use of an intermediate representation to bridge the gap between elements of mechanistic knowledge and executable models, INDRA *Statements* are related to the graphical meta-model for rule-based modeling developed by Basso-Blandin *et al* (2016), which represents binding and modification actions at the level of structural features within agents (e.g., domains and key residues). In this way, their framework is complementary to INDRA *Statements*, and the two approaches could be productively integrated.

Integrated Network and Dynamical Reasoning Assembler's ability to assemble information from knowledge sources into annotated, exchangeable, and extensible models relies heavily on the existence of community standards such as SBO (Courtot *et al*, 2011) and MIRIAM (Le Novère *et al*, 2005), and on structured resources including identifiers.org (Juty *et al*, 2012), UniProt (The UniProt Consortium, 2015), CHEBI (Degtyarenko *et al*, 2008). For an extensive overview of the role of these resources in building large, reusable models, we refer the reader to (Waltemath *et al*, 2016). Early instances of software systems for converting input and output formats allowed one-to-one conversion from BioPAX to SBML (Ruebenacker *et al*, 2009). Cell Designer (Funahashi *et al*, 2008) accepts input in formats such as BioPAX and makes plugins such as SBML Squeezer (Dräger *et al*, 2015) available for export into SBML. Similarly, Cytoscape (Cline *et al*, 2007) makes it possible to import protein interactions from multiple databases and output the results to SBML. More recent one-to-many tools translate information from a single knowledge source into multiple output formats (Wrzodek *et al*, 2013), while many-to-one tools aggregate pathway information from many sources but target a single output format (Turei *et al*, 2016).

By uncoupling knowledge-level statements from a particular formal implementation, whether graphical or mathematical, natural language modeling is complementary to and compatible with a wide variety of input and output formats. In the case of INDRA, an intermediate representation enables a wide variety of many-to-many conversions involving text, BioPAX, BEL, PySB, BNGL, SBML, ODEs, logical models and graph-based formats such as SBGN (an INDRA-assembled SBGN graph of the model presented in Fig 5C is shown in Appendix Fig S9). Further integration of natural language and graphical modeling, for example, by coupling INDRA to SBGNViz graphical interface (Sari *et al*, 2015), will improve the quality of human–machine interaction and further facilitate model assembly and exploration.

## Limitations and future extensions of INDRA

An appealing feature of using natural language to build models is that it is immediately accessible to all biologists. However, this does not necessarily imply that INDRA will allow modeling laypersons to directly build and use sophisticated models, as the use of natural language does not in and of itself address many of the other challenges in developing a meaningful dynamical model, including determination of parameter sensitivity, investigation of network dynamics, insight into combinatorial complexity, multistability, oscillations. We

therefore propose that natural language modeling would be useful in facilitating collaboration between biologists with domain-specific expertise and computational biologists. The advantage of natural language in this context is that it makes it easy for teams to communicate about biological hypotheses and mechanisms without becoming mired in details of model implementation. For experienced modelers, INDRA offers a means to efficiently build multiple model types from a single set of high-level assumptions, provided that the model can be described in terms of molecular mechanisms and assembled using available policies. By design, the software does not perform parameter estimation, simulation, or model analysis, leaving these tasks to the many existing tools and methods.

Limitations in model construction using INDRA can be grouped into two categories: (i) issues relating to the reading of natural language by external NLP systems and (ii) limitations in the representation and assembly of mechanisms in INDRA. In this paper, we construct models using simple declarative sentences that lack much of the complexity and ambiguity of spoken language and the scientific literature. Declarative language can express a wide variety of biological mechanisms at different levels of detail and ambiguity and it reduces many of the difficulties associated with NLP-based extraction of biological mechanisms. Although TRIPS and INDRA are robust to variation in syntax and naming conventions, they cannot understand all possible ways a concept can be stated; for example, "*Wip1 makes ATM inactive*" is not recognized as a substitute for "*Wip1 inactivates ATM*" (Fig 5). In such cases, rephrasing is usually successful.

The TRIPS system (as well as other NLP systems we tried, such as REACH) *can* be used to process the more complex and ambiguous language used in scientific publications and they are both state-of-the-art systems with different strengths and weaknesses. Empirical results presented in Allen *et al* (2015) show that TRIPS compares favorably in precision and recall to ten other NLP systems on an event extraction task from biomedical publications, and reaches precision and recall levels close to those produced by human curators. While reading from the biomedical literature is less robust as compared to reading the declarative language used in this paper, the fundamental challenge in generating models directly from literature information is not reading but knowledge assembly. The assembly challenge involves multiple interconnected issues, including (i) the large amount of full and partial redundancy of knowledge generated when mechanisms are read at scale (e.g., MEK phosphorylates ERK vs. MEK1 phosphorylates ERK), (ii) inconsistencies between knowledge collected from multiple sources which may or may not be resolvable based on context, (iii) the distinction between direct physical interactions and indirect effects, and (iv) technical errors such as erroneous entity disambiguation and normalization. In the approach described here, human experts simplify machine reading and assembly by paraphrasing statements about mechanisms into simplified, declarative language. As illustrated in the POMI models of p53 dynamics, the use of simplified language is not only useful for machines, it helps to clarify complex issues for humans as well. However, we are actively working to extend INDRA so it can assemble information from the primary scientific literature into coherent models.

The domains of knowledge covered by INDRA are limited by the scope of the natural language processing, intermediate representation, and assembly procedures developed to date, which do not include all types of biological mechanisms (e.g., lipid biology, microRNA function, epigenetic regulation remain future

extensions). However, INDRA can extract and represent comparable proportions of reactions in signaling, transcriptional regulation, and metabolism, which are widely curated in existing databases (Appendix Table S1). To further extend this coverage, we are (i) updating processors to retrieve a wider range of information, (ii) adding new *Statement* types, and (iii) creating new assembly procedures. Other areas of future development include automated retrieval of binding affinities and kinetic rates for parameter estimation. Encouragingly, the Path2Models software has shown that automated retrieval of kinetic parameters from databases is feasible for metabolic models (Büchel *et al*, 2013), and this approach may be adaptable to signaling pathways as well. Another planned extension involves capturing more abstract observations in addition to mechanistic information. For instance, the experimental finding *"IRS-1 knockdown resulted in reduction of insulin stimulated Akt1 phosphorylation at Ser 473"* (Varma & Khandelwal, 2008) cannot be directly represented as a molecular mechanism. Literature and databases contain a wealth of such indirect, non-mechanistic information that could be used as biological constraints to infer or verify mechanistic models. However, we expect that INDRA will primarily remain a tool for investigating properties of linked biochemical reactions rather than as a general-purpose mathematical modeling tool. As illustrated in the p53 modeling example above (Fig 5), this emphasis requires the modeler to provide an explicit molecular basis for phenomenological properties such as oscillations, switches, delays.

A system such as INDRA allowing biologists to "talk" to a machine about a biological pathway in natural language suggests the possibility that an improved machine could also "talk back" to the human user in a manner analogous to Apple's Siri (Carvunis & Ideker, 2014). At its most basic level, such a system would allow humans and machines to jointly curate knowledge, thereby resolving ambiguities or errors in NLP or assembly. A more sophisticated machine would use its internal knowledge base to autonomously identify relevant reactions, inconsistencies in a user's input, or novel hypotheses arising from model simulation. A computer agent could interact with many human experts simultaneously, facilitating curation and modeling efforts by communities of biologists. We anticipate that such human–machine collaborative systems will be increasingly valuable in making sense of the large and complex datasets and fragmentary mechanistic knowledge that characterize modern biomedicine.

## Materials and Methods

### Software availability

Integrated Network and Dynamical Reasoning Assembler is available under the open-source BSD license. Code and documentation are available via http://indra.bio; the documentation is also included as part of the Appendix. The TRIPS/DRUM system for extracting mechanisms from natural language is available at http://trips.ihmc.us/parser/cgi/drum. INDRA version 1.5.0 was used to obtain all results in the manuscript. INDRA can be imported in a Python environment and integrated with existing Python-based tools directly. To allow the integration of INDRA with non-Python tools, including graphical modeling environments, a REST API is available, through which all input processing and assembly functionalities of INDRA can be used (for more details

on the REST API, see the INDRA documentation attached as part of the Appendix).

### TRIPS interface

The INDRA TRIPS *Interface* is invoked using the top-level function *process_text*. This function queries the TRIPS/DRUM web service via HTTP request, sending the natural language content as input and retrieving extracted events in the EKB-XML format. The *Interface* then creates an instance of the *TripsProcessor* class, which is then used to iteratively search the EKB-XML output, via XPath queries, for entries corresponding to INDRA *Statements*. Extracted *Statements* are stored in the *statements* property of the *TripsProcessor*, which is returned by the *Interface* to the calling function.

### BioPAX/Pathway commons interface

Integrated Network and Dynamical Reasoning Assembler's BioPAX *Interface* either queries the Pathway Commons web service or reads an offline BioPAX OWL file (Box 2). The *Interface* contains three functions that can be used to query the Pathway Commons database via the web service: (i) *process_pc_neighborhood*, which returns the reactions containing one or more query genes; (ii) *process_pc_pathsbetween*, which returns reaction paths connecting the query genes, subject to a path length limit; and (iii) *process_pc_pathsfromto*, which returns reaction paths from a source gene set to a target gene set, subject to a path length limit. The BioPAX *Interface* processes the resulting OWL files using PaxTools (Demir *et al*, 2013), yielding a BioPAX model as a Java object accessible in Python via the *pyjnius* Python-Java bridge (https://github.com/kivy/pyjnius). INDRA's BioPAX *Processor* then uses the BioPAX Patterns package (Babur *et al*, 2014) to query the BioPAX object model for reaction patterns corresponding to INDRA *Statements*.

### BEL/NDEx interface

Integrated Network and Dynamical Reasoning Assembler's BEL *Interface* either reads an offline BEL-RDF file or obtains BEL-RDF from the BEL Large Corpus via the Network Data Exchange (NDEx) web service (Pratt *et al*, 2015). Subnetworks of the BEL Large Corpus are obtained by calling the method *process_ndex_neighborhood*, which retrieves BEL Statements involving one or more query genes. The BEL *Processor* then uses the Python package *rdflib* to query the resulting RDF object for BEL Statements corresponding to INDRA *Statements* via the SPARQL Protocol and RDF Query Language (SPARQL; https://www.w3.org/TR/sparql11-overview).

### Assembly of rule-based models

Assembly of rule-based models is performed by instances of the PySB *Assembler* class. Given a set of INDRA Statements and assembly policies as input, the *make_model* method of the PySB *Assembler* assembles models in two steps. First, information is collected about all molecular entities referenced by the set of *Statements*. This defines the activity types, post-translational modification sites, binding sites, and mutation sites for each Agent, which can then be used to generate the agent "signatures" for the rule-based model. In PySB, the molecular entities of the model are

represented by a set of instances of the PySB *Monomer* class. Because assembly policies chosen by the user govern the nature of binding interactions (e.g., one-step vs. two-step modification), the binding sites and agent signatures must be generated in accordance with the chosen policies at this step. For policies involving explicit binding between proteins (e.g., the *two-step* policy for post-translational modifications), each PySB *Monomer* is given a unique binding site for each interacting partner. The second step is the generation of reaction rules corresponding to each of the input Statements. The PySB *Assembler* iteratively processes each *Statement*, calling the assembly function specific to the Statement type and chosen policy. Depending on the *Statement* type and policy, one or more PySB rules may be generated and added to the PySB model. The PySB model returned by the *make_model* function can then be converted into other formats (Kappa, BNG, SBML, Matlab, etc.) depending on the type of simulation or analysis to be performed (Lopez *et al*, 2013). Importantly, the PySB *Assembler* adds annotations to the generated PySB model that link molecular entities referenced in the model to their identities in reference ontologies (e.g., HGNC and UniProt). These annotations are in turn propagated into SBML and other model formats by existing PySB model export routines.

## Models of p53 activation in response to single- and double-strand break DNA damage

The text defining each model was submitted to the TRIPS web service for processing via INDRA's TRIPS *Interface*. The TRIPS system returned Extraction Knowledge Base graphs (Box 1 and Appendix Section 2.1) from which INDRA's TRIPS *Processor* extracted INDRA *Statements*. These *Statements* were then assembled using INDRA's PySB *Assembler* into a rule-based model. The default "one-step" assembly policy was used, which generates rules in which the subject of an activation, inhibition, and modification changes the state of the object without binding.

The eight sentences constituting the SSB damage response model (Fig 5B) resulted in eight INDRA *Statements* (each of type Activation or Inhibition). For example, the sentence "Active p53 activates Mdm2" was represented as an Activation *Statement* with an additional condition on the *Agent* representing p53, requiring that it be active. During INDRA *Statement* construction, names of genes are standardized to their HGNC gene symbol (Eyre *et al*, 2006); thus, the *Agent* representing "Mdm2" is renamed "MDM2", and the *Agent* representing "p53" is renamed "TP53". Default initial conditions (10,000 molecules, based on a default concentration of $10^{-8}$ molar in a typical HeLa cell volume of $1.6 \times 10^{-12}$ l) generated by the PySB *Assembler* were used for each protein in its inactive state and simulations were started with an initial one active ATR molecule to initiate the activation pathway. The forward rates for activation and inhibition rules were set to $10^{-7}$ molec$^{-1}$ s$^{-1}$ (using a conversion rate of $10^5$ M$^{-1}$ s$^{-1}$ in a typical HeLa cell volume, as above). The forward rate of the rules corresponding to ATR auto-activation and p53 inactivation by Wip1 was modified to be $5 \times 10^{-7}$ molec$^{-1}$ s$^{-1}$, that is, faster than the forward rate of other rules (a summary of all rules and rates is given in the Appendix Section 2.3). PySB's reaction network generation and simulation functions were then used to instantiate the model as a set of eight ordinary differential equations. The model was simulated using the

*scipy* package's built-in *vode* solver for up to 20 h of model time while tracking the amount of active p53, which was then plotted (Fig 5B). Natural language processing for this model took 10 s (here and in the following, this includes network traffic time to and from the web service); the assembly and simulation of the model took < 1 s.

The method for constructing the simple DSB response model (Fig 5C) with ATM was analogous to the SSB model. The same initial amounts and forward rate constants were used as in the previous model, except in this case an initial condition of one active ATM molecule was used, and the inactivation of ATM by Wip1 was given a forward rate of $10^{-5}$ molec$^{-1}$ s$^{-1}$. For this model, the nine natural language sentences were captured in nine INDRA *Statements* and generated into a model of nine rules and finally nine ODEs. The model was again simulated up to 20 h while observing the active form of p53. Similar to the SSB response model, natural language processing for this model took around 10 s, with assembly and simulation taking < 1 s.

The POMI1.0 model (Fig 5E) extends the basic DSB response model by specifying the activation/inhibition processes in more mechanistic detail. The model is described in 10 sentences yielding 12 INDRA *Statements* and a model containing 11 PySB rules and 12 ODEs (via the PySB *Assembler* using the "one-step" policy). The same rate constants were used as in the simple DSB response model; additionally, the degradation rate of Mdm2 was set to $8 \times 10^{-2}$ s$^{-1}$ and the rate of synthesis of Mdm2 by p53 to $2 \times 10^{-2}$ molec$^{-1}$ s$^{-1}$ (a full list of rules and associated rate constants is given in the Appendix Section 2.3). Natural language processing for this model took 14 s; assembly and simulation took < 1 s.

## Models of response to BRAF inhibition

The sentences for the MEMI1.0, 1.1, and 1.2 models were processed with the TRIPS web service via INDRA's TRIPS *Interface*. Natural language processing took 37 s for MEMI1.0, 60 s for MEMI1.1, and 75 s for MEMI1.2. The resulting INDRA *Statements* were then assembled using INDRA's PySB *Assembler* module into a rule-based model using the "two-step" policy for assembling post-translational modifications. Kinetic rate constants were set manually and the initial amounts of each protein were set to correspond in their order of magnitude to typical absolute copy numbers measured across a panel of cancer cell lines in Appendix Tables S5 of Shi *et al* (2016). A summary of the kinetic rates and initial amounts is given in Appendix Tables S5–S7. Each model was instantiated as a system of ordinary differential equations and simulated using the *scipy* Python package's built-in *vode* solver. Each model was started from an initial condition with all proteins in an inactive, unmodified, and unbound state. The models were run to steady state and the values of GTP-bound RAS (active RAS) and phosphorylated ERK were saved. Another simulation was then started from the steady-state values with vemurafenib added and the time courses of active RAS and phosphorylated ERK were normalized against their unperturbed steady-state values and plotted.

## Extensible and executable RAS pathway map

The pathway map was created by processing 47 sentences with TRIPS (see Appendix Section 2.5) to generate 141 INDRA

*Statements*. Reading and extraction of *Statements* took a total of 160 s. The *Statements* were then assembled using INDRA's *Graph Assembler*, which produced a network that was laid out using Graphviz (Ellson *et al*, 2002) as shown in Fig 7A. The same set of *Statements* was then assembled using the INDRA SIF *Assembler* which produced a list of positive and negative interactions between genes that can be interpreted by network visualization software (Shannon *et al*, 2003) and Boolean network simulation tools. The logical functions for each node were generated by combining the state of parent nodes such that the presence of any activating input in an *on* state and the absence of any inhibitory inputs in an *on* state resulted in the node's value taking an *on* state at the next time step (logical rules are given in Appendix Section 2.5). Boolean network simulations were performed using the *boolean2* package (Albert *et al*, 2008). The presence of growth factors was modeled by setting the value of the "GROWTH-FACTOR" node to True; inhibitor effects were modeled by clamping the values of the inhibited protein nodes to False. For each condition, 100 independent traces were simulated using asynchronous updates on the nodes (which results in stochastic behavior) and the average of the value of each node (with 0 corresponding to the low and 1 to the high state of each node) was taken across all simulations to produce the time course plots in Fig 7D.

### Data availability

The POMI1.0 and MEMI1.0-1.2 models are provided in Model Files EV1 in SBML, BNGL, Kappa, and PySB formats, in addition to the natural language text files used to build them. The RAS pathway model and its extension are provided in SIF and Boolean network formats. Code used to generate these models is part of the INDRA repository and can be found in the *models* folder of https://github.com/sorgerlab/indra.

**Expanded View** for this article is available online.

### Acknowledgements

This work was funded by ARO Grants W911NF-14-1-0397 and W911NF-15-1-0544 to PKS and W911NF-14-1-0391 to LG under the DARPA Big Mechanism and Communicating with Computers programs, and by NIGMS Grant P50GM107618 to PKS. We would like to acknowledge Russ Harmer, Walter Fontana, and Dexter Pratt for useful discussions and their valuable suggestions, as well as James Allen, Choh Man Teng, and Will de Beaumont for their contribution to the development of the TRIPS NLP system.

### Author contributions

BMG and JAB designed and implemented INDRA. BMG, JAB, KS, JLM, LG, and PKS conceived the overall approach. BMG, JAB, KS, and PKS wrote the paper.

### Conflict of interest

The authors declare that they have no conflict of interest.

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
