## [Review Process File · Molecular Systems Biology]

From word models to executable models of signaling networks using automated assembly

Benjamin M Gyori, John A Bachman, Kartik Subramanian, Jeremy L Muhlich, Lucian Galescu & Peter K Sorger

Corresponding author: Peter K Sorger, Harvard Medical School

Review timeline:

Submission date:	23 March 2017
Editorial Decision:	03 May 2017
Revision received:	14 September 2017
Editorial Decision:	18 October 2017
Revision received:	27 October 2017
Accepted:	27 October 2017

Editor: Maria Polychronidou

Transaction Report:

1st Editorial Decision

03 May 2017

Thank you again for submitting your work to Molecular Systems Biology. We have now heard back from the four referees who agreed to evaluate your study. As you will see below, the reviewers appreciate that the presented approach seems interesting and potentially useful for the scientific community. However, they raise a number of concerns, which should be carefully addressed in a revision of the manuscript.

The reviewers' recommendations are rather clear so I think that there is no need to repeat all the points listed below. A particular issue that needs to be addressed refers to the sensitivity of the approach to specific wording, since this seems to represent a significant limitation. Moreover, the reviewers recommend extending the discussion on the potential limitations of the approach, how generalizable it is to other types of biological networks and the degree of expert knowledge required to use the approach.

REVIEWER REPORTS

Reviewer #1:

The manuscript by Gyori and colleagues 'From word models to executable models...' addresses the problem of developing mechanistic dynamic models of cell signaling with increased automation and

reduced technical barriers, and it proposes an approach based on natural language processing (NLP) for this purpose. Specifically, the INDRA software system combines: (i) existing NLP systems tailored to the identification of biological mechanisms from text (statements); (ii) mapping of interpreted text results and other information sources such as BioPAX and BEL to an internal representation of mechanisms (template-based Statements); and (iii) assembly of executable dynamic models of different type (e.g., ODE-based or Boolean) using policies for the specification of all required details on model granularity. Three examples from cell signaling of increasing complexity are presented to demonstrate model assembly from simple (manually written) textual statements, illustrating, for example, simple ways of augmenting incomplete pathway information (p53 example), and using INDRA statements for collaborative curation of pathway knowledge (Ras example).

The manuscript describes a novel approach for model construction in systems biology by recruiting NLP methods to the field, which has the potential of facilitating model construction as well as communication on models without need for deep technical capabilities. The core methodological contributions are the internal representation of mechanisms of cell signaling via templates (statements), which interface to NLP and databases, and the assembly policies for interfacing to existing rule-based modeling (and similar) systems. The approach is technically sound, with overall consistent evaluation of the methods (e.g., regarding wording restrictions on text processing; but see qualifications below). The example studies recapitulate known biology (p53 and Ras examples), but they also provide hypotheses on novel mechanisms (melanoma MAPK example). Finally, limitations of the current system, in particular regarding model construction from direct parsing of literature or of databases, appear adequately discussed.

Overall, the proposed system and approach are clearly novel and of potentially wider use (as alternatives to graphical or formal modeling language-based systems that currently exist) for mechanistic model development in systems biology. The examples demonstrate ease of model development for cell signaling. However, the approach might not generalize to the extent suggested in the manuscript, as discussed below.

Major comments:

(i) INDRA Statements are the key concept of the approach, serving as an intermediate representation (without additions or assumptions, claimed in the manuscript). The Statement templates and their structure appear, however, very tailor-made for certain aspects of cell signaling (e.g., the summary in Fig. S2 contains very specific classes such as RasGEF and RasGAP), and there is a risk that the design will not sufficiently generalize to other types of biological networks. Given standard ontologies and community standards for network representations (e.g., SBO and BioPAX), the definitions of templates require more justification and precision, in particular regarding the formal mapping between INDRA and standard classes (which should be coded in the Processor modules).

(ii) Similarly, in terms of generalization and compatibility with existing systems, it is advisable to clarify relations to BioPAX and BEL, and give specific reasons for why automated extraction (instead of manual curation of statements after database searches already implemented in INDRA) fails or is limited. For example, statements such as p.25 'Perhaps unexpectedly, constructing executable models from pathway databases ...' appear to general and they may rely on outdated references. In particular, comments on how specific processors need to be, for different (alternative) inputs, and to what extent processors are extensible (in contrast to the need to develop specific processors for each potential data source) could clarify these points.

(iii) Another aspect that warrant further discussion regards transparency of the INDRA system, especially concerning debugging. For example, how can users distinguish between cases of correct and incorrect processing of word models as in Fig. 4D? Does the INDRA system support textual or (ideally, as in Fig. 2 examples) graphical output at the statement level? At which levels are errors generated for word statements that cannot be processed, or can be processed only partially?

(iv) On usability, the manuscript emphasizes ease of communication on models and limited technical challenges for their construction. However, for the p53 and MAPK examples, completion or modification of models to represent biological phenomena as discussed in the text requires basic expertise in systems dynamics / modeling (e.g., identification of missing reactions, association of

positive feedback associated with time delay) that are not commonly found among 'traditional' biologists. In this view, statements such as p.16 'Introducing alternative assumptions and mechanisms using natural language is straightforward and can be accomplished by individuals with little or no technical expertise.' appear exaggerated and a more detailed discussion of technical requirements on the user for model implementation vs. analysis is warranted.

(v) Finally, among the examples of INDRA usage, the example of Ras signaling should be expanded with additional evidence. A 'visually comparable' pathway map represents as a rather weak argument, given that Ras signaling was used more rigorously in the original DRUM publication (Allen, 2015); it could also be revealing to compare INDRA output (formally) with standardized pathway maps, such as those published in SBGN notation for several mammalian signaling pathways. In addition, while the Boolean simulation for Ras demonstrates a key concept of INDRA (assembly into models of different type), the discussed simulation represent a very simple case study that could be omitted.

Minor comments:

(i) Title, abstract and general: The term 'word models' is not defined until late in the introduction; a definition in the abstract would be helpful.

(ii) Legend to Fig. 1C: '... creates [an] Extraction Knowledge ...'.

(iii) The introduction provides verbal arguments on the conceptual differences between INDRA and existing systems, which are well summarized in Fig. 8 and associated text. To better convey the concepts behind INDRA in a non-technical way it may be advisable to move this figure and (parts of the) text to the introduction.

(iv) More recent developments towards community standards in systems biology should be referenced, for example, for whole-cell modeling (IEEE TRANSACTIONS ON BIOMEDICAL ENGINEERING, VOL. 63, NO. 10, OCTOBER 2016).

Reviewer #2:

The manuscript by Peter Sorger and colleagues describes a set of algorithms and semantics (or as they call it an approach) where signaling models described in human readable text are translated in computable signaling models (e.g. using ODEs) that allow simulation of the dynamics of those networks. The article first describes the steps of the approach and then illustrates the use on building models for p53 and EGFR/RAS/RAF signaling. The article is well written and well understandable. As far as I see the approach seems to work (though there is no benchmark by which one can really measure this), but the individual steps are not really compared to the state of the art. When does the natural language processing fail? How complicated can these texts be?

My main concern is that I don't really see that the approach is a major step forward in making signaling models.

The authors sell it as it is the way so that also laymen can generate complex models, but I actually see it the other way around. The models are generated basically by a black box, and very similar statements can lead to very different models, especially given the combinatorial complexity, and the authors show one example of such combinatorial explosion.

The major advantage of computational models is (at least that is my view of it) that these are very explicit about the modeling assumption. This benefit is gone with this approach, so that other (more graphical ways) would be more explicit, such as SBGN editors or other editors.

The other application where I could see this approach being useful would be to extract crude models from full text. However, for this the text mining seems to be rather limited in potential.

Reviewer #3:

Gyori, Bachman, and colleagues describe a new framework for building computational models of molecular systems, implemented through the software package INDRA, in which natural language commonly used to describe molecular interactions and mechanisms is parsed and converted into executable models with well-defined structures. The approach involves multiple layers of processing and incorporates multiple external resources - both to carry out the processing steps as well as to draw on existing knowledge from biological databases. INDRA is applied to three test cases of human signaling networks that represent varying levels of complexity as well as different modeling purposes. These examples serve to demonstrate the software's utility and limitations.

The study is timely because of a general consensus in the biomedical community about the disparity between data and knowledge. It also tackles the challenging task of natural language processing (though the functionality of the method is somewhat limited to very specific types of grammatical forms used to describe biological systems). As a piece of computer software, INDRA is well described, documented, and conceptually easy to follow. It is likely to be a valuable contribution to the systems biology community. I did not test the software as part of this review but trust that, like most open-source resources, the greatest positive impact on its maturation as a tool will come from the public.

My comments are intended to sharpen the paper, mainly by drawing out precisely where INDRA belongs as a tool in the grand scheme of deriving quantitative models from the vast sum of experimental knowledge available to us.

Major comments:

1. Through the exercise of deciphering word models, the study highlights the crux of model building: the encoding of imprecise or often vague ideas about biological mechanism into unambiguous mathematical form. For me, the greatest take-away from the paper was that, even with a tool that understands spoken language, building a molecular model is an iterative dialogue between human and computer in which common ground is achieved by balancing model purpose with the requirement for discrete structures. This principle becomes clear through all three pathway examples, which each required some degree of expert knowledge in order to achieve a specific modeling purpose.
2. It seems that INDRA ought to be able to handle assembly of statements in which two mechanisms are embedded. For example, we might say that "EGFR activates SOS" and that "EGFR activates SHC". Both statements are true. The first statement is correct but has not defined the intermediate (SHC) that mediates the activation of SOS by EGFR. Similarly, "MEK1 phosphorylates ERK2" and "MEK1 phosphorylates ERK2 at threonine 185" should be merged into a common mechanism that reduces to the second statement. Does INDRA allow for these types of overlapping word models to be supplied and reconciled?
3. Supposing INDRA may be applied to large collections of natural language, how would it handle apparently contradictory information (e.g., "X activates Z; X inhibits Z")? One could imagine a scenario in which both statements are true (e.g., an incoherent feedforward loop in which X activates Z directly but inhibits Z through intermediate Y). It may be that this capability of assembling language into an executable model is beyond the scope of the current implementation of INDRA, which is aimed primarily at providing a modeling interface that begins with natural language. However, the capability of INDRA to establish equivalence between multiple aliases of the same biological entities (described on p. 16) leads me to believe that it is intended (at least eventually) for application to large databases of mechanistic information in which contradictory mechanisms are almost certain to exist. I would like to see a more precise description of the context/limitations/eventual utility of INDRA.
4. An unexpected consequence of INDRA is that it may serve a reverse and complementary function: to improve the precision of how we speak about biological mechanisms. The example of the problematic phrase "ATM activates itself" highlights how a tool like INDRA could be useful in this regard.

Minor comments:

1. The statement "...there is little evidence that creation of new word models is being routinely supplemented by formal modeling approaches" (p. 2) is unclear at this point in the manuscript. I think it becomes clear in the Results section that Assembly can be of great benefit to improving the precision of word models and to refining and challenging existing statements about biological mechanism.
2. The acronym SBML is used before it is defined (p. 2).
3. The acronym UML is used before it is defined (p. 8).
4. EKB is defined in the glossary on p. 5 but then redefined on pp. 8 and 10.
5. In general, the paper is very acronym-rich, but I don't see any way around this.
6. It would be useful for many readers to have a reference associated with the sentence that ends "...reducing the identifiability of the model." (p. 14).

Reviewer #4:

Summary

- Describe your understanding of the story

The authors describe a novel computational approach to bridge the gap between natural language and executable models. The paper describes how events returned from natural language parsers can be processed into "statements" which capture biological processes. The process to assemble these statements into existing modeling formats suitable for solving with existing software is also described.

- What are the key conclusions: specific findings and concepts

- Executable models can be generated from natural language
- Executable models generated in this way qualitatively recreate expected behavior
- Natural language processed with the tool presented here is an efficient collaborative tool to improve collaboration between experts in a biological domain with experts in computational modeling.

- What were the methodology and model system used in this study

- A novel software package (INDRA) was developed in python.
- An existing natural language processes (TRIPS) was leveraged along with interfaces to existing databases such as PAX and the existing modeling framework PySB was used for execution.
- Three biological domains were analyzed by constructing natural language statements and compared the quality of simulated outputs to expected results.

General remarks

- Are you convinced of the key conclusions?

The authors successfully demonstrate that executable models can reliably generated from natural language statements using their newly developed tool.

They also successfully show -qualitatively- that expected behavior can be recapitulated by models generated from natural language.

The reliability of the approach and potential of natural language models in enabling collaboration is somewhat convincing. However, the examples in the study failed to produce expected results with initial natural language descriptions and iterative adjusting of phraseology and re-running the pipeline was required to obtain the desired result. The sensitivity of the approach to specific wording limits the use of the technique to those with some knowledge of the conversion process as those unfamiliar with underlying data structures will quickly resort to a trial and error approach to rewording the descriptions.

Inevitable combinatorial complexity (acknowledged by the authors), combined with the author's

choice of reaction scheme (two-step), generated models that were unnecessarily complex. This complexity hampers further modeling and parameterization required to make predictive models and would also be a barrier to the use of the described technique as a collaborative tool.

- Place the work in its context.

Most progress in the area of enabling technologies/software for computational modeling has been focused on software environments for the construction of pathway diagrams and the creation of models from these diagrams (Cell Designer, COPASI, SimBiology). Diagrammatic representations have been widely and successfully used as tools for collaboration between biological and computational experts.

The approach described here provides an alternative to diagrammatic formalisms. This study represents an important component in a long-term goal of systems biology, which is mining biological literature in a manner amenable to theoretical analysis. When combined with automated literature text-mining and existing parameter estimation techniques the work presented here will presumably provide an important component of a complete automated model construction pipeline able to construct executable models for any biological domain.

- What is the nature of the advance (conceptual, technical, clinical)?

The work describes a technical advance in converting parsed text statements into executable modeling.

- How significant is the advance compared to previous knowledge?

In isolation in its current form the study represents a moderate advance, however if issues of complexity explosion and sensitivity to wording are improved then the tool presented here will be an important advance enabling widely-useful, automated, model construction and analysis from scientific literature.

- What audience will be interested in this study?

The study will be of interest to systems biologists working on model construction and dynamical pathway analysis.

Major points

-Specific criticisms related to key conclusions

The sensitivity to specific wording generating qualitatively distinct behavior (demonstrated by the ATM trans-phosphorylation example) seems to be a significant limitation in the applicability of the approach. While the study claims to enable model construction by individuals with little or no technical expertise, significant knowledge of which phrasing changes can generate significantly different dynamical systems is required to successfully use the technique.

Most successful model-driven studies utilize elegant models containing ~1 reaction for each biochemically significant process. The "word model" format here is amenable to such elegant model construction, however, in its current implementation that is not achieved.

The authors argue, somewhat unconvincingly, that natural language provides a superior collaborative framework to traditional wiring diagrams. Many of the weaknesses of diagrammatic representations are shared by natural language descriptions of processes. Vague wording inhibit quantitative analysis in the same way vague diagrammatic representations do. Large lists of plain-text statements become difficult to understand with scale in the same way diagrammatic representations do. The authors even use an "informal" diagram to construct a word model, a step that requires similar effort and expertise as converting an "informal" diagram into a mathematically complete notation such as SBGN in widely used software.

-Specify experiments or analyses required to demonstrate the conclusions

Sensitivity to wording should be addressed. Rather than allowing "ATM trans-phosphorylation" to silently generate significantly different models to "Active ATM phosphorylates ATM" a step in which ambiguities such as these are identified and clarified when INDRA is run would greatly improve the approach.

Combinatorial explosion must be addressed. Michaelis-Menten kinetics should be included in addition to the 1- and 2-process mechanisms to enable construction of elegant models that include saturating kinetics. If the number of generated ODEs is similar to the number of phrases used as input the technique will be greatly improved.

The study's focus on replacing diagrammatic representations should be adjusted to instead focus on the important role of INDRA in forming part of a highly enabling, automated pipeline from literature to automatically generated and parameterized models.

Where appropriate the specific sections of the manuscript of most relevance to answering these concerns are highlighted below.

Minor points

-Easily addressable points

Introduction.

Paragraph 1:

Include modeling of ERK, P53, NFkB and other signaling networks into examples of mechanisms elucidated by computational modeling.

Is it possible to generate some estimate of how many papers contain formal language of high enough quality to enable "word model" construction and don't contain a diagram in a formal notation enabling model construction.

Paragraph 2:

Add Copasi (Hoops et al 2006) and BioModels database (Juty et al. 2015) to list of tools for improving mechanistic modeling.

Paragraph 5:

Natural language suffers the same issues of differences in style/syntax between authors as differences in diagrammatic notation. There are many phrases that could represent the same arrow on a diagram. While paragraph 3 dismisses diagrams as struggling to deal with large complexity and large scope, how natural language descriptions overcome this is unsatisfactorily described.

"sophisticated NLP algorithms" is vague phraseology that should be replaced with a better description of the algorithmic innovation or novel application of existing specified algorithms.

Results

Page 13:

The "one-step policy" in which an enzyme-mediated reaction is represented by a single process does not necessarily ignore enzyme saturation and require only a single parameter, as this single step process could be represented with Michaelis-Menten (MM) kinetics, which is supported by PySB and uses 2 parameters. The one step policy is valid in excess enzyme regimes. The one step policy with MM kinetics extends the one-step policy to include enzyme saturating regimes. This is the most widely used formalism in model building and is currently ignored. The 2-step policy is only required if MM assumptions are not met. The addition of MM kinetics may prevent a model with 28 PySB rules exploding into a model with 99 differential equations, and a model with 34 rules exploding into 275 ODES, while still capturing important kinetic characteristics.

Page 15:

In reference to the constitutive negative regulation of Mdm2 and Wip1 clarity should be provided as to which parameters in the ODE model of Batchelor et al, 2011 were missing from the diagram. I am unable to locate p14ARF or HIPK2 inhibition in the paper, supplement or MATLAB files of Batchelor et al. If other studies were used for the addition of this mechanism then that should be clarified as it seems the model introduced here required additional mechanism to recreate the same behavior seen in Batchelor et al. 2011.

Time delay is added using a positive feedback however this limits the applicability of the approach as positive feedback does not exist in all systems with delay. INDRA should either be extended to accept word models that include "X activates Y with a delay", or this limitation should be explicit. "Oscillation was robust to changes in kinetic parameters" may be confusing as properties such as amplitude and frequency are highly parameter sensitive. Suggest: "The presence of oscillations was robust to changes".

While it is clear that essential reactions for the dynamical system were omitted from diagrams they were also omitted from the text and had to be manually added. This is not an argument for word-models over diagrammatic representations as both were insufficient from a single study. Please clarify. The text describes the manual construction of word models from diagrams and other studies and then states that "machine-assembled" word models are useful. Please clarify that these are manually constructed word models that are then machine-assembled into an executable form.

Page 16:

Biologically descriptive phrases such as ATM trans-phosphorylates itself should be added to the tested phrases here.

Page 21:

While the combinatorial complexity problem is addressed here the previous paragraph describes INDRA as a starting point for modeling. Models of this complexity are poor starting points and this must be should be addressed before IDRA can be considered a recommending starting point for modeling over diagrammatic inputs like symbiology or user-friendly packages such as COPASI. The time investment required to learn software such as COPASI/SimBiology/CellDesigner is not far greater than the learning required to understand the impact of phraseology on dynamics and in the corrent format they generate more elegant models as "starting points". Either address complexity of generated models as described above or clarify the expected use of the large models generated, I would not recommend them as a starting point for model construction but perhaps as tools to identify knowledge gaps, generate diagrams and target manual model construction. Intead highlight the importance of this approach in enabling automated, text-mined, model construction from the literature here.

Page 25:

The following sentence is inaccurate: "molecular species are directly instantiated as variables and related to each other using one or more differential equations for each mass action reaction". Rather in dynamical systems modeling molecular species are each represented by a differential equation and the terms of the differential equations are determined by the reactions.

Page 27:

Cytoscape offers importing from various databases and outputting SBML among other formats and should be mentioned here (Cline et al 2007 Nature Protocols). Cell Designer also enables input from a number of file formats including BIOPAX, simulation within Cell Designer and export as SBGN (Funahashi et al 2003) including plugins such as SBMLsqueezer which enable generation of rate equations from diagrams that can be created by easily without mathematical skills. Include these here.

-Presentation and style

Good

-Trivial mistakes

Figure 4: Rule 1 of one step policy uses 'kc' parameter in diagram and 'kf' parameter in code.

Response to General Comments Raised by Multiple Reviewers

On the issue of the robustness, transparency, and accessibility of the natural language modeling approach, reviewers raised multiple interrelated concerns, addressed in detail below.

GC1. Robustness of the NLP itself.

INDRA makes use of two independent NLP systems, TRIPS/DRUM (Allen et al. 2015) and REACH (Valenzuela-Escarcega et al. 2015). Notably, both of these systems can be, and currently are, used to extract mechanisms not only from human-curated text but also from the scientific literature. The performance of the TRIPS system (our primary focus in this paper) and ten other NLP systems was formally evaluated by a third party (MITRE Corporation) for its performance in extracting mechanisms from passages in the scientific literature (Figures 8 and 9 of Allen et al., 2015). The performance of TRIPS compared favorably to that of human curators in terms of precision and recall (NB: the authors of our manuscript include members of one of the two teams of human curators used as a reference in the NLP evaluation exercise; Figure 9 of Allen et al. 2015). Both the TRIPS and REACH NLP systems are available open-source, and have been subjected to continuous improvement over a period of several years via user feedback.

For the approach described in this paper, in which a human uses the software system to extract mechanisms from curated declarative text, a similar evaluation of NLP performance alone is difficult because a human quickly adapts to the existing capabilities of the system. The most meaningful evaluation of the joint human-machine system would therefore involve evaluation of human performance on a modeling task both with and without machine assistance, a study we have not yet undertaken. It is also not clear what the criterion should be: efficiency versus precision in model assembly, for example.

Within the scope of this study, we have attempted to qualitatively illustrate robustness by exploring alternative phrasings for statements such as "Wip1 inactivates ATM" (Figure 4D). We show that eight different phrasings result in extraction of the same mechanism and assembly of identical models; among three phrasings that do not result in successful extraction, two involve grammatical or spelling errors (that humans but not machines can overcome) and only one ("Wip1 makes ATM inactive") is syntactically valid. As biological modelers ourselves, we have found the TRIPS system, though imperfect, to be quite robust for the purposes described in this paper; we also invite the reviewers to experiment directly with the TRIPS/DRUM parser at <http://trips.ihmc.us/parser/cgi/drum>.

GC2. Role of wording in determining mechanisms.

On the issue of the sensitivity of the approach to specific wording, Reviewer 2 pointed to the fact that "*similar statements can lead to very different models*" and Reviewer 4 pointed to "*iterative adjusting of phraseology,*" in particular the distinction between "*ATM phosphorylates itself*" (autophosphorylation in *cis*) and "*ATM phosphorylates ATM*" (autophosphorylation in *trans*). We believe that these concerns reflect a misinterpretation of our work and that sensitivity to these differences in phrasing is a feature rather than a bug of the system. We have extensively re-written the relevant section of the manuscript to make this point clear (pages 17 and 18).

One of the primary virtues of formal computational models is that they allow the assessment of the impact of assumptions on system behavior; this often uncovers innocuous assumptions that have a major impact. The sensitivity of models on assumptions is true whether a model is constructed from sets of differential equations, interaction rules, or sentences. Constructing models using natural language highlights the fact that the verbal descriptions constituting the foundation of most knowledge about biochemical mechanisms (e.g. in reviews or textbooks) can be unexpectedly ambiguous or inaccurate and can have dramatic effects on model behavior. As a simple example, consider a model built from the three mechanistic assertions "*A binds B; B binds C; C binds A.*" This leads not only to assembly of an anticipated A-B-C heterotrimer but also an unlimited number of polymers of the form A-B-C-A-B-C-A, etc. This is the result of the fact that the mechanistic information contained in the assertions does not constrain the system to the assembly of trimers, and is an illustration of combinatorial complexity. Depending on the context of the modeling study, the appearance of polymers could represent a meaningful new biological hypothesis or merely an unintended consequence of insufficiently explicit biochemical assertions (which could then be clarified).

We do not claim that the use of natural language for modeling absolves a modeler of precisely defining a mechanism. In this sense, we agree with Reviewer 2 that "*vague wording inhibit[s] quantitative analysis in the same way vague diagrammatic representations do.*" Indeed, one of our goals (illustrated most clearly in the case of the p53 models) is to provide a means by which word models, diagrams and computational models can be precisely aligned and ambiguity avoided. As Reviewer 3 correctly notes, modeling with natural language highlights the ambiguities implicit in verbal descriptions, with the potential "*unexpected consequence*" of improving "*the precision of how we speak about biological mechanisms.*" A simple example would be "MEK phosphorylates ERK" versus "Active MEK phosphorylates ERK".

Reviewer 2 points out that "*the major advantage of computational models is...that these are explicit about the modeling assumptions.*" This is only partly true. A formal model does make mechanistic assumptions explicit—but only because a human modeler has *already resolved* ambiguities inherent in knowledge sources (which include domain experts and the scientific literature in the form of verbal assertions). In this case, the conversion of high-level mechanistic knowledge into a formal representation occurs *only in the mind of the modeler* and is subject to (generally undocumented) modeling practices, conventions, and biases (see also Figure 8B in our manuscript). In INDRA, any ambiguities in the natural language input are resolved during the specification of explicit policies (Figure 4). Moreover, intermediate stages in model construction (such as INDRA Statements) and the final formal model resulting from automated assembly can be inspected in multiple formats (including SBML and, with a recent extension, SBGN), making assembly as transparent as any existing process involving humans alone.

Formal models are often accompanied by informal descriptions or diagrams summarizing the model for non-experts, but the underlying mechanistic details, which can be highly consequential for model behavior, are usually absent from such informal summaries and thus hidden from readers. It is precisely this point that we aimed to make with the p53 modeling example (Figure 5): namely that an informal description of the p53 network and its dynamics (in this case a diagram) was an *incomplete representation* of the underlying formal assumptions. On the other hand, the full natural language model, (along with selected assembly policies as in Figure 4), is a fully self-contained description of the assumptions needed to generate p53 oscillations and pulses.

In conclusion, we are in broad agreement with the reviewer’s point of view except that we believe we have demonstrated that INDRA serves to resolve ambiguities that already exist in natural language. We have modified the text extensively to make this point (pages 17 and 18).

GC3. Transparency of the parsing and assembly process.

For the reasons outlined in GC2 above, we believe that for many readers of modeling studies, natural language modeling represents a more, rather than less, transparent approach. This is because assumptions and policies are transparent and explicit, and assembled models are automatically associated with semantic annotations. For non-experts, computational models are often highly inaccessible and natural language is one way of breaking down the barrier.

We acknowledge however, that for experienced model builders unfamiliar with rule-based modeling and INDRA, our approach might be less transparent than modeling directly in a formal language. To address this issue, we have enhanced INDRA to support alternative representation formats for Statements (i.e., textual and graphical) to make clear how specific phrases are converted into Statements and subsequently assembled into models. We highlight the new features of the system in the revised text and in two new interactive iPython notebooks attached as part of the Appendix. In the first iPython notebook, we study in detail two sentences highlighted by the reviewers (“Active ATM phosphorylates itself” and “Active ATM phosphorylates another ATM molecule”) and show how inspection of the underlying Statements makes clear the difference in mechanism resulting from parsing the two phrases.

Specifically, the first sentence generates an Autophosphorylation INDRA Statement which is displayed as Autophosphorylation(ATM(activity: True)), whereas the second sentence yields Phosphorylation(ATM(activity: True), ATM()). The hierarchical structure of these statements can be inspected graphically using newly developed tools. For example, using draw_stmt_graph, the Phosphorylation statement resulting from the second sentence is displayed as in Figure 1. The iPython notebook also includes examples of the same INDRA Statements in a human-readable and editable JSON format that allows specific fields to be inspected.

Figure 1: Graphical representation of a Phosphorylation Statement

The other issue raised by the reviewers with respect to transparency of the modeling process with INDRA concerns the final reaction model produced by assembly. In the case of the ATM example, the supplementary iPython notebook highlights the ability to interactively inspect the monomers, rules and annotations associated with the assembled PySB models. In the case of the Autophosphorylation Statement, the assembled model is seen to contain a first-order rule involving a single ATM molecule,

```
ATM(activity='active', phospho='u') >> ATM(activity='active', phospho='p')
```

whereas the model with the Phosphorylation Statement clearly contains a bimolecular reaction rule involving distinct molecules:

```
ATM(activity='active') + ATM(phospho='u') >>  
ATM(activity='active') + ATM(phospho='p').
```

Unexpected combinatorial complexity of the reaction networks produced by INDRA was a particular concern of reviewers. As discussed in GC2 above, in general we believe that combinatorial complexity in reactions can and should be addressed through iterative refinement of mechanistic assertions, or the choice of suitable policies. A prerequisite for such iterative refinement is the ability to inspect the reaction model to identify the source of unexpected combinatorial explosions. In a second accompanying iPython notebook, we use a simplified example drawn from the BRAF model discussed in the paper (MEMI 1.2) to show how species and reactions engendered by a particular set of assumptions can be inspected interactively. Specifically, we show how assembly of a model from natural language using a two-step policy leads to a number of unrealistic species, including a complex with vemurafenib, RAF, MEK, and ERK simultaneously bound. We show how refining the language (e.g., “RAF *not bound to Vemurafenib* phosphorylates MEK”) cuts down on combinatorial complexity. Responding to the concerns voiced by Reviewer 4, we also show how the use of a one-step Michaelis-Menten policy (which we have newly implemented in the revised manuscript) can further reduce the complexity of the model.

GC4. Generalizability to other types of biological networks

An acknowledged limitation of our original submission was that INDRA Statements lacked the ability to represent general chemical conversions, since it focused on specific types of conversions relevant to signal transduction (e.g., post-translational modifications and GEF/GAP relationships). To address this issue, we have created a *Conversion Statement* to represent reactions where one or more reactants are converted into one or more products, with or without an enzymatic controller. This extension, along with the associated processor and model assembly procedures, now allows us to capture a significant fraction of all metabolic reactions.

To evaluate the generalizability of the INDRA *Statement* representation and the procedures for extracting Statements from BioPAX and BEL, we queried Pathway Commons and the BEL large corpus for mechanisms in the neighborhoods of 15 genes and 5 metabolites across the categories of signaling, transcriptional regulation, and metabolism. For each neighborhood, we quantified the fraction of the original set of mechanisms captured as INDRA *Statements*. In the case of BEL we determined the fraction of BEL Statements (in the original neighborhood) for which a corresponding INDRA Statement could be extracted; in the case of Pathway Commons and BioPAX we determined the fraction of controlled BiochemicalReactions or TemplateReactions (which are used to represent transcriptional regulation) that could be extracted as INDRA *Statements*. Table 1, shown below,

shows coverage statistics for genes and metabolites involved in metabolism, both before (-) and after (+) the addition of the Conversion *Statement* class. We added a new Appendix Table 1 to the revised manuscript which shows coverage statistics for all 15 genes and 5 metabolites.

Gene / metabolite	BioPAX total	BioPAX extracted (-)	BioPAX % (-)	BioPAX total (+)	BioPAX extracted (+)	BEL total	BEL extracted (-)	BEL % (-)	BEL extracted (+)	BEL% (+)
GLUL	32	20	63%	27	84%	0	-	-	-	-
NOS1	31	21	68%	25	81%	5	3	60%	3	60%
IDH1	27	8	30%	17	63%	0	-	-	0	-
DHFR	33	14	42%	28	85%	0	-	-	-	-
PFKL	18	6	33%	13	72%	2	2	100%	2	100%
glutamine (CHEBI:28300)	11	0	0%	11	100%	0	-	-	-	-
β -D-fructofuranose 6-phosphate (CHEBI:16084)	13	0	0%	12	92%	0	-	-	-	-
5,6,7,8-tetrahydrofolic acid (CHEBI:20506)	15	0	0%	6	40%	0	-	-	-	-
pyruvic acid (CHEBI:32816)	82	0	0%	36	44%	0	-	-	-	-
nitric oxide (CHEBI:16480)	16	0	0%	8	50%	0	-	-	-	-

Table 1: Percentages of reactions or direct statements in pathway neighborhoods extracted by INDRA from Pathway Commons or the BEL Large Corpus, before addition of Conversion INDRA Statements (-) and after (+).

The results show that before the addition of Conversions, INDRA captured no reactions involving small molecule metabolites and a relatively smaller fraction of reactions involving metabolic and second-messenger enzymes. Further, as seen from Appendix Table 1 of the revised manuscript, after inclusion of conversions, Statement coverage is comparable across signaling, transcriptional regulation, and metabolism. This shows that with respect to the types of reactions encoded in widely-used databases, INDRA is fairly effective and general. As discussed in the revised manuscript, the template-based extraction procedures used by INDRA prioritize the precision of extraction over recall, which explains why the percent extraction is less than 100% in most cases. We continue to extend work on this issue so as to get better coverage of database-encoded mechanisms; however, this is complicated by the widely varying ways in which mechanisms are encoded by human curators across source databases (see 1.1, below).

In addition to adding extraction of Conversions from databases, we have also extended the TRIPS *Processor* to extract Conversions from natural language. For example, a user can now process the sentence "HK2 converts D-glucose and ATP into glucose-6-phosphate and ADP" and obtain a Conversion *Statement* containing the relevant chemical entities (grounded to their CHEBI identifiers). We thank the reviewers for pointing out these shortcomings in INDRA and believe that newly added functions address reviewers' concerns and substantially improve the manuscript.

GC5. Accessibility of the approach to non-experts

Reviewer 1 points out that the "completion or modification of models...requires basic expertise in systems dynamics/modeling"; Reviewer 2 also notes that "the authors sell it as it is the way so that also laymen can generate complex models, but I actually see it the other way around."

We agree with the reviewers that we have not collected empirical data on the usability of INDRA by naïve users. We have tested the system extensively on users in our own lab, but we accept that this is not the same thing and we have clarified the manuscript accordingly. As we discussed in GC1, a proper evaluation of modeling task efficiency across different user populations would involve substantial human subjects studies. Moreover, we are currently unaware of any evaluations of this kind that have been performed for other systems biology modeling software.

We acknowledge that even when the use of natural language makes model assumptions more transparent, use of natural language does not in and of itself address many of the other challenges in developing a meaningful dynamical model, including determination of parameter sensitivity, investigation of network dynamics, insight into combinatorial complexity, multistability, oscillations, etc. On the contrary: the use of natural language for modeling tends to *more directly expose* the ambiguities and unintended implications of common ways of describing biological mechanisms. Whether this is a strength (cf. Reviewer 3) or a weakness (cf. Reviewers 2 and 4) may depend on one's perspective; however, it does concretely raise the question of who the ideal user of a system such as INDRA might be, and what use cases it best supports.

As modelers ourselves, we have insight into the value of using INDRA for the following use cases:

1. *For an experienced modeler wishing to build multiple model types from the same set of assumptions.* Using natural language, a modeler can build a dynamical model, a Boolean network, or a coarse-grained pathway map from the same set of high-level assertions. We use INDRA in this way for our own work and are not aware of other modeling software providing this capability.

2. *For an experienced modeler to rapidly prototype small models.* In our own work, we have begun to use INDRA, rather than rule-based modeling or tools such as MATLAB to rapidly test hypotheses by putting together small models involving standard assumptions (e.g., one or two-step catalysis, independent binding sites, etc.). Also anecdotally, an experienced modeler (Dr. Matthew Peterson at MITRE Corporation) found it efficient to use INDRA and natural language to create the machine-readable reference set of mechanisms used to evaluate the performance of NLP for literature reading, as discussed in (Allen et al. 2015) (see GC1).

The use of INDRA for the following proposed use cases is more speculative:

3. *For interdisciplinary teams involving modeling experts and biological domain experts.* This is a use case discussed in the manuscript. In this setting, the modeler and domain experts would work together to articulate the model assumptions in natural language, with modelers subsequently exploring issues relating to combinatorial complexity and system behavior, which could then be refined through further dialog with the domain expert. Building the model in natural language ensures that the model assumptions remain transparent to biologists working in the domain. We are currently exploring precisely this possibility in collaboration with Neal Rosen, a biochemist and cancer biologist whose highly influential study of BRAF biology we analyze in the current paper.

4. *For modeling novices to build coarse-grained "pathway-map" style models through a simplified interface.* The reviewers note that building dynamical models raises issues that are difficult to resolve without a strong modeling background. However, constructing qualitative "pathway-map" style models (similar in scope and complexity to the Ras Pathway Map, for example) does not raise these issues. We have recently developed an initial prototype of a system that uses INDRA in the

form of a web service, and exposes a visual web interface to allow users to describe mechanisms in natural language and view results in the form of a pathway map that is automatically linked to UniProt, HGNC, and other resources (Figure 2). Although the results are presented as a simple network, the assembled rule-based model can also be downloaded for further exploration or refinement. A robust version of such an interface could be used to enable communities of biologists to collaboratively curate and share information on mechanisms, similar to the effort undertaken manually by Dr. Frank McCormick in the context of the NCI RAS Initiative.

Figure 2: Screenshots of a prototype web interface for natural language construction of pathway maps. On the left, sentences describing a simple MAP kinase cascade; on the right, simplified network representation of the underlying mechanisms. Colored regions of nodes indicate expression levels of RAS, RAF, MEK, and ERK family members in a particular melanoma cell line (A101D).

Finally, we note that the natural language modeling and assembly approach described here lays the necessary foundation for two important use cases outside the scope of the paper that we are currently pursuing, namely i) building models directly from the scientific literature, and ii) creating intelligent machine assistants for biomedical scientists (Carvunis and Ideker 2014).

DETAILED RESPONSES TO REVIEW

In the responses below, we have added numbering to each comment to allow cross-referencing, and we have added our responses to reviewers in blue font for clarity.

1. Reviewer 1 Major Comments

1.1 (i) *INDRA Statements are the key concept of the approach, serving as an intermediate representation (without additions or assumptions, claimed in the manuscript). The Statement templates and their structure appear, however, very tailor-made for certain aspects of cell signaling (e.g., the summary in Fig. S2 contains very specific classes such as RasGEF and RasGAP), and there is a risk that the design will not sufficiently generalize to other types of biological networks. Given standard ontologies and community standards for network representations (e.g., SBO and BioPAX), the definitions of templates require more justification and precision, in particular*

regarding the formal mapping between INDRA and standard classes (which should be coded in the Processor modules).

The reviewer makes two important points here. The first concerns the ability of the INDRA Statement types to capture mechanistic information across a broad variety of biological subject areas. We addressed this concern in the General Comments (**GC4**, above), showing that while INDRA does not capture all mechanisms present in pathway databases, the revised tool does not

Figure 3: Selected BioPAX BiochemicalReactions involving MAPK1/3 and MAP2K1 in Pathway Commons, displayed using the Chisio BioPAX Editor (ChIBE) (Babur et al. 2009).

appear to show a strong bias towards reactions involved in signaling.

The reviewer's second point concerns the formal mapping between INDRA and existing knowledge representation formats (e.g. BioPAX and SBO). To address this issue the revised manuscript includes an expansion of the INDRA Statement representation to include Systems Biology Ontology (SBO) annotation. These annotations apply not only to the *Statement* types (e.g., Phosphorylation, Activation, etc.) but also to *Statement* arguments (e.g., Kinase, Enzyme, etc.). We have integrated these links to SBO categories in the INDRA Statement JSON format (for example, see the "sbo" links in the example INDRA *Statement* graph in Figure 1). With this addition it is possible to make a principled correspondence between INDRA *Statements* and any other knowledge repositories that use SBO annotations.

A formal mapping between BioPAX and INDRA *Statement* classes is problematic because information in BioPAX comes in the form of chemical reactions that are not "typed"; that is, they do not contain high-level semantic information, for example that a reaction represents a phosphorylation modification. Instead, this must be inferred by examining the left- and right-hand sides of the reaction to determine what kind of reaction occurs, and involving what proteins. For example, Pathway Commons contains (among others) the five distinct reactions shown in Figure 3 above (also Appendix Figure S10 of the manuscript) describing the phosphorylation of ERK (MAPK1/3) by MAP2K1. The semantics of reaction 1 assert that a version of MAPK3 is on the left hand side, another version is on the right hand side, and MAP2K1 serves as the controller. It does not explicitly indicate that the reaction is a phosphorylation event; such information can only be determined by examining the pattern of changes occurring between the left and right hand sides.

The BioPAX developers created the BioPAX Patterns software package to specifically identify sets of reactions matching pattern-based criteria (Babur et al. 2014). The INDRA BioPAX *Processor* reuses the BioPAX patterns package to identify and process reactions matching pre-specified patterns corresponding to INDRA *Statements*. Once extracted into INDRA *Statements*, the (low-level) biochemical reactions in BioPAX acquire high-level semantic annotations (e.g., SBO annotations) that can be propagated into assembled models.

Reaction 5 in Figure 3 also represents a phosphorylation event, but it takes a substantially different form: here both MAP2K1 and MAPK1 are part of a complex together with ATP and ADP. This complex appears on the left-hand side and right-hand side of the reaction, and also serves as its controller. Due to the differences in structure between reactions, extraction patterns developed to match reaction 1 might not also extract reaction 5 as a phosphorylation event. The “best” patterns for extraction, in terms of precision and recall, depend on how reactions are curated or converted into BioPAX in practice; therefore, they must be developed and evaluated empirically. For this reason, we do not believe that a “formal mapping” between BioPAX/BEL and INDRA is feasible or useful.

The mapping between BEL statements, BioPAX reactions and INDRA *Statements* is implemented in the code for the INDRA processors. To make the criteria used for extraction more explicit, the revised manuscript extensively documents extraction functions for BioPAX and BEL. (see INDRA documentation in Appendix and as part of online documentation, e.g. http://indra.readthedocs.io/en/latest/modules/sources/bel/index.html#indra.sources.bel.processor.BelProcessor.get_modifications)

1.2 (ii) *Similarly, in terms of generalization and compatibility with existing systems, it is advisable to clarify relations to BioPAX and BEL, and give specific reasons for why automated extraction (instead of manual curation of statements after database searches already implemented in INDRA) fails or is limited. For example, statements such as p.25 'Perhaps unexpectedly, constructing executable models from pathway databases ...' appear to general and they may rely on outdated references.*

The challenge of assembling models directly from databases that we alluded to in the manuscript was not primarily one of extraction (as shown by Supplementary Table 1 of the revised manuscript, INDRA extractions into *Statements* shows good coverage of databases), but rather one of assembly. Consider the reactions shown in Figure 3 above, which co-exist in the Pathway Commons database. Clearly, these interactions cannot simultaneously be assembled into a single, coherent model. Hence, a human must either pick a specific *Statement* extracted from databases for inclusion in a model, or rely on knowledge acquired from databases to paraphrase relevant knowledge into natural language, as we have done in some of our case studies. To clarify this point, we have rephrased the last paragraph of the Discussion’s “Challenges in generating executable models from text” section (page 26 and 27). To emphasize, the issue here is not a limitation of INDRA but rather the wide variation in the representation of reactions in knowledge sources.

In particular, comments on how specific processors need to be, for different (alternative) inputs, and to what extent processors are extensible (in contrast to the need to develop specific processors for each potential data source) could clarify these points.

Processors are similar to each other in terms of their high-level approach: they query for patterns of template mechanisms and extract them as INDRA Statements. But due to the diversity of representations (e.g. biochemical reactions, semantic triples, event trees) and formats (e.g. RDF, JSON, XML) they need to be customized for each source. Compatibility with standardized exchange formats such as BioPAX and BEL ensure that INDRA can already draw on dozens of databases (Reactome, KEGG, Phosphosite, BioGRID, Panther, String, etc.) and NLP tools (TRIPS, REACH, and all tools that read into BEL (Rinaldi et al. 2016)). Existing and newly added support for widely used exchange standards and the modular architecture of INDRA (in which adding a new Processor module is straightforward) will allow us to support novel input sources in the future.

1.3 (iii) *Another aspect that warrant further discussion regards transparency of the INDRA system, especially concerning debugging. For example, how can users distinguish between cases of correct and incorrect processing of word models as in Fig. 4D? Does the INDRA system support textual or (ideally, as in Fig. 2 examples) graphical output at the statement level?*

This is an excellent point. As discussed in detail in GC3 above, we have now implemented transparently inspectable graphical, textual, and JSON representations for *Statements*. We have also revised and extended the "INDRA Statements represent mechanisms from multiple sources" section of the Results, added a new Supplementary Figure 2, and provided examples of inspecting *Statements* and PySB models in two Appendix iPython Notebooks.

At which levels are errors generated for word statements that cannot be processed, or can be processed only partially?

Errors in sentence processing are best identified by inspecting the *Statements* returned by the TRIPS Processor. If a sentence cannot be parsed, or the resulting TRIPS EKB structure does not match one of the TRIPS Processor extraction templates, no *Statements* will be returned for that sentence. Similarly, if a sentence is processed only partially, that will be apparent from inspection of the returned *Statements*. If the *Statements* yielded from text are satisfactory then the modeler can move on to assembly. We clarify this point at the end of the Results section "INDRA Statements represent mechanisms from multiple sources".

1.4 (iv) *On usability, the manuscript emphasizes ease of communication on models and limited technical challenges for their construction. However, for the p53 and MAPK examples, completion or modification of models to represent biological phenomena as discussed in the text requires basic expertise in systems dynamics / modeling (e.g., identification of missing reactions, association of positive feedback associated with time delay) that are not commonly found among 'traditional' biologists. In this view, statements such as p.16 'Introducing alternative assumptions and mechanisms using natural language is straightforward and can be accomplished by individuals with little or no technical expertise.' appear exaggerated and a more detailed discussion of technical requirements on the user for model implementation vs. analysis is warranted.*

This is a good point and is addressed in detail in GC5, above. In the revised manuscript, we have modified the section referred to by the reviewer (now on page 17-18) to read:

"The foregoing analysis of the Lahav and Purvis review illustrates several beneficial features of direct text to model conversion: (i) the possibility of identifying subtle gaps and deficiencies in word models with the potential to profoundly affect network dynamics and

function; (ii) the ability to maintain precise congruence between verbal, pictorial and computational representations of a network; and (iii) a reminder to include neglected negative regulatory mechanisms when explaining network dynamics. We propose that future figures of this type include accompanying declarative text (precisely stated word models) on the basis of which graphs and dynamical models can be created. We have found that it is remarkably informative to experiment with language and then render it in computational form: it was this type of experimentation that led us to rediscover for ourselves the importance of negative regulation and nonlinear positive feedback in generating p53 oscillations.”

1.5 (v) *Finally, among the examples of INDRA usage, the example of Ras signaling should be expanded with additional evidence. A 'visually comparable' pathway map represents as a rather weak argument, given that Ras signaling was used more rigorously in the original DRUM publication (Allen, 2015); it could also be revealing to compare INDRA output (formally) with standardized pathway maps, such as those published in SBGN notation for several mammalian signaling pathways. In addition, while the Boolean simulation for Ras demonstrates a key concept of INDRA (assembly into models of different type), the discussed simulation represent a very simple case study that could be omitted.*

To clarify, the pathway model created from natural language by INDRA as shown in Figure 7 is structurally *identical* to the original diagram. In particular, the INDRA-assembled pathway map (i) includes the same set of proteins, (ii) represents the same set of interactions among these proteins and (iii) recapitulates the semantics and level of mechanistic detail of the original diagram in that interactions are represented as directed positive and negative edges or undirected edges indicating complex formation. To state this explicitly, we have extended the corresponding paragraph on page 23 of the revised manuscript.

By “visually comparable,” we were referring to the ability of the graph layout program used in conjunction with INDRA assembly (GraphViz) to produce an interpretable visualization of the model contents. In principle, the automated layout could have produced a result that was not amenable to human interpretation, leading to the conclusion that from a visualization perspective, a model assembled from natural language could not substitute for a hand-drawn pathway map. In contrast, we found that the pathway graph assembled and displayed automatically was “visually comparable” to the original model in terms of layout and readability. We have clarified this point on page 23 of the revised manuscript.

Published research around Ras signaling was used in the original TRIPS/DRUM publication (Allen et al. 2015) as an initial use case to focus evaluations event extractions from the *literature*. In the model presented in Figure 7 of this manuscript (full text given in Appendix Section 2.5), *every sentence* used to define the Ras pathway map was correctly processed by TRIPS. This means that for this model description, as expected, superior precision is achieved as compared to results in Allen et al. (2015) for event extraction from literature.

We appreciate the reviewer’s critique of the Boolean network example, but we have decided to keep it in the manuscript in order to emphasize that INDRA can be used to assemble multiple model types from a single set of assertions. We consider this one of its key features, and the examples take only a small amount of space. We argue that the simulation also demonstrates the

fact that the resulting model is amenable to principled computational analysis – even if a simple one.

Minor comments:

1.6 (i) Title, abstract and general: *The term 'word models' is not defined until late in the introduction; a definition in the abstract would be helpful.*

We have modified the first sentence of the abstract to read “Word models (natural language descriptions of molecular mechanisms) are a common currency in spoken and written communication in biomedicine ...”. This edit fits within the 175-word limit.

1.7 (ii) Legend to Fig. 1C: *'... creates [an] Extraction Knowledge ...'.*

Thank you, we fixed this typo in the legend of Figure 1C.

1.8 (iii) The introduction provides verbal arguments on the conceptual differences between INDRA and existing systems, which are well summarized in Fig. 8 and associated text. To better convey the concepts behind INDRA in a non-technical way it may be advisable to move this figure and (parts of the) text to the introduction.

We carefully considered this suggestion but feel that Figure 8 requires a broader discussion of existing approaches (mathematical modeling, rule-based modeling) and we therefore think that it is better positioned in the Discussion. We appreciate that this is a matter of style, however.

1.9 (iv) More recent developments towards community standards in systems biology should be referenced, for example, for whole-cell modeling (IEEE TRANSACTIONS ON BIOMEDICAL ENGINEERING, VOL. 63, NO. 10, OCTOBER 2016).

This is an excellent point. The ability of INDRA to assemble annotated models from knowledge sources benefits heavily from the existence of community standards and curated resources. We have added a new paragraph to the “Relationship to previous work” section of the Discussion on page 28 of the revised manuscript, in which we refer to the many standards INDRA makes use of, including the reference suggested by the reviewer.

Reviewer 2:

2.1 *As far as I see the approach seems to work (though there is no benchmark by which one can really measure this), but the individual steps are not really compared to the state of the art. When does the natural language processing fail? How complicated can these texts be?*

These important points are addressed in detail in GC1, above. There we refer to the formal evaluation of the TRIPS/DRUM system in comparison with other NLP systems for event extraction from the literature (Allen et al. 2015).

2.2 *My main concern is that I don't really see that the approach is a major step forward in making signaling models.*

The authors sell it as it is the way so that also laymen can generate complex models, but I actually see it the other way around. The models are generated basically by a black box, and very similar statements can lead to very different models, especially given the combinatorial complexity, and the authors show one example of such combinatorial explosion.

As discussed in detail in GC5, above, we have reframed the likely use cases of the system and now explicitly state that we consider the “layperson modeler” use case to be more speculative (“Limitations and future extensions of INDRA” on page 29 and 30 of the revised manuscript). We discuss the issue of sensitivity of models to wording in GC2 and transparency of the internal processing steps in GC3. In particular, with the revisions made as part of this resubmission, it is now possible to examine the extraction and assembly steps in INDRA in detail. Our approach is most definitely not intended to be a “black box” and we apologize for not making this point clearly enough.

2.3 *The major advantage of computational models is (at least that is my view of it) that these are very explicit about the modeling assumption. This benefit is gone with this approach, so that other (more graphical ways) would be more explicit, such as SBGN editors or other editors.*

While graphical interfaces can be intuitive, like other low-level modeling formalisms, such interfaces convolve biological knowledge with reaction implementation. As discussed in detail in GC2, we believe that the approach described here is transparent because the knowledge-level assertions and associated assembly policies are distinct and explicit, rather than implicit and embedded in the final model (see also Figure 8B in the manuscript). In addition, both the intermediate stages (INDRA Statements, PySB rules) and concrete implementation resulting from assembly (species and reactions, SBML, SBGN, etc.) can be inspected as with any other model (see GC3, above).

2.4 *The other application where I could see this approach being useful would be to extract crude models from full text. However, for this the text mining seems to be rather limited in potential.*

As noted in GC1, processing full text articles is *possible* with the current system in place. Even now, INDRA *Statements* extracted from the literature can be used in models (an experimental prototype of such a system built on INDRA is available online at <https://twitter.com/therasmachine> and <https://tinyurl.com/yaayuoy3>) However, the challenge in assembling fragments in the literature into coherent executable models *automatically* involves new algorithms that go far beyond the scope of this paper. Some of the outstanding challenges involve resolving full and partial redundancies and conflicts. While NLP from the scientific literature is challenging, the bottleneck is not NLP itself but rather the imprecise and overlapping ways we describe biochemical mechanisms, which results in an assembly challenge. The NLP tools used by INDRA have been validated on full scientific literature text (Allen et al. 2015; Valenzuela-Escarcega et al. 2015).

Reviewer 3:

Major comments:

3.1 *1. Through the exercise of deciphering word models, the study highlights the crux of model building: the encoding of imprecise or often vague ideas about biological mechanism into unambiguous mathematical form. For me, the greatest take-away from the paper was that, even*

with a tool that understands spoken language, building a molecular model is an iterative dialogue between human and computer in which common ground is achieved by balancing model purpose with the requirement for discrete structures. This principle becomes clear through all three pathway examples, which each required some degree of expert knowledge in order to achieve a specific modeling purpose.

We appreciate this comment and agree with the reviewer's interpretation of the relationship between model definition at the level of knowledge and model implementation at the level of low-level executable forms (e.g. mathematical equations).

3.2 2. *It seems that INDRA ought to be able to handle assembly of statements in which two mechanisms are embedded. For example, we might say that "EGFR activates SOS" and that "EGFR activates SHC". Both statements are true. The first statement is correct but has not defined the intermediate (SHC) that mediates the activation of SOS by EGFR. Similarly, "MEK1 phosphorylates ERK2" and "MEK1 phosphorylates ERK2 at threonine 185" should be merged into a common mechanism that reduces to the second statement. Does INDRA allow for these types of overlapping word models to be supplied and reconciled?*

This is an important feature and would indeed be required for the *unsupervised*, automated assembly of models from primary sources (literature and databases)—a use case distinct from what we present in this paper. As mentioned above, INDRA has a prototype module that resolves overlaps such as the one provided by the reviewer (see INDRA documentation at <http://indra.readthedocs.io/en/latest/modules/preassembler/index.html#module-indra.preassembler>). This capability fits into a larger framework of assembly features currently under development that are beyond the scope of this work. In the current manuscript, we address only use cases where the modeler is in full control of the *content* of the model and is responsible for providing non-conflicting and non-redundant mechanisms. However, the reviewer is correct in asserting that a generalized, automated text-to-model system will require the ability to handle overlapping mechanisms.

3.3 3. *Supposing INDRA may be applied to large collections of natural language, how would it handle apparently contradictory information (e.g., X activates Z; X inhibits Z)". One could imagine a scenario in which both statements are true (e.g., an incoherent feedforward loop in which X activates Z directly but inhibits Z through intermediate Y). It may be that this capability of assembling language into an executable model is beyond the scope of the current implementation of INDRA, which is aimed primarily at providing a modeling interface that begins with natural language. However, the capability of INDRA to establish equivalence between multiple aliases of the same biological entities (described on p. 16) leads me to believe that it is intended (at least eventually) for application to large databases of mechanistic information in which contradictory mechanisms are almost certain to exist. I would like to see a more precise description of the context/limitations/eventual utility of INDRA.*

Like the reviewer, we also anticipate that with additional development, INDRA could be used to automatically construct models from large databases and the literature. Identifying and resolving conflicts, particularly when the conflicts may have resulted from curation or NLP errors, is one of the many assembly issues that arise in this process.

Such conflicts could be resolved in multiple ways. First, conflicts can often be resolved if more *context* is available (e.g. X activates Z in the nucleus vs. X inhibits Z in the cytoplasm). Because INDRA Statements carry contextual information (if available from a source) such conflict resolution may be possible. Second, it may be possible to determine that one of the conflicting statements has substantially more evidentiary support than the other, allowing the conflict to be resolved in its favor *a priori*. Since Statements are collected from multiple sources with different levels of reliability it could be possible to determine the corroboration of one Statement by another thereby providing information on the reliability of each mechanism overall (see INDRA documentation at <http://indra.readthedocs.io/en/latest/modules/belief/index.html> for an early prototype of such an approach). Third, in some cases it may be possible to include both conflicting Statements in a model and then condition the model on a particular dataset to determine which mechanisms are more likely to be true or relevant *a posteriori*.

In the previous version of the Discussion we mentioned that “Elsewhere we will describe progress on the task of extracting pathway information from the literature, which presents challenges not only for NLP but also for assembly (due to the large amount of irrelevant, redundant, overlapping, and erroneous information returned).” We have significantly extended this section to now read (page 30):

“The TRIPS system (as well as other NLP systems we tried, such as REACH) can be used to process the more complex and ambiguous language used in scientific publications and they are both state of the art systems with different strengths and weaknesses. Empirical results presented in (Allen et al, 2015) show that TRIPS compares favorably in precision and recall to ten other NLP systems on an event extraction task from biomedical publications, and reaches precision and recall levels close to those produced by human curators. While reading from the biomedical literature is less robust as compared to reading the declarative language used in this paper, the fundamental challenge in generating models directly from literature information is not reading but knowledge assembly. The assembly challenge involves multiple interconnected issues, including: (i) the large amount of full and partial redundancy of knowledge generated when mechanisms are read at scale (e.g. MEK phosphorylates ERK vs. MEK1 phosphorylates ERK); (ii) inconsistencies between knowledge collected from multiple sources which may or may not be resolvable based on context; (iii) the distinction between direct physical interactions and indirect effects; and (iv) technical errors such as erroneous entity disambiguation and normalization. In the approach described here, human experts simplify machine reading and assembly by paraphrasing statements about mechanisms into simplified, declarative language. As illustrated in the POMI models of p53 dynamics, the use of simplified language is not only useful for machines, it helps to clarify complex issues for humans as well. However, we are actively working to extend INDRA so it can assemble information from the primary scientific literature into coherent models.”

3.4 4. *An unexpected consequence of INDRA is that it may serve a reverse and complementary function: to improve the precision of how we speak about biological mechanisms. The example of the problematic phrase "ATM activates itself" highlights how a tool like INDRA could be useful in this regard.*

This is a good point and we agree that INDRA provides a principled way to test whether a mechanism, as described in natural language, corresponds to a model with the intended behavior.

This can be used to improve the wording of scientific claims and ensure that what is *said* is congruent with what is *meant*.

Minor comments:

3.5 1. *The statement "...there is little evidence that creation of new word models is being routinely supplemented by formal modeling approaches" (p. 2) is unclear at this point in the manuscript. I think it becomes clear in the Results section that Assembly can be of great benefit to improving the precision of word models and to refining and challenging existing statements about biological mechanism.*

We have edited the last two sentences of the first paragraph of the introduction to read: "The challenge arises in linking a rich ecology of word models to computational representations of these models that can be simulated and analyzed. The technical environments used to create and explore dynamical models remain unfamiliar to many biologists and a substantial gap persists between the bulk of the literature and formal systems biology models." This edit is intended to emphasize the issue that word models are currently created without a sense of their formal implications, rather than suggesting that word models should be exclusively supplanted by formal models.

3.6 2. *The acronym SBML is used before it is defined (p. 2).*

We now spell out SBML explicitly on page 2.

3.7 3. *The acronym UML is used before it is defined (p. 8).*

We added the definition of UML on page 8.

3.8 4. *EKB is defined in the glossary on p. 5 but then redefined on pp. 8 and 10.*

In the case of page 8, we believe spelling out "extraction knowledge base" again helps readability. We removed the redefinition on page 10. We are happy to be guided by the journal style on this point.

3.9 5. *In general, the paper is very acronym-rich, but I don't see any way around this.*

We hope that the glossary helps with this problem; we can further expand the number of terms in it, if the editors approve.

3.10 6. *It would be useful for many readers to have a reference associated with the sentence that ends "...reducing the identifiability of the model." (p. 14).*

We added a reference to (Raue et al. 2009) which gives a good overview of structural and practical identifiability in dynamical systems biology models.

Reviewer 4:

4.1 *General remarks*

- Are you convinced of the key conclusions?

The authors successfully demonstrate that executable models can reliably generated from natural language statements using their newly developed tool. They also successfully show -qualitatively- that expected behavior can be recapitulated by models generated from natural language.

The reliability of the approach and potential of natural language models in enabling collaboration is somewhat convincing. However, the examples in the study failed to produce expected results with initial natural language descriptions and iterative adjusting of phraseology and re-running the pipeline was required to obtain the desired result. The sensitivity of the approach to specific wording limits the use of the technique to those with some knowledge of the conversion process as those unfamiliar with underlying data structures will quickly resort to a trial and error approach to rewording the descriptions.

The issue of the sensitivity of the final model to wording is discussed in detail in GC2. In fact we show that INDRA is not particularly sensitive to alternative wording except when the alternatives—however subtle—refer to fundamentally different mechanisms (e.g. cis and trans phosphorylation). “*Iterative adjusting of phraseology*” was not necessary to obtain a desired result beyond the mechanistically significant variants we discuss in the text. We believe that the reviewer has misunderstood our point with respect to alternative ways of describing a mechanism verbally and apologize for our lack of clarity. We have substantially rewritten the relevant sections of the manuscript in an attempt to address this misunderstanding (pages 17-18).

The example of the p53 model was intended to highlight the fact that informal descriptions that are regularly used by biologists (e.g., the p53 diagram drawn from (Jeremy E. Purvis and Lahav 2013)) are often incomplete and, when taken literally, cannot reproduce the dynamic behavior they are meant to explain. The fact that the initial natural language description “failed to produce expected results” reflects on the limited content of the diagram, not on the software or approach. The process of improving the language was included in the manuscript precisely to show the series of steps required to identify and clarify, in natural language, the mechanistic gaps *in the original diagram*. This is a feature rather than a bug of our system.

To further address this issue and make the conversion process more transparent we have added features to INDRA that allow *Statements* extracted from sentences to be inspected in a variety of formats, and have provided examples in the form of two Appendix iPython Notebooks.

4.2 Inevitable combinatorial complexity (acknowledged by the authors), combined with the author's choice of reaction scheme (two-step), generated models that were unnecessarily complex. This complexity hampers further modeling and parameterization required to make predictive models and would also be a barrier to the use of the described technique as a collaborative tool.

Combinatorial complexity resulting from natural language descriptions is addressed in GC2, above. In the INDRA approach, models will be as complex as implied by the description given by the user. Typically, providing stricter context on mechanisms (by adding more assumptions) will result in reduced combinatorial complexity. An example is “MEK binds ERK” versus “MEK not bound to PP2A binds ERK”. Here the former implies the existence of more individual species and biochemical reactions but has fewer assumptions compared to the latter. The number of parameters scales with the number of rules (which is proportional with the number of INDRA *Statements*) and not the

number of biochemical reactions and species. Therefore, while combinatorial complexity tends to make simulation more computationally expensive, it does not in itself make parameterization more challenging (as an example, consider a simple polymerization model with an infinite number of species and reactions but only a single reaction rate parameter).

To address the reviewer's concern, we have expanded on the number of reaction policies to include more sophisticated one-step mechanisms that tend to mitigate the problem of combinatorial complexity.

4.3 - Place the work in its context.

Most progress in the area of enabling technologies/software for computational modeling has been focused on software environments for the construction of pathway diagrams and the creation of models from these diagrams (Cell Designer, COPASI, SimBiology). Diagrammatic representations have been widely and successfully used as tools for collaboration between biological and computational experts.

This work is not intended to undermine the value of graphical standards and tools in systems biology modeling. Rather, it aims to highlight a distinct approach in which knowledge-level statements (which generally take the form of natural language) are decoupled from a particular formal implementation, whether in graphical or mathematical form. Indeed, the revised INDRA software is integrated with graphical formats and interfaces (SBGN, GraphViz, CyJS, NDEx). We emphasize this now on page 29 of the Discussion:

"By uncoupling knowledge-level statements from a particular formal implementation, whether graphical or mathematical, natural language modeling is complementary to and compatible with a wide variety of input and output formats. In the case of INDRA, an intermediate representation enables a wide variety of many-to-many conversions involving text, BioPAX, BEL, PySB, BNGL, SBML, ODEs, logical models and graph-based formats such as SBGN (an INDRA-assembled SBGN graph of the model presented in Figure 5C is shown in Supplementary Figure S9). Further integration of natural language and graphical modeling, for example by coupling INDRA to SBGNViz graphical interface (Sari et al, 2015), will improve the quality of human-machine interaction and further facilitate model assembly and exploration."

We provide INDRA as a library that can be easily imported and built upon in existing (graphical) tools, and we have also implemented a REST API that can be run and called locally, making the natural language modeling features of INDRA usable from non-Python environments (we have highlighted this new feature in the "Software and model availability" section of Materials and Methods). Thus, we believe that natural language and graphical modeling are natural allies and not alternatives, and we provide a basis for integration.

4.4 *The approach described here provides an alternative to diagrammatic formalisms. This study represents an important component in a long-term goal of systems biology, which is mining biological literature in a manner amenable to theoretical analysis. When combined with automated literature text-mining and existing parameter estimation techniques the work presented here will presumably provide an important component of a complete automated model construction pipeline able to construct executable models for any biological domain.*

We appreciate this comment and agree with the reviewer that our work addresses part of the challenge of automated modeling from literature. Although not complete, our preliminary results from ongoing developments indicate that the architecture and conceptual approach of INDRA will be extensible to this grander task.

4.5 - What is the nature of the advance (conceptual, technical, clinical)?

The work describes a technical advance in converting parsed text statements into executable modeling.

4.6 - How significant is the advance compared to previous knowledge?

In isolation in its current form the study represents a moderate advance, however if issues of complexity explosion and sensitivity to wording are improved then the tool presented here will be an important advance enabling widely-useful, automated, model construction and analysis from scientific literature.

We address the question of combinatorial complexity and sensitivity to language above.

4.7 - What audience will be interested in this study?

The study will be of interest to systems biologists working on model construction and dynamical pathway analysis.

Major points

4.8 - Specific criticisms related to key conclusions

The sensitivity to specific wording generating qualitatively distinct behavior (demonstrated by the ATM trans-phosphorylation example) seems to be a significant limitation in the applicability of the approach. While the study claims to enable model construction by individuals with little or no technical expertise, significant knowledge of which phrasing changes can generate significantly different dynamical systems is required to successfully use the technique.

We have addressed sensitivity to wording in detail in GC2 above, and as described in GC5, have edited the manuscript to de-emphasize the use of INDRA for the construction of dynamical models by non-experts ("Limitations and future extensions of INDRA", page 29 and 30). We hope that these changes address this important issue.

4.9 *Most successful model-driven studies utilize elegant models containing ~1 reaction for each biochemically significant process. The "word model" format here is amenable to such elegant model construction, however, in its current implementation that is not achieved.*

There have been a number of successful modeling studies in systems biology that have made use of combinatorially complex models, typically for studies of signaling (Deeds et al. 2012; Suderman and Deeds 2013; Hlavacek et al. 2003; Sneddon, Faeder, and Emonet 2011; Chen, Niepel, and Sorger 2010). However, it is true that combinatorially complex models are less widely used than their simpler counterparts and represent the most straightforward modeling use case.

Construction of models containing one reaction for each biochemically significant process is in fact currently achievable using INDRA, if that is the modeling goal, by using the “one-step” policy during assembly. This was the approach taken in the p53 example described in the manuscript, which did not result in a combinatorially complex model. To further support this use case, we have also implemented a one-step policy with a Michaelis-Menten rate law as suggested by the reviewer (see answer to 4.13 below).

In models involving binding or multiple sites of post-translational modification, eliminating combinatorial complexity requires making interactions specific to a single species by adding additional assumptions in the form of natural language (a simple example involving MEK and ERK is given above in the answer to 4.2; other examples of using natural language to clarify context are shown in Appendix iPython Notebook 2 and in the BRAF model in Figure 6 of the manuscript). Reducing combinatorial complexity in models of biological systems that are inherently combinatorial (e.g., signalosome assembly, multisite post-translational modification, polymerization, etc.) requires that these assumptions be made, regardless of the modeling formalism used—there is no free lunch in this respect. In the natural language modeling approach these assumptions are declared explicitly as part of the high-level mechanistic assumptions, rather than being incorporated implicitly in the final model (see GC2 for further discussion).

4.10 *The authors argue, somewhat unconvincingly, that natural language provides a superior collaborative framework to traditional wiring diagrams. Many of the weaknesses of diagrammatic representations are shared by natural language descriptions of processes. Vague wording inhibit quantitative analysis in the same way vague diagrammatic representations do. Large lists of plain-text statements become difficult to understand with scale in the same way diagrammatic representations do.*

In the original manuscript, our only comment regarding graphical approaches concerned the challenges associated with modeling combinatorial complexity that is typical of signaling systems. For such systems, a verbal description can be relatively compact whereas a graphical representation that involves the representation of specific chemical species will quickly become unwieldy. For large systems in which the correspondence between sentences and reactions is closer to one-to-one (e.g., large metabolic models), it is indeed possible that curation of a large body of natural language text could be onerous.

However, our more general goal is distinguishing between curated knowledge about mechanisms and specific executable implementations. We have implemented a system in which mechanistic knowledge is curated in natural language, but one could develop an analogous system in which the knowledge-level assertions were curated graphically. In fact, the authors of the mEPN graphical pathway modeling language identified precisely the same drawback in pre-existing graphical formalisms for systems biology, noting that “currently, pathway depiction and pathway modelling are generally considered to be separate disciplines.” (O’Hara et al. 2016). The approach of mEPN differs from INDRA in other respects (it does not incorporate a separate assembly step, instead supporting direct simulation of the pathway diagram itself) but it is aimed at the same conceptual problem. As described in the answer to 4.3, it is not difficult to imagine a multimodal knowledge curation system incorporating both natural language and diagrams that could subsequently be simulated.

4.11 *The authors even use an "informal" diagram to construct a word model, a step that requires similar effort and expertise as converting an "informal" diagram into a mathematically complete notation such as SBGN in widely used software.*

This comment may be the result of a misunderstanding. In both the p53 and Ras pathway examples, we drew mechanistic knowledge from a previously published "informal diagram" and the accompanying review article. These diagrams were manually created by experts other than us in a form that was not directly amenable to computation.

The purpose of these examples was *not* to show that natural language is the most efficient way to build a computable model from such a diagram. Instead, we aimed to explore the possibility that, in the context of a system such as INDRA the original curators of these diagrams *could have used word models to curate the same knowledge originally*. By doing so, they would have ended up with 1) a description of the knowledge-level mechanistic assertions that would be transparent to any biologist, 2) one or more executable models based on those assertions (Figures 5, 6 and 7), and 3) an accompanying diagram automatically generated from those assertions (Figure 7).

While it is true that the original authors could have curated the p53 diagram and Ras pathway map in SBGN, we believe that a model definition in English language has many advantages: it is highly intuitive, can be viewed and edited without specialized software, and does not require learning a specialized syntax. Though we acknowledge that the possibility that such a system could be widely used by biologists unfamiliar with modeling is speculative, we believe that for certain qualitative modeling applications this is achievable (see also GC5).

4.12 *-Specify experiments or analyses required to demonstrate the conclusions*

Sensitivity to wording should be addressed. Rather than allowing "ATM trans-phosphorylation" to silently generate significantly different models to "Active ATM phosphorylates ATM" a step in which ambiguities such as these are identified and clarified when INDRA is run would greatly improve the approach.

As discussed in GC2 and GC3, we have provided additional tools and examples showing how the different Statements (and subsequently models) yielded by these sentences can be evaluated by the user before proceeding to model analysis. We have extended the corresponding text on page 17 of the manuscript to highlight the fact that the cis- vs. trans- phosphorylation was interesting not because it reflected an unexpected result from TRIPS or INDRA, but because of the impact these two distinct mechanisms had on the dynamics. We believe that this in combination with multiple revisions to the text address the reviewer's concern.

4.13 *Combinatorial explosion must be addressed. Michaelis-Menten kinetics should be included in addition to the 1- and 2-process mechanisms to enable construction of elegant models that include saturating kinetics.*

Combinatorial complexity is addressed in GC2, 4.2, and 4.9. We agree with the Reviewer that the lack of support for Michaelis-Menten kinetics represented an important limitation and have incorporated it into the latest release of INDRA and into our revised manuscript. Assembly with Michaelis-Menten kinetics is now a featured policy in Figure 4, and an example of assembly using this policy is given in Appendix iPython Notebook 2. We have also implemented a Hill equation-

based policy for RegulateAmount INDRA Statements to allow for the straightforward modeling of saturable transcriptional activation.

4.14 *If the number of generated ODEs is similar to the number of phrases used as input the technique will be greatly improved.*

We agree and argue that this is possible with INDRA as discussed in 4.9 above.

4.15 *The study's focus on replacing diagrammatic representations should be adjusted to instead focus on the important role of INDRA in forming part of a highly enabling, automated pipeline from literature to automatically generated and parameterized models. Where appropriate the specific sections of the manuscript of most relevance to answering these concerns are highlighted below.*

We agree; as discussed in 4.3 and 4.10, it was not our intention to imply that natural language would replace formal diagrammatic representations, but rather that it represents a highly complementary approach. We have updated the corresponding section on page 28 and 29 under "Relationship to Previous work" in the revised manuscript.

4.16 *Minor points*

-Easily addressable points

Introduction.

Paragraph 1:

Include modeling of ERK, P53, NFkB and other signaling networks into examples of mechanisms elucidated by computational modeling.

In the first paragraph of introduction we now cite (Hoffmann et al. 2002) (NFkB), (J. E. Purvis et al. 2012) (P53 / DNA damage response), and (Chen et al. 2009) (ERK) as further examples of mechanisms elucidated by computational modeling.

4.17 *Is it possible to generate some estimate of how many papers contain formal language of high enough quality to enable "word model" construction and don't contain a diagram in a formal notation enabling model construction.*

To our knowledge, the most comprehensive database of dynamical models is the BioModels database. As of June 2017 there were 640 curated and 995 non-curated models in the BioModels database with 15 curated models annotated as having been published in 2016. In contrast, 140 articles were annotated in PubMed as being mechanistically relevant to a single gene, BRAF, in 2016 alone (we note that these annotations, while containing no false positives, are only a lower bound on the number of publications relevant to BRAF). This seems to suggest that the vast majority of publications that describe molecular mechanisms do not contain a formal modeling component. Our preliminary results show that the literature is rich in declarative statements of the form used in word models. However, statements in the literature, taken globally, are redundant, conflicting and apply in distinct contexts which are not necessarily relevant to a particular model. As we argue in GC1 above, while natural language processing tools (Allen et al. 2015; Valenzuela-Escarcega et al. 2015) connected with INDRA can already extract Statements from the literature, turning the collection of these Statements into coherent and causally sound models is a challenging assembly problem that goes beyond our current work.

4.18 Paragraph 2:

Add Copasi (Hoops et al 2006) and BioModels database (Juty et al. 2015) to list of tools for improving mechanistic modeling.

We included both references suggested by the reviewer into the second paragraph of the Introduction of the revised manuscript.

4.19 Paragraph 5:

Natural language suffers the same issues of differences in style/syntax between authors as differences in diagrammatic notation. There are many phrases that could represent the same arrow on a diagram. While paragraph 3 dismisses diagrams as struggling to deal with large complexity and large scope, how natural language descriptions overcomes this is unsatisfactorily described. "sophisticated NLP algorithms" is vague phraseology that should be replaced with a better description of the algorithmic innovation or novel application of existing specified algorithms.

We agree that natural language varies in style and syntax, but we argue that the TRIPS system effectively normalizes variability in language with the *same underlying meaning* into a single logical form (see Figure 5D, Box 1, Supplementary Methods 2.1, and Allen et al. (2015)).

We have removed "sophisticated" from the sentence referred to by the reviewer and added more detail to state which specific concept in the TRIPS system handles variability in style and syntax. The revised sentence now reads: "...INDRA can accommodate flexibility in style and syntax through the use of NLP algorithms that normalize variability in expression into logical forms that effectively represent the underlying meaning (Box 1)."

On the note of diagrams, our intention in paragraph 3 was not to "dismiss" diagrams but to point out one of their important limitations in modeling signaling pathways, which we believe is justified.

4.20 Results

Page 13:

The "one-step policy" in which an enzyme-mediated reaction is represented by a single process does not necessarily ignore enzyme saturation and require only a single parameter, as this single step process could be represented with Michaelis-Menten (MM) kinetics, which is supported by PySB and uses 2 parameters. The one step policy is valid in excess enzyme regimes. The one step policy with MM kinetics extends the one-step policy to include enzyme saturating regimes. This is the most widely used formalism in model building and is currently ignored. The 2-step policy is only required if MM assumptions are not met. The addition of MM kinetics may prevent a model with 28 PySB rules exploding into a model with 99 differential equations, and a model with 34 rules exploding into 275 ODES, while still capturing important kinetic characteristics.

We agree with the reviewer on all points regarding the use of each type of kinetics. As mentioned above, we have implemented a one-step Michaelis-Menten policy and featured it in Figure 4 of the revised manuscript as well as Appendix iPython Notebook 2.

4.21 Page 15:

In reference to the constitutive negative regulation of Mdm2 and Wip1 clarity should be provided as to which parameters in the ODE model of Batchelor et al, 2011 were missing from the diagram. I

am unable to locate p14ARF or HIPK2 inhibition in the paper, supplement or MATLAB files of Batchelor et al. If other studies were used for the addition of this mechanism then that should be clarified as it seems the model introduced here required additional mechanism to recreate the same behavior seen in Batchelor et al. 2011.

As described above in 4.11, the purpose of the p53 example was not to fully recapitulate the formal model in (Batchelor et al. 2011), but rather to highlight the dual value of natural language as a medium for communication about mechanisms and (automated) construction of formal models. As we describe in the text, the negative regulation of Mdm2 and Wip1 are missing from the original diagram. We have added p14ARF and HIPK2 ourselves, as known negative regulators of Mdm2 and Wip1 (citations where these mechanisms are described in detail are given in the manuscript). In fact, these negative regulations *are* modeled in Batchelor et al. (2011) but are only defined up to a generic negative term in the corresponding ODEs and not identified mechanistically. The reviewer is therefore correct that p14ARF and HIPK2 are not present in the Batchelor et al model.

4.22 *Time delay is added using a positive feedback however this limits the applicability of the approach as positive feedback does not exist in all systems with delay. INDRA should either be extended to accept word models that include "X activates Y with a delay", or this limitation should be explicit.*

Our approach to modeling mechanisms (positive feedback or otherwise) behind time delay is biochemically explicit and "with a delay" will not directly be translatable into an underlying mechanism. INDRA is aimed to explore the properties of linked sets of biochemical mechanisms, drawn from high-level assertions, and is not capable of assembling mechanistically undefined phenomenological kinetic properties such as "with a delay".

4.23 *"Oscillation was robust to changes in kinetic parameters" may be confusing as properties such as amplitude and frequency are highly parameter sensitive. Suggest: "The presence of oscillations was robust to changes".*

We agree, and we changed the sentence of page 16 of the revised manuscript as the reviewer suggests.

4.24 *While it is clear that essential reactions for the dynamical system were omitted from diagrams they were also omitted from the text and had to be manually added. This is not an argument for word-models over diagrammatic representations as both were insufficient from a single study. Please clarify.*

In the case study presented in Figure 5 of the Results, we show that word models can be turned into mathematical models with INDRA, allowing hypotheses, as described verbally, to be rigorously evaluated. Indeed, informal diagrams alone and informal text alone aren't enough to rigorously evaluate whether a description of a mechanism is "valid" and sufficient to reproduce a given behavior. However, once assembled into a model by INDRA (see Figure 5 and associated text), it was immediately clear that some necessary mechanisms were not made explicit in the original diagram. We therefore added these mechanisms as additional sentences and were able to show that they were sufficient to produce an oscillatory model. The main argument here is the use of models (in this case automatically derived from text by INDRA) over non-formalized descriptions.

4.25 *The text describes the manual construction of word models from diagrams and other studies and then states that "machine-assembled" word models are useful. Please clarify that these are manually constructed word models that are then machine-assembled into an executable form.*

We changed the corresponding paragraph and the sentence now reads "By converting word models directly into executable computational models, we ensure that verbal descriptions and dynamical simulations are congruent." on page 16 of the revised manuscript.

4.26 *Page 16:*

Biologically descriptive phrases such as ATM trans-phosphorylates itself should be added to the tested phrases here.

To make the model in Figure 5E more clear and to emphasize the mechanistic (rather than linguistic) distinction between the cis- and trans-phosphorylation models, we rephrased the first sentence of the model shown in Figure 5E to "Active ATM phosphorylates another ATM molecule". We also tested the sentence "ATM trans-phosphorylates itself", which yields the expected trans-phosphorylation mechanism. The corresponding paragraph (page 17) in the manuscript now reads:

"By adding and removing different aspects of the underlying mechanism using natural language we observed that including the mechanism "Active ATM phosphorylates another ATM molecule" was essential for oscillation; the phrase "ATM phosphorylates itself" generated a valid set of reactions but did not create oscillations for any of the parameter values we sampled. The difference is that "Active ATM phosphorylates another ATM molecule" corresponds to a trans-phosphorylation reaction (other phrasings also work, such as "Active ATM trans-phosphorylates itself")—i.e. one molecule of ATM phosphorylates another molecule of ATM—which produces the non-linearity necessary for a time delay. In contrast, "ATM phosphorylates itself" implies modification in cis, which is incapable of generating oscillations in the p53 network."

4.27 *Page 21:*

While the combinatorial complexity problem is addressed here the previous paragraph describes INDRA as a starting point for modeling. Models of this complexity are poor starting points and this must be should be addressed before INDRA can be considered a recommending starting point for modeling over diagrammatic inputs like symbiology or user-friendly packages such as COPASI. The time investment required to learn software such as COPASI/SimBiology/CellDesigner is not far greater than the learning required to understand the impact of phraseology on dynamics and in the corrent format they generate more elegant models as "starting points". Either address complexity of generated models as described above or clarify the expected use of the large models generated, I would not recommend them as a starting point for model construction but perhaps as tools to identify knowledge gaps, generate diagrams and target manual model construction. Intead highlight the importance of this approach in enabling automated, text-mined, model construction from the literature here.

We discuss the issue of combinatorial complexity in detail in GC2 and GC3 above. We also argue that the multitude of specific complexes assembled around BRAF (including Vemurafenib, RAS and MEK as constituents) is essential for the understanding of the resistance to Vemurafenib.

4.28 Page 25:

The following sentence is inaccurate: "molecular species are directly instantiated as variables and related to each other using one or more differential equations for each mass action reaction". Rather in dynamical systems modeling molecular species are each represented by a differential equation and the terms of the differential equations are determined by the reactions.

We have corrected this sentence, which is on page 27 of the revised manuscript, to now read "For example, in an ODE-based model, molecular species are directly instantiated as variables and related to each other using one or more differential equations containing terms determined by each mass action reaction".

4.29 Page 27:

Cytoscape offers importing from various databases and outputting SBML among other formats and should be mentioned here (Cline et al 2007 Nature Protocols). Cell Designer also enables input from a number of file formats including BIOPAX, simulation within Cell Designer and export as SBGN (Funahashi et al 2003) including plugins such as SBMLsqueezer which enable generation of rate equations from diagrams that can be created by easily without mathematical skills. Include these here.

We have extended paragraph 2 of the "Relationship to previous work" section of the discussion on page 29 of the revised manuscript, and included all the references suggested by the reviewer.

4.30 -Presentation and style

Good

-Trivial mistakes

Figure 4: Rule 1 of one step policy uses 'kc' parameter in diagram and 'kf' parameter in code.

We fixed this typo, now both the diagram and the code contain 'kc' as the parameter.

References

- Allen, James, Will de Beaumont, Lucian Galescu, and Choh Man Teng. 2015. "Complex Event Extraction Using DRUM." In *ACL-IJCNLP*, 1–11. Beijing, China.
- Babur, Özgün, Bülent Arman Aksoy, Igor Rodchenkov, Selçuk Onur Sümer, Chris Sander, and Emek Demir. 2014. "Pattern Search in BioPAX Models." *Bioinformatics* 30 (1). Oxford Univ Press: 139–40.
- Babur, Özgün, Ugur Dogrusoz, Emek Demir, and Chris Sander. 2009. "ChiBE: Interactive Visualization and Manipulation of BioPAX Pathway Models." *Bioinformatics*. doi:10.1093/bioinformatics/btp665.
- Batchelor, Eric, Alexander Loewer, Caroline Mock, and Galit Lahav. 2011. "Stimulus-Dependent Dynamics of p53 in Single Cells." *Molecular Systems Biology* 7 (488): 488. doi:10.1038/msb.2011.20.
- Carvunis, Anne Ruxandra, and Trey Ideker. 2014. "Siri of the Cell: What Biology Could Learn from the iPhone." *Cell*. doi:10.1016/j.cell.2014.03.009.
- Chen, William W, Mario Niepel, and Peter K Sorger. 2010. "Classic and Contemporary Approaches to Modeling Biochemical Reactions." *Genes & Development* 24 (17). Cold Spring Harbor Lab: 1861–75.
- Chen, William W, Birgit Schoeberl, Paul J Jasper, Mario Niepel, Ulrik B Nielsen, Douglas A Lauffenburger, and Peter K Sorger. 2009. "Input-Output Behavior of ErbB Signaling Pathways as Revealed by a Mass Action Model Trained against Dynamic Data." *Molecular Systems Biology* 5: 239.
- Deeds, Eric J., Jean Krivine, Jérôme Feret, Vincent Danos, and Walter Fontana. 2012. "Combinatorial Complexity and Compositional Drift in Protein Interaction Networks." *PLoS ONE* 7 (3). doi:10.1371/journal.pone.0032032.
- Hlavacek, William S., James R. Faeder, Michael L. Blinov, Alan S. Perelson, and Byron Goldstein. 2003. "The Complexity of Complexes in Signal Transduction." *Biotechnology and Bioengineering*. doi:10.1002/bit.10842.
- Hoffmann, Alexander, Andre Levchenko, Martin L Scott, and David Baltimore. 2002. "The I κ B-NF- κ B Signaling Module: Temporal Control and Selective Gene Activation." *Science (New York, N.Y.)* 298 (5596): 1241–45. doi:10.1126/science.1071914.
- O'Hara, Laura, Alessandra Livigni, Thanos Theo, Benjamin Boyer, Tim Angus, Derek Wright, Sz Hau Chen, et al. 2016. "Modelling the Structure and Dynamics of Biological Pathways." *PLoS Biology* 14 (8). doi:10.1371/journal.pbio.1002530.
- Purvis, J. E., K. W. Karhohs, C. Mock, E. Batchelor, A. Loewer, and G. Lahav. 2012. "p53 Dynamics Control Cell Fate." *Science* 336 (6087): 1440–44. doi:10.1126/science.1218351.
- Purvis, Jeremy E., and Galit Lahav. 2013. "Encoding and Decoding Cellular Information through Signaling Dynamics." *Cell*. doi:10.1016/j.cell.2013.02.005.
- Raue, Andreas, C. Kreutz, T. Maiwald, J. Bachmann, M. Schilling, U. Klingmüller, and J. Timmer. 2009. "Structural and Practical Identifiability Analysis of Partially Observed Dynamical Models by Exploiting the Profile Likelihood." *Bioinformatics* 25 (15): 1923–29. doi:10.1093/bioinformatics/btp358.
- Rinaldi, Fabio, Tilia Renate Ellendorff, Sumit Madan, Simon Clematide, Adrian van der Lek, Theo Mevissen, and Juliane Fluck. 2016. "BioCreative V Track 4: A Shared Task for the Extraction of Causal Network Information Using the Biological Expression Language." *Database* 2016: baw067. doi:10.1093/database/baw067.
- Sneddon, Michael W, James R Faeder, and Thierry Emonet. 2011. "Efficient Modeling, Simulation and Coarse-Graining of Biological Complexity with NFsim." *Nature Methods* 8 (2): 177–83. doi:10.1038/nmeth.1546.

Suderman, Ryan, and Eric J. Deeds. 2013. "Machines vs. Ensembles: Effective MAPK Signaling through Heterogeneous Sets of Protein Complexes." *PLoS Computational Biology* 9 (10). doi:10.1371/journal.pcbi.1003278.

Valenzuela-Escarcega, Marco A., Hahn-Powell Gus, Hicks Thomas, and Mihai Surdeanu. 2015. "A Domain-Independent Rule-Based Framework for Event Extraction." *Proceedings of the 53rd Annual Meeting of the ACL and the 7th International Joint Conference on NLP*, 127–32.

Thank you again for submitting your work to Molecular Systems Biology. We have now heard back from the three referees who agreed to evaluate your study. As you will see below, they think that most of the previously raised issues have been satisfactorily addressed. Reviewers #1 and #4 raises a few remaining minor concerns, which we would ask you to address by modifying the text in a minor revision.

 REVIEWER REPORTS

Reviewer #1:

The revision by Gyori and colleagues is very thorough and commendable, addressing all critical issues raised in the first round of reviews. The following are minor questions / corrections that the authors may consider:

- (i) p.17: The statement '... which produces the non-linearity necessary for a time delay...' may not be clear to the target audience - perhaps a short explanation is warranted.
- (ii) p.17: 'Such ambiguities are picked up by INDRA ...' - strictly speaking, INDRA does not flag ambiguities but it may be helpful in identifying them.
- (iii) p.20: 'assembled into 28 PySB rules and 99 differential equations;' - at this point, it may be good to explain the pertinent aspect of combinatorial expansion (e.g., which species are responsible, ...).
- (iv) p.20: 'amount of active RAS depends only on the amount [of] EGF'
- (v) p.22, second para: In discussing assembly policies, I suggest adding statements on the what the scope of individual policies is (application to Statements individually), and how consistency across a model could be ensured (e.g., avoiding cases in which the same enzyme acting on different substrates is captured differently, with potential consequences such as biased / incorrect mass balances).
- (vi) p.30: '(i) issues relating to the reading [of] natural language by external NLP systems'
- (vii) Fig. 7: Caption for panel (D) appears to be missing.

Reviewer #3:

The authors have provided a careful exegesis and response to my comments. I have no further criticisms. This study represents a valuable contribution to the field of computational model assembly.

Reviewer #4:

The authors have done a good job responding to the points raised by us and the other reviewers and greatly improved the manuscript as a result. My only remaining concern is with "GC5: Accessibility of the approach to non-experts." The authors agree in response to multiple reviewers that they have not shown that the approach presented here improves "accessibility", and I agree that it is difficult and unwise to try and do so. I believe INDRA is not necessarily more accessible than alternative approaches, but it has other distinguishing advantages as described in the text (enabling collaboration while maintaining rigorous and transparent translation from natural language into mechanism and forming part of a future pipeline from literature to model). Accessibility is not emphasized in the text but does appear in the abstract. The authors state that their approach "increases the accessibility and transparency of models for the broader biology community." I believe this sets an expectation for a software solution that overcomes some of the hurdles facing the broader biological community in constructing

mathematical models rather than exposing such ambiguities and difficulties to ensure robust and accurate collaboration. All reviewers and readers will have expectations better set for the important work contained in the paper if the word accessibility is removed from the abstract. Consider: "increases the transparency of models for collaboration with the broader biology community." If that is addressed I recommend publication.

2nd Revision - authors' response

27 October 2017

Response to reviewers

Reviewer 1:

The revision by Gyori and colleagues is very thorough and commendable, addressing all critical issues raised in the first round of reviews. The following are minor questions / corrections that the authors may consider:

(i) p.17: The statement '... which produces the non-linearity necessary for a time delay...' may not be clear to the target audience - perhaps a short explanation is warranted.

We added an explanation, and the corresponding section on page 17 now reads:

"ATM trans-phosphorylation represents a form of positive feedback since the flux through the phosphorylation reaction increases with the concentration of the reaction product, namely, phosphorylated ATM. As described in detail by Novák and Tyson, positive feedback in such reaction mechanisms can create the "dynamical hysteresis" necessary for a time delay (Novák & Tyson, 2008)"

(ii) p.17: 'Such ambiguities are picked up by INDRA ...' - strictly speaking, INDRA does not flag ambiguities but it may be helpful in identifying them.

We modified the corresponding sentence to read: "Such ambiguities are propagated by INDRA and can be identified by the user at multiple (intermediate) stages of the extraction and assembly process"

(iii) p.20: 'assembled into 28 PySB rules and 99 differential equations;' - at this point, it may be good to explain the pertinent aspect of combinatorial expansion (e.g., which species are responsible, ...).

We have added an additional sentence to this paragraph to note that "65 of the 99 species in the model involve complexes assembling around EGFR, which are generated by the biochemical reactions described in the sentences that constitute the word model."

(iv) p.20: 'amount of active RAS depends only on the amount [of] EGF'

We fixed the phrase to read "amount of active RAS depends only on the amount of EGF" on page 20.

(v) p.22, second para: In discussing assembly policies, I suggest adding statements on the what the scope of individual policies is (application to Statements individually), and how consistency across a model could be ensured (e.g., avoiding cases in which the same enzyme acting on different substrates is captured differently, with potential consequences such as biased / incorrect mass balances).

We agree with the reviewer's point and chose the last paragraph of page 14 as the most appropriate place to discuss this issue. The end of the paragraph has been extended with:

"Assembly policies can be applied globally to the model or to specific *Statement* types (e.g., a one-step policy for *IncreaseAmount Statements* vs. a two-step policy for *Phosphorylation Statements*). In the current implementation of INDRA, policies cannot be applied to individual *Statements*; this extension is feasible but would require that the user maintain consistency among *Statements* involving the same reactants."

(vi) p.30: '(i) issues relating to the reading [of] natural language by external NLP systems'

We fixed the phrase to read "issues relating to the reading of natural language" on page 30.

(vii) Fig. 7: Caption for panel (D) appears to be missing.

We added a caption for Figure 7 (D) as follows:

"(D) Simulation results of Boolean models assembled from natural language under different inhibitor conditions. The "Basic model" contains the links shown in Figure 7A; the "Extended model" contains the extensions shown in Figure 7C. Each trace represents the activity of JUN in the presence of growth factors averaged over 100 stochastic simulations (see Methods)."

Reviewer 4:

The authors have done a good job responding to the points raised by us and the other reviewers and greatly improved the manuscript as a result.

My only remaining concern is with "GC5: Accessibility of the approach to non-experts." The authors agree in response to multiple reviewers that they have not shown that the approach presented here improves "accessibility", and I agree that it is difficult and unwise to try and do so. I believe INDRA is not necessarily more accessible than alternative approaches, but it has other distinguishing advantages as described in the text (enabling collaboration while maintaining rigorous and transparent translation from natural language into mechanism and forming part of a future pipeline from literature to model). Accessibility is not emphasized in the text but does appear in the abstract. The authors state that their approach "increases the accessibility and transparency of models for the broader biology community." I believe this sets an expectation for a software solution that overcomes some of the hurdles facing the broader biological community in constructing mathematical models rather than exposing such ambiguities and difficulties to ensure robust and accurate collaboration. All reviewers and readers will have expectations better set for the important work contained in the paper if the word accessibility is removed from the abstract. Consider: "increases the transparency of models for collaboration with the broader biology community."

If that is addressed I recommend publication.

Following the suggestion by Reviewer 4, we have removed "accessible" from the abstract, and revised the sentence in question to read: "The use of natural language makes the task of developing a model more efficient and it increases model transparency, thereby promoting collaboration with the broader biology community."

Corresponding Author Name: Peter K. Sorger

Manuscript Number: MSB-17-7651